# ER remodelling is a feature of ageing and depends on ER-phagy

Eric K. F. Donahue ⓘ [1], Nathaniel L. Hepowit ⓘ [1], Elizabeth M. Ruark[1], Alexandra G. Mulligan ⓘ [1], Brennen Keuchel[1], Nicholas D. Urban[2], Li Peng[1], Stedman Stephens[3], Derek J. Johnson[1], Natalie S. Wallace ⓘ [4], Lauren P. Jackson ⓘ [3,4], Mark H. Ellisman[5], Rafael Arrojo e Drigo[6], Andrew W. Folkmann[3], Matthias C. Truttmann ⓘ [2], Jason A. MacGurn ⓘ [1] & Kristopher Burkewitz ⓘ [1] ✉

The endoplasmic reticulum (ER) comprises an array of subdomains, each defined by a characteristic structure and function. Although altered ER processes are linked to age-onset pathogenesis, it is unclear whether shifts in ER structure or dynamics underlie these functional changes. Here we establish ER structural and functional remodelling as a conserved feature of ageing across yeast, *Caenorhabditis elegans* and mammals. Focusing on *C. elegans* as the exemplar of metazoan ageing, we reveal striking age-related reductions in ER volume across diverse tissues and a morphological shift from rough sheets to tubular ER. This morphological transition corresponds with large-scale shifts in ER proteome composition from protein synthesis to lipid metabolism, a phenomenon conserved in mammalian tissues. We show that Atg8 and ULK1-dependent ER-phagy drives age-associated ER remodelling through tissue-specific factors, including the previously uncharacterized ER-phagy regulator TMEM-131 and the IRE-1–XBP-1 branch of the unfolded protein response. Providing support for a model where ER remodelling is adaptive, diverse lifespan-extending paradigms downscale and remodel ER morphology throughout life. Furthermore, mTOR-dependent lifespan extension in yeast and worms requires ER-phagy, indicating that ER remodelling is a proactive and protective response during ageing. These results reveal ER-phagy and ER dynamics as pronounced, underappreciated mechanisms of both normal ageing and age-delaying interventions.

The endoplasmic reticulum (ER) houses various cellular processes fundamental to cell homeostasis and healthy ageing, including protein quality control, lipid biosynthesis, autophagy initiation, carbohydrate metabolism and calcium signalling[1,2]. The ER also coordinates intracellular signalling and the morphological dynamics of other diverse membrane-bound organelles through a malleable array of inter-organelle contact sites[3–5]. By integrating a variety of nutrient-sensing pathways with its roles in facilitating secretion, ER status is also a key determinant of healthy intercellular signalling in metazoans[6,7]. This centrality of the ER in cell and organismal physiology highlights the organelle as a critical driver of healthy versus pathological ageing trajectories.

Regarding ER roles in ageing, the ER unfolded protein response (UPR) has traditionally held the focus of the field. Because the UPR activates expression of ER chaperones, membrane remodelling factors and other genes capable of boosting ER functional capacity[1,8], the UPR is generally considered protective, but its role is nuanced. Consistent with a protective role, a number of ER chaperones and

**Fig. 1 | Endogenous ER subdomain markers for live imaging in *C. elegans*.**
**a**, Super-resolution image of a cortical region of the *C. elegans* hypodermis demonstrating expected subdomain enrichment of the ER reporters RET-1::mKate2 (ER tubules and sheet edges) and GFP::SEC-61.B (patches of ER sheets). Images are representative of seven animals from three independent experiments. Scale bar, 2 μm. **b**, Confocal imaging of the ER marked by RET-1::GFP (ER tubules and sheet edges) and TRAP-1::mCherry (rough ER) in *C. elegans* hypodermis. The merged images represent maximum intensity (top right; max IP) and 3D (bottom) projections. Arrowheads indicate RET-1::GFP enriched on the edges of TRAP-1-labelled rough ER sheets. Images are representative of 50 animals from three independent experiments. Scale bars, 5 μm. **c**, Confocal imaging of GFP::SEC-61.B and RET-1::mKate2 depicting differential enrichment of ER subdomain markers between tissues. Images are representative of 100 animals from three independent experiments. Scale bars, 100 μm (top) and 10 μm (bottom).

acute UPR inducibility decline with age[9–12]. However, UPR signalling becomes maladaptive at advanced ages when unresolved ER stress triggers chronic UPR activation, in turn promoting inflammation and metabolic dysfunction[1,6,13,14]. Intriguingly, multiple models of longevity reset UPR signalling to low baselines while maintaining high stress resistance[15,16]. How this seemingly paradoxical resilience of long-lived animals is achieved at the cell biology level remains unclear but suggests a model where cells pre-empt damage by constraining

protein flux through the ER. Interestingly, global protein synthesis also declines during the natural ageing process[17]. Although this downshift is sometimes viewed as a symptom of wider proteostasis defects, that assumption is challenged by the observation that reduced protein synthesis is both a common correlate of life-extending interventions and sufficient to extend lifespan[18,19]. These observations raise the possibility that age-related declines in cell and ER protein synthesis may reflect adaptive remodelling.

While UPR stress-signalling pathways are critical during ageing, ER morphodynamics represent an understudied mechanism for cells to shape ER metabolic outputs and maintain homeostasis. The functions of the ER are compartmentalized into distinct structural subdomains[2,4,20,21]. Stacked ER sheets are the primary sites of protein synthesis and maturation. The sheets are studded with ribosomes and enriched with protein translocation and quality control machinery[2,4]. The sheet morphologies are optimized for accommodating more polyribosome docking and supporting protein maturation through larger luminal volume ratios[2,4]. Conversely, highly curved tubular structures provide less ribosome docking, and are associated with membrane and lipid-droplet biogenesis and interactions with other organelles[2,5]. The relative abundance of sheet versus tubule subdomains varies widely between cell types and corresponds to the functional specialization of each cell[2]. Furthermore, (patho)physiological fluctuations can also trigger altered ER morphology[22–24] and, reciprocally, defects in ER-shaping factors are linked to the pathogenesis of neurodegeneration and metabolic disease[24–27]. However, whether ER morphological transitions are age-dependent remains underexplored.

Live-cell imaging and reconstitution assays have revealed the factors shaping ER subdomains to be dynamic and multifactorial. The relative abundances of sheet- versus tubule-shaping proteins is one method for cell control of ER morphology[4,28]. These include conserved reticulon- and receptor expression enhancing (REEP)/Yop1-family proteins, which insert hairpin-like reticulon homology domains (RHDs) into the ER membrane to stabilize highly curved tubules and sheet edges[2,28,29]. Functional partners also affect ER shaping; for instance, extensive ribosome docking contributes to the stabilization of ER sheet subdomains[30]. Although less explored, an alternative pathway for cells to remodel ER structure and function involves selective targeting of ER subdomains for degradation via ER-phagy. ER-phagy occurs through multiple pathways including autophagosomal engulfment in macro-ER-phagy, vesicle-based ER-to-lysosome-associated degradation and direct endolysosomal engulfment of ER fragments via micro-ER-phagy[31,32]. Studies in yeast and tissue-culture models have revealed a suite of ER-phagy receptors that coordinate the selective targeting of ER subdomains for degradation[31,32]. Including single transmembrane and multiple RHD proteins, these canonical receptors reside in the ER membrane, bind autophagy machinery through LC3-interacting regions (LIRs) and contain intrinsically disordered regions that contribute to membrane fragmentation[32,33]. Intriguingly, previous screens designed to discover ER-phagy receptors utilized nutrient deprivation and mTOR inhibitors[31,34,35], interventions known for extending lifespan, as the contexts for inducing high-level ER-phagy. Furthermore, key ER-phagy pathways share some molecular machineries with macro-autophagy, which recurrently emerges as essential for diverse pathways of lifespan extension. However, whether ER-phagy plays a role in defining ageing outcomes remains underexplored.

Here, using *Caenorhabditis elegans*, yeast and mammalian models, we reveal the dynamics of the ER during normal ageing and roles for ER-phagy in lifespan determination. By visualizing ER subdomains across *C. elegans* tissues in vivo, we find that ER remodelling is among the earliest and most profound biological changes in cells after animals reach adulthood. The ER networks in diverse cell types of aged animals exhibit substantial declines in ER mass, particularly of rough ER, thus revealing a cell biological link to the reduced proteostatic capacity during ageing. We demonstrate that ER remodelling is driven by ER-phagy

and that ER-phagy activation seems to be an adaptive step coordinated with changes in the proteostasis network. Consistent with this model, we show that diverse lifespan-extension paradigms in *C. elegans* proactively promote alternative ER morphologies and reveal causal roles for ER-phagy factors in lifespan extension. Finally, we demonstrate in *C. elegans* that age-onset ER-phagy occurs through conserved tissue-specific pathways involving an ER collagen chaperone and the UPR. Together these findings illuminate important roles for ER-phagy and ER morphological dynamics in modulating age-dependent decline.

## Results

### In vivo imaging of ER dynamics in adult *C. elegans*

Many ER proteins are enriched in specific structural and functional subdomains[20,28,36]. To study ER dynamics during ageing, we developed transgenic *C. elegans* strains with labels capable of distinguishing rough or tubular ER. SEC-61.B is an essential component of the ER translocon enriched in rough ER sheets[28]. Genomic green fluorescent protein (*GFP*) fusion produced a native marker, GFP::SEC-61.B, with enrichment in rough ER and relative exclusion from established smooth ER subdomains, such as muscle sarcoplasmic reticulum (Extended Data Fig. 1a). Importantly, the use of native markers circumvents the loss of subdomain fidelity associated with overexpressed and/or heterologous markers[28,37] (Extended Data Fig. 1b). To visualize tubular ER subdomains, we inserted mKate2 at the carboxy terminus of the sole *C. elegans* reticulon RET-1. Among the most highly enriched smooth ER proteins[20], reticulons are conserved hairpin-domain proteins that stabilize membrane curvature in ER tubules and sheet edges[29]. Super-resolution imaging of these endogenous GFP::SEC-61.B and RET-1::mKate2 proteins revealed canonical SEC-61.B-enriched rough ER sheets linked by a RET-1-enriched network of tubules (Fig. 1a). To further validate the labelling, we swapped fluorophores on RET-1 from mKate2 to GFP and combined this with an alternative translocon subunit[38], TRAP-1::mCherry. Super-resolution imaging of *trap-1::mCherry; ret-1::GFP* animals revealed sheet structures bearing distinct enrichment of RET-1 on tubules and sheet edges, with TRAP-1 concentrated on the sheet faces (Fig. 1b). When co-labelled, SEC-61.B and TRAP-1 exhibited strong co-localization (Extended Data Fig. 1c,d).

Interestingly, the relative abundance of SEC-61.B or RET-1 labelling in different tissues also reflected expected rough or smooth ER specialization (Fig. 1c). Enrichment of smooth ER tubules, observed as high levels of RET-1, was noted in neuronal projections, muscle sarcoplasmic reticulum and the smooth muscle-like spermathecae, whereas the hypodermis and intestine, which execute robust collagen and lipoprotein secretion, were highly enriched for SEC-61.B-labelled rough ER[36] (Fig. 1c). Together, these results provide support for native labelling of these ER proteins as a reliable platform for monitoring ER subdomain dynamics in vivo across tissues in *C. elegans*.

### Ageing is associated with declines in total ER mass and remodelling of ER-structure function

To investigate whether ER morphology is dynamic across lifespan, we focused first on the hypodermis, a thin and metabolically flexible cell type well-suited for visualizing fine ER structures in vivo. In young adults, SEC-61.B marked a dense network of cisternal structures extending throughout the cell, with relatively few resolvable RET-1-enriched tubules (Fig. 2a). We then aged animals to day 7 of adulthood, a stage coinciding with the onset of age-dependent functional decline. Consistent with previous descriptions of ageing and progeria models[39,40], the SEC-61.B-marked nuclear envelope developed deep invaginations and other structural distortions at this age (Fig. 2a). However, the peripheral ER network was also markedly remodelled (Fig. 2a–c), despite being relatively unexplored in ageing contexts. Chief among the changes was a striking decline in the total amount of ER but the morphology was also altered, with SEC-61.B localizing to small sporadic clusters connected by a largely tubular network (Fig. 2a–c).

To quantify these changes and avoid co-expression artifacts, we imaged single-tagged GFP::SEC-61.B and RET-1::GFP animals individually (Extended Data Fig. 2a–h). Both markers showed substantial declines in intensity (approximately 70%) and footprint, or area of the cell occupied by the ER (25–30%), alongside a twofold increase in the perimeter-area ratio (PAR; Extended Data Fig. 2c–h). These findings indicate reduced organelle volume and content with age combined with a shift towards a more tubular network. We confirmed these trends independently of fluorescence imaging using epitope tags and immunoblotting in whole-animal lysates (Extended Data Fig. 2i–l). Interestingly, these blots revealed a more pronounced decline in RET-1 levels across the organism than for SEC-61.B. This result probably reflects certain tissue-specific contributions to the overall protein levels in whole-animal lysates (compare RET-1 and SEC-61.B levels in the intestine, discussed in the next section) but the low levels of RET-1 still raised the question of how a largely tubular network would be supported by tubule-promoting proteins in aged animals. We thus tested the complementary ER tubulating factor in *C. elegans*, YOP-1 (ref. 41). In contrast to RET-1, the levels of YOP-1 remained largely stable (Extended Data Fig. 2m,n), indicating potential compensation between tubulating factors. Finally, we investigated whether luminal, widely distributed ER proteins also exhibit similar age-dynamics. The GRP78 (also known as BiP) orthologues HSP-3::mScarlet and HSP-4::mScarlet exhibited similar expression and localization trends with age (Extended Data Fig. 3a–d). On the other hand, the levels of the mitochondrial outer membrane marker TOMM-20::mCherry remained stable, demonstrating the specificity of this dynamic to the ER (Extended Data Fig. 3e,f). Collectively these results provide support for a major decline in ER mass while highlighting that individual ER-shaping factors are differentially impacted by ageing.

Next, we tracked the kinetics and ultrastructural impacts of ER remodelling. Daily imaging of GFP::SEC-61.B animals revealed a 36% decline in intensity over the first three days of adulthood, revealing ER remodelling to be an early stage transition (Fig. 2d,e). We also imaged the ER in ageing wild-type animals via transmission electron microscopy (TEM). Consistent with fluorescence imaging, young animals exhibited densely packed stacks of rough ER sheets that gave way to sparse tubular networks in the hypodermis with a reduced footprint and increased PAR (Fig. 2f–h and Extended Data Fig. 4a–d).

These age-dependent morphological changes may correspond to functional shifts from proteostasis of the rough ER sheets to those that are more tubule-associated, such as lipid metabolism[2]. Notably, this potential transition would be consistent with widely reported declines in global protein synthesis with age[17,42]. To explore this model, we mined recent proteomic datasets examining the age-dependent proteome of *C. elegans* across tissues[43]. We identified ER-resident proteins associated with either proteostasis or lipid metabolism and compared their levels in young and aged animals (Fig. 2i–k and Supplementary Table 1). Although the proteostasis network of the ER undergoes broadscale decline, mirroring the loss of rough ER (Fig. 2i,j), proteins involved in lipid metabolism generally stayed consistent with roughly half increasing with age (Fig. 2i,k). We conclude that ER morphological shifts mirror a functional shift from proteostasis to lipid metabolism based on the declines in total ER abundance and shift in ER-structure function from tightly packed stacks of rough ER cisternae towards diffuse tubular networks.

### Age-dependent ER remodelling occurs across distinct tissue types

We also aimed to determine which age-dependent changes in the ER network might be generalizable across cell types. Consistent with the secretory functions of the intestine, intestinal cells possess dense, rough ER networks[36] (Fig. 1c). Similar to the hypodermis, the intestine experienced dramatic loss of SEC-61.B at the protein level, reduction in ER footprint and an increase in PAR (Fig. 3a–d). Both the loss of ER mass and sheet-like

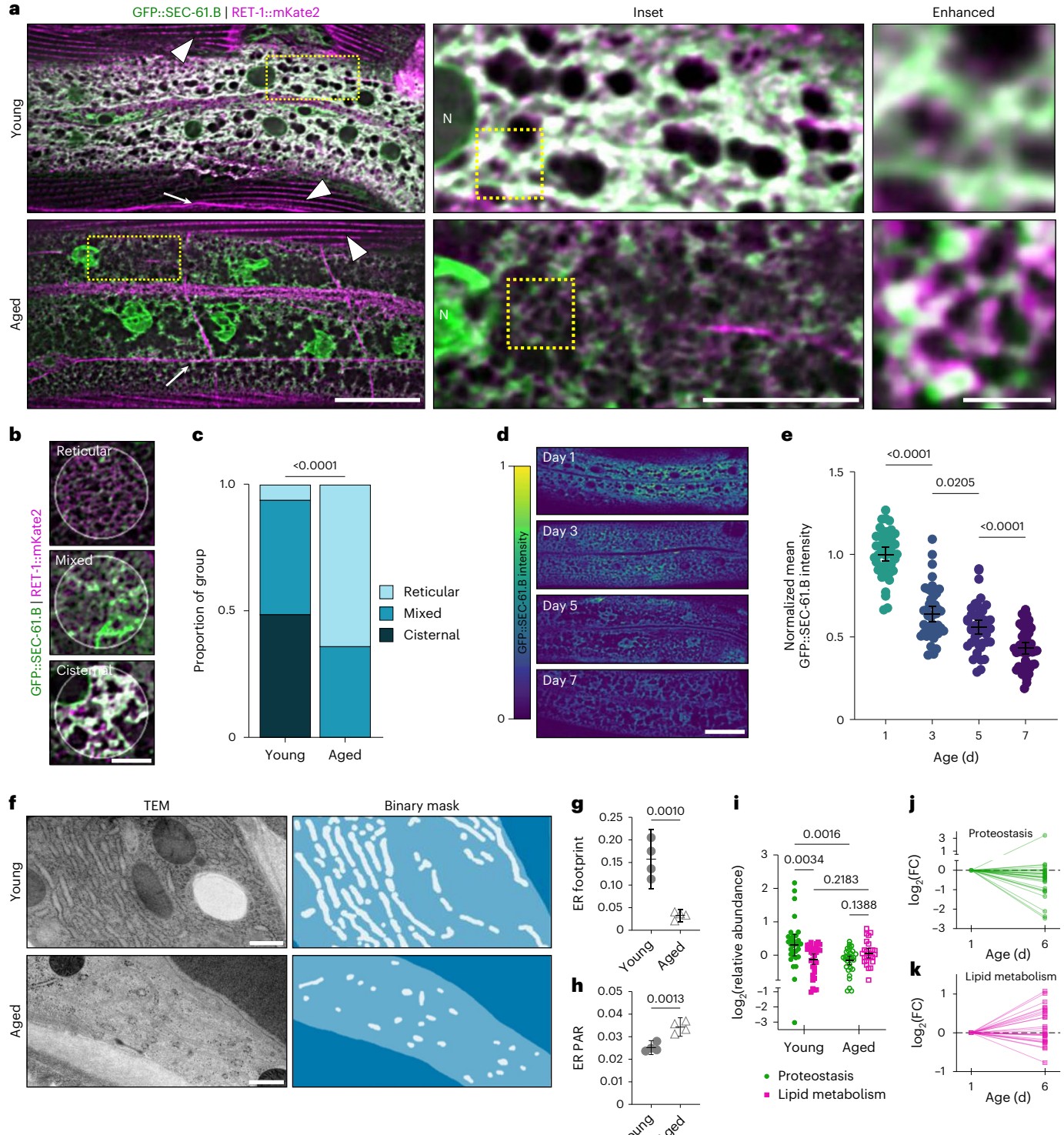

**Fig. 2 | Remodelling of the peripheral ER during ageing in *C. elegans*.**
**a**, Fluorescence imaging of GFP::SEC-61.B; RET-1::mKate2 in young
(day 1; top) and aged (day 7; bottom) adults. Arrows and arrowheads indicate
smooth ER subdomains of neuronal projections and muscle sarcoplasmic
reticulum, respectively. Insets: magnified views of the boxed regions depicting
hypodermal ER network. Scale bars, 25 μm (left), 10 μm (middle) and 2.5
μm (right). N, nucleus. **b,c**, Representative examples of ER morphology
(**b**) and categorical morphologic analysis (**c**) of *n* = 123 regions from 42 young
worms and 120 regions from 44 aged worms, pooled from two independent
experiments. **b**, Scale bar, 5 μm. **c**, Analysed using a one-tailed $\chi^2$ test.
**d,e**, Representative images (**d**) and quantification of time course imaging of
hypodermal GFP::SEC-61.B (**e**) in day 1, 3, 5 and 7 adults. **d**, Scale bar, 10 μm.
**e**, Analysed using a one-way analysis of variance (ANOVA), followed by a
two-tailed Šidák post-hoc test; *n* = 45 worms per age, pooled from three

replicates. **f**, TEM images (left) and binary masks (right) of the ER (white) in
the hypodermis (lighter blue) of young (top) and aged (bottom) worms. Scale
bars, 500 nm. **g,h**, Hypodermal ER footprint (**g**) and PAR (**h**). Analysed using
a two-tailed Student's *t*-test; *n* = 4 worms per group. **i–k**, The ER proteome in
young (day 1) and aged (day 6) worms reveals relative fold change (FC) shifts
in functional subdomain composition during ageing. Data points represent
the relative abundance of a specific protein within each functional category
(proteostasis, *n* = 31 proteins in both groups; lipid metabolism, *n* = 32 proteins
in young worms and 26 proteins in aged worms). Analysed using a restricted
maximum-likelihood mixed-effects model, followed by a post-hoc two-tailed,
uncorrected Fisher's least-significant-difference test. **j,k**, Age-dependent
trajectories were normalized to abundance in young animals. **c,e,g–i**, *P* values
are shown. Error bars indicate the mean ± 95% confidence interval (CI). Source
numerical data are provided.

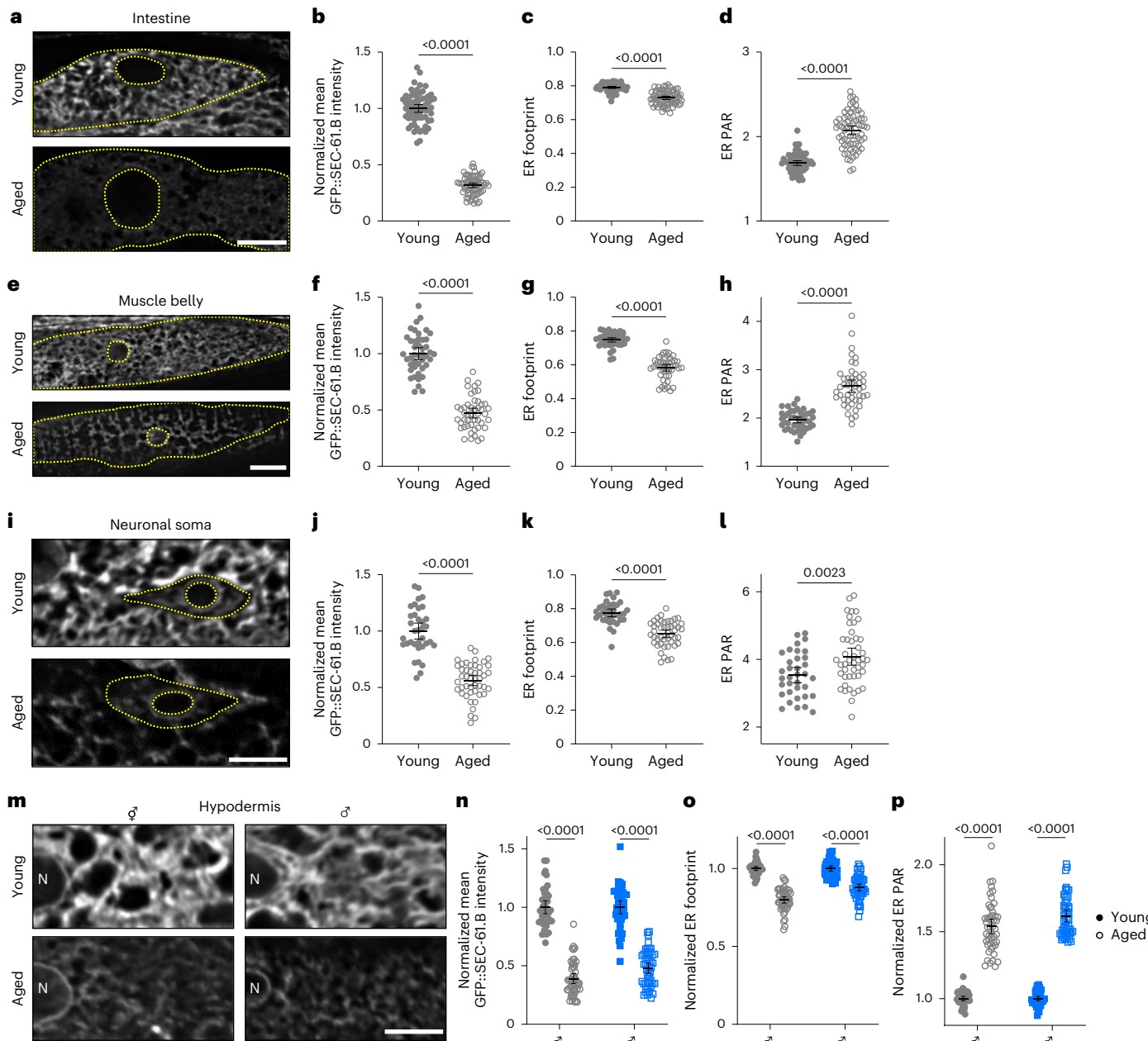

**Fig. 3 | Age-onset ER remodelling occurs across tissues and sex. a**, Imaging of intestinal GFP::SEC-61.B in young (top) and aged (bottom) worms. **b–d**, Normalized mean intensity (**b**), footprint (**c**) and PAR (**d**) of intestinal GFP::SEC-61.B in young and aged worms; *n* = 64 young and 70 aged worms pooled from three replicates. **e**, Imaging of GFP::SEC-61.B in the body-wall muscle belly of young (top) and aged (bottom) worms. **f–h**, Normalized mean intensity (**f**), footprint (**g**) and PAR (**h**) in muscle of young and aged GFP::SEC-61.B worms; *n* = 44 young and 47 aged worms pooled from three replicates. **i**, Imaging of GFP::SEC-61.B in the soma of the ALM neuron of young (top) and aged (bottom) worms. **j–l**, Normalized mean intensity (**j**), footprint (**k**) and PAR (**l**) of GFP::SEC-61.B in the ALM soma of young and aged worms; *n* = 35 young and 46 aged worms pooled from three replicates. **m**, Imaging of hypodermal GFP::SEC-61.B in young (top) and aged (bottom) hermaphrodite (♀; left) and male (♂; right) worms. N, nucleus. **n–p**, Mean intensity (**n**), footprint (**o**) and PAR (**p**) of hypodermal GFP::SEC-61.B in young and aged worms, normalized within each sex to day 1. Hermaphrodites, *n* = 41 young and 50 aged worms; males, *n* = 43 young and 49 aged worms; pooled from three replicates. **a,e,i**, Dotted yellow lines demarcate the cell membrane and nuclear envelope. **a,e,i,m**, Scale bars, 10 μm. **b–d,f–h,j–l,n–p**, Analysed using a two-tailed Student's *t*-test; *P* values are shown. Error bars indicate the mean ± 95% CI. Source numerical data are provided.

structures was supported by TEM imaging (Extended Data Fig. 4e,f). Furthermore, we found that male worms exhibited similar age-dependent declines in SEC-61.B intensity, footprint and morphology in both the hypodermis (Fig. 3m–p) and intestine (Extended Data Fig. 5a–d), revealing that ER loss is independent of egg production[44]. Finally, muscle cells (Fig. 3e–h) and neurons (Fig. 3i–l) also exhibited declines in SEC-61.B intensity, ER footprint and shifts in ER morphology, as measured in the muscle belly and soma, respectively. These results reveal that ER remodelling occurs across most major tissue types in *C. elegans*.

Although aspects of age-dependent ER dynamics seem to be generalizable across many cell types, we also observed tissue-specific changes. For example, RET-1::GFP labels the young intestinal ER but is virtually undetectable in aged intestine (Extended Data Fig. 5e,f), exhibiting a more pronounced decline than SEC-61.B in this tissue. This near-complete loss of RET-1 in the intestine probably explains why the RET-1 western blots measured from whole animals decline more sharply than SEC-61.B (Extended Data Fig. 2). In addition, muscle cells and neurons each harbour specialized smooth ER subdomains within

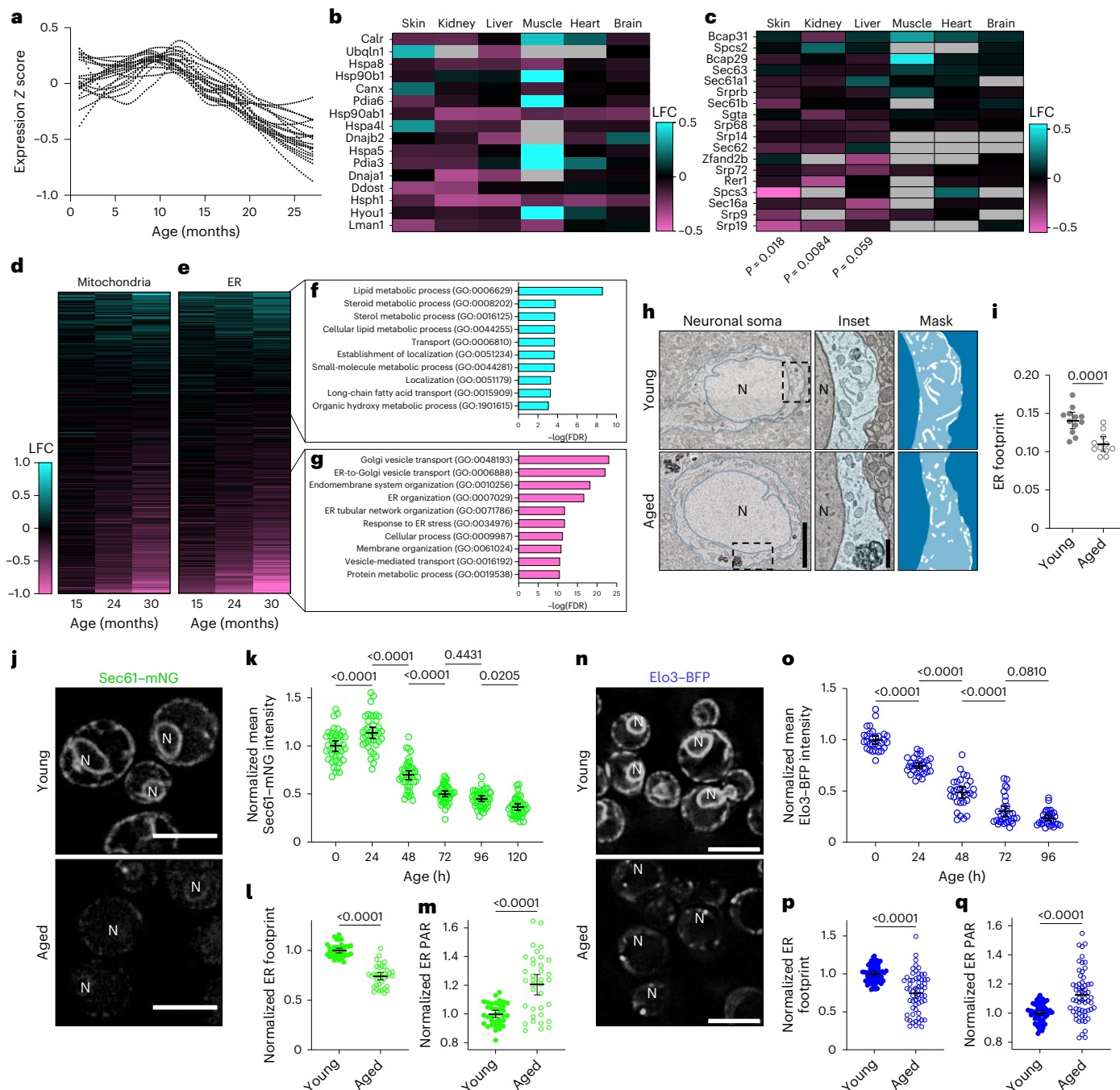

**Fig. 4 | Age-dependent ER remodelling is evolutionarily conserved. a**, Protein processing in the ER. Age-trajectories of individual transcripts with roles in ER protein processing averaged across all tissues. **b**,**c**, Log₂-transformed fold change (LFC) in protein levels for ER protein processing (**b**) and protein localization (**c**) in ER proteins in aged (30 months) versus young (6 months) mice. Analysis of ER protein localization was performed using the Fisher's exact test. **d**,**e**, Fold change in every protein localized to mitochondria (**d**) and the ER (**e**) in mice of different ages. **f**,**g**, Top ten enriched gene categories among ER proteins that were up- (**f**) or downregulated (**g**) in the skin with age. **h**, Representative scanning electron microscopy images of L2 motor cortex neurons in young (6 months; top) and aged (18 months; bottom) male mice. Magnified views (middle) and binary masks of ER (right; white) of the boxed regions are provided. Scale bars, 2.5 μm (left) and 500 nm (right). **i**, ER footprint in murine cortical motor neurons. Analysed using a two-tailed Student's *t*-test; *n* = 12 young and 11 aged cells. **j**, Imaging of Sec61–mNeonGreen (mNG) during chronological ageing in young

(0 h; top) and aged (72 h; bottom) yeast cells. Scale bars, 5 μm. **k**, Sec61–mNG intensity during chronological ageing in yeast. Analysed using a one-way ANOVA, followed by a two-tailed post-hoc Šidák test; *n* = 40 cells per group, pooled from four replicates. **l**,**m**, Normalized ER footprint (**l**) and PAR (**m**) in young and aged yeast cells. Analysed using a two-tailed Student's *t*-test; *P* values are shown; *n* = 36 cells per group, pooled from three replicates. **n**, Imaging of Elo3–blue fluorescent protein (BFP) during chronological ageing in young (top) and aged (bottom) yeast cells. Scale bars, 5 μm. **o**, Elo3–BFP intensity during chronological ageing in yeast. Analysed using a one-way ANOVA, followed by a two-tailed post-hoc Šidák test; *n* = 30 cells per group, pooled from three replicates.
**p**,**q**, Normalized ER footprint (**p**) and PAR (**q**) in young and aged yeast cells. Analysed using a two-tailed Student's *t*-test; *n* = 60 cells per group, pooled from replicates. **i**,**k**–**m**,**o**–**p**, *P* values are shown. Error bars indicate the mean ± 95% CI. N, nucleus. Source numerical data are provided.

anatomically distinct regions of the cell: the myofilament lattice and neurites, respectively[36,45]. Providing support for a model where rough ER subdomains are preferentially lost during ageing, these exclusively smooth ER subdomains seem to be resistant to age-related ER declines, with an approximate 15% decline in RET-1::GFP intensity and 8% decline in sarcoplasmic-reticulum footprint (Extended Data Fig. 5g–i) compared with declines of approximately 55% and 22% in the ER of the muscle belly (Fig. 3e–h). Contrasting every other tissue, neurite ER exhibits notable increases in RET-1 levels during ageing (Extended Data Fig. 5j–l). We observed an approximately twofold increase in RET-1::wrmScarlet intensity with age in representative sensory (ALN and PLM) and motor (DA7) neurons, whereas the cytosolic GFP levels remained static, indicating that the increases in RET-1 are independent of broadscale neuronal changes (Extended Data Fig. 5k,l).

Collectively, these results highlight that age-dependent ER remodelling is a ubiquitous phenomenon in *C. elegans* but can take distinct forms depending on the tissue. Although a decline in ER volume is the most generalizable effect, notable exceptions such as neurites exist. Furthermore, multiple major metabolic tissues exhibit structural changes consistent with preferential turnover of rough ER and associated sheet-like morphologies during ageing.

## Age-dependent ER remodelling is conserved from yeast to mammals

We next set out to determine whether age-onset remodelling of ER is evolutionarily conserved by examining eukaryotes ranging from yeast to mammals. First, the Tabula Muris Senis project[46,47] recently performed RNA sequencing analysis of diverse mouse tissues and revealed tissue-specific and animal-wide gene expression trajectories at the messenger RNA level. When gene expression changes were averaged across tissues to identify universal ageing signatures, one of the most strongly downregulated clusters was enriched for genes functioning in 'protein processing in the ER' (KEGG mmu0414; Fig. 4a).

Furthermore, a parallel atlas of proteomic changes across several mouse tissues during ageing enabled us to determine whether these mRNA trajectories correspond with protein-level remodelling[48]. To mine this dataset of samples taken from whole-tissue lysates, we extracted data from the same set of genes involved in ER protein processing identified by Schaum and colleagues[46], comparing young (6 months) and aged (30 months) animals. We found consistent downregulation of these genes at the protein level in multiple mouse tissues, most notably skin, liver and kidney (Fig. 4b). Unbiased gene ontology enrichment analysis of these complete proteomic datasets additionally revealed another signature of rough ER remodelling focused on the translocon and protein targeting to the ER, again revealing the strongest and most consistent downregulation in skin, liver and kidney (Fig. 4c). Given that mitochondrial dysfunction is an established hallmark of ageing, we next extracted genes annotated for localization to either the mitochondria or ER to compare the scope and magnitude of changes, focusing on the skin, which showed the most consistent downregulation of ER translocon factors. The mitochondrial and ER proteomes both exhibited substantial progressive alterations relative to the complete cellular proteome ($P = 3.21 \times 10^{-7}$ and $9.11 \times 10^{-8}$, respectively) and the downregulation in each case was similar in scope and magnitude (47% ER proteins downregulated versus 45% of mitochondrial proteins with false detection rate (FDR) < 0.05; Fig. 4d,e). On examination of the categories of proteins that are either down- or upregulated in the ER, we again found notable similarities to *C. elegans*. The downregulated proteins were enriched for functions in secretion, protein metabolism and ER organization, whereas the smaller group of upregulated genes were enriched for roles in lipid metabolism (Fig. 4f,g). Although the proteomes of the liver and kidney revealed smaller-scale downregulation of the ER proteome, the downregulated proteins were similarly enriched for roles in secretion (Extended Data Fig. 5m,n). Intriguingly, ER proteins with roles in autophagy were consistently upregulated in

these tissues (Extended Data Fig. 5m,n). Overall, these results uncover the downregulation of rough ER-associated protein synthesis and secretion as a widespread aspect of the ageing process in mammals.

Finally, we aimed to visualize ER network changes with age in other species. Early neuropathological studies previously suggested that outstanding characteristics of the ageing rodent brain include a decrease in ribosomes and progressive 'disorganization' of ER cisternae in neurons[49,50], leading us to investigate neurons in mouse brain sections by scanning electron microscopy. We examined cortical neurons of three- and 18-month-old mice, and confirmed statistically significant declines in ER volume with age, even at this relatively early stage of ageing (Fig. 4h,i). Surprisingly, we did not consistently identify stereotypical stacked cisternae even in young animals and thus, were unable to confirm previous observations of cisternal disorganization at this particular age (PAR unchanged, $P = 0.71$). Finally, we also examined the ER of chronologically aged yeast cells using fluorescence microscopy. We labelled distinct ER proteins known to localize broadly across ER subdomains, Sec61 and Elo3. Both exhibited early and striking declines in intensity and footprint (Fig. 4j–q), overall mirroring findings in *C. elegans*, although ultrastructural analysis of the uniquely organized yeast ER requires further study. Together, these results demonstrate that functional and structural ER remodelling during ageing is a conserved phenomenon, however, extrapolation across diverse mammalian types necessitates deeper dedicated analyses.

## ER-phagy drives age-onset ER remodelling

Next, we set out to identify mechanisms promoting ER remodelling during ageing. Given the pronounced loss of both ER protein and membrane mass during ageing, we reasoned that autophagic and/or lysosomal degradation processes may be involved. Current models suggest that multiple distinct pathways act in concert to deliver ER components to lysosomes: (1) macro-ER-phagy, which requires both canonical autophagy initiation factors like ULK1/Atg1/*unc-51* and LC3/GABARAP/Atg8 lipidation; (2) micro-ER-phagy, independent of both ULK1 and Atg8; (3) RecovER-phagy, a micro-ER-phagy variant that utilizes Atg8 and (4) vesicular ER-to-lysosome pathways, which also generally depend on Atg8 lipidation[31,32]. To help distinguish between these routes, we fed animals double-stranded RNA (dsRNA) targeting either Atg1/*unc-51* or Atg8/*lgg-1*. Autophagy inhibition via depletion of Atg1/*unc-51* and Atg8/*lgg-1* had relatively small effects on ER size and shape in young adults, indicating that autophagy-mediated turnover plays a limited role in shaping the ER through development (Fig. 5a–d). However, knockdown of Atg1/*unc-51* strongly suppressed age-onset changes in SEC-61.B protein levels, footprint and PAR (Fig. 5a–d), and this suppression of age-effects was virtually complete during Atg8/*lgg-1* impairment (Fig. 5a–d). These trends generally remained consistent between ER markers (Extended Data Fig. 6a–d) and tissues (Extended Data Fig. 6e–l), although Atg8/*lgg-1* RNA interference (RNAi) resulted in modest elevations in RET-1 levels in young animals (Extended Data Fig. 6a,b) and especially strong elevation of RET-1 levels in the aged intestine (Extended Data Fig. 6i,j). Notably, ULK1/*unc-51* knockdown also caused the appearance of expanded sheet-like structures (Fig. 5a and Extended Data Fig. 6a), which were not apparent in the Atg8/*lgg-1* animals. Together, these findings suggest that macro-ER-phagy, dependent on ULK1/UNC-51, is a primary route for ER-sheet turnover, whereas Atg8-dependent ULK1-independent (vesicular and RecovER-phagy) routes contribute to bulk ER degradation and help balance network morphology.

The ER is an important source of membrane for autophagosome formation as well as a potential target of autophagosomes via ER-phagy, and both of these roles could potentially result in reduced ER mass[31,51,52]. To confirm the ER itself is targeted as cargo for ER-phagy, we employed TRAP-1::mCherry. Unlike GFP and mKate2 labels, mCherry is resistant to lysosomal acidity and degradation, enabling analysis of lysosomal targeting[53]. In young adult animals, TRAP-1::mCherry

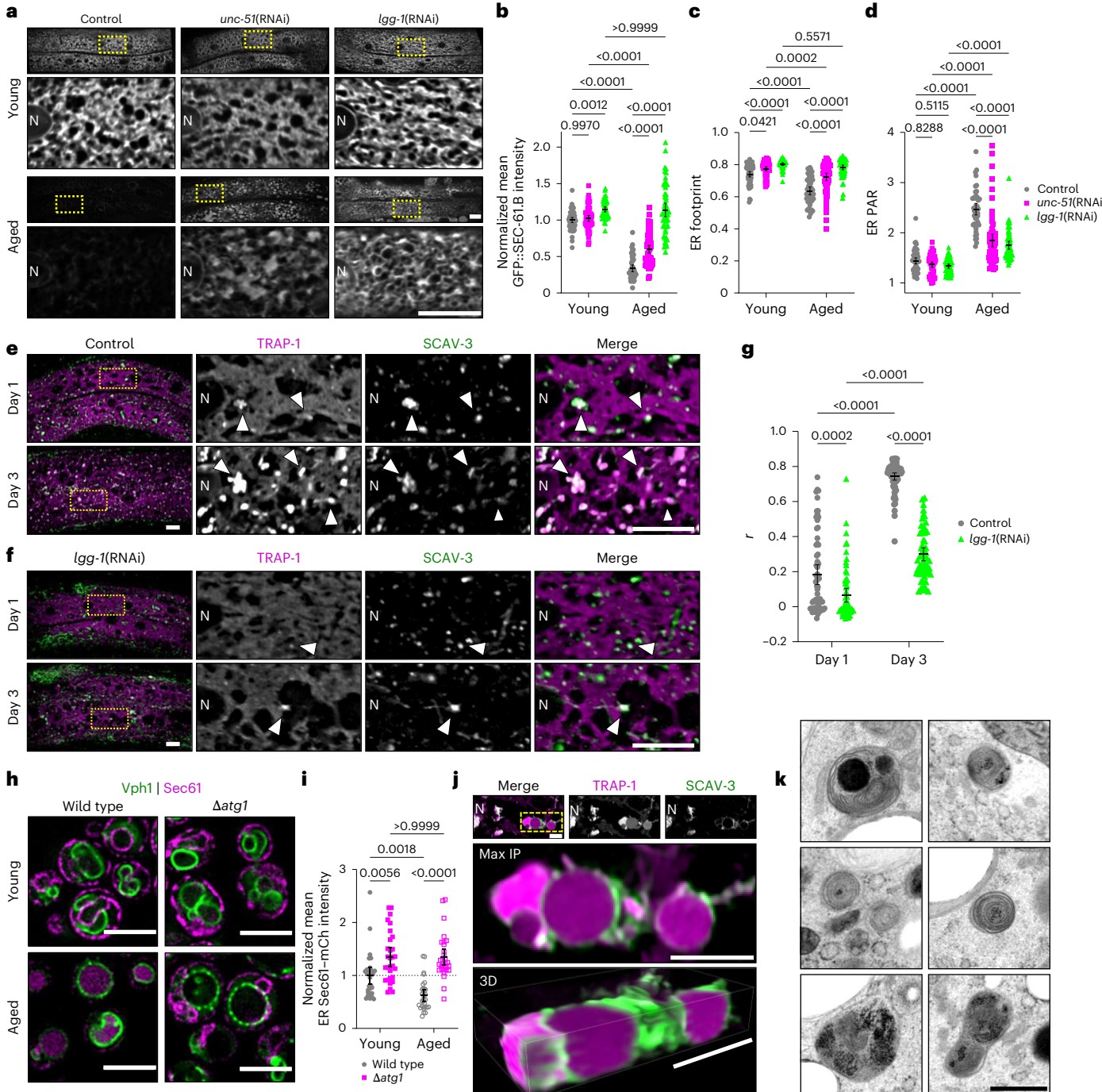

**Fig. 5 | ER-phagy drives age-onset ER remodelling. a**, Imaging of hypodermal GFP::SEC-61.B in young (top) and aged (bottom) control worms and worms fed dsRNA targeting *unc-51* or *lgg-1* (*unc-51*(RNAi) and *lgg-1*(RNAi), respectively). **b–d**, Hypodermal GFP::SEC-61.B normalized mean intensity (**b**), footprint (**c**) and PAR (**d**) in young and aged control, *unc-51*(RNAi) and *lgg-1*(RNAi) worms. Control (empty vector), *n* = 57 young and 60 aged worms; *unc-51*(RNAi), *n* = 60 young and 67 aged worms; *lgg-1*(RNAi), *n* = 54 young and 56 worms; pooled from three replicates. **e,f**, Hypodermal TRAP-1::mCherry and SCAV-3::GFP in day 1 (top) and day 3 (bottom) control (**e**) and *lgg-1*(RNAi) (**f**) worms. Arrowheads mark regions of co-localization. **a,e,f**, Scale bars, 10 μm. **g**, Pearson's correlation coefficient (*r*) between mCherry and GFP in the hypodermis of control and *lgg-1*(RNAi) worms at day 1 and day 3. Control (empty vector), *n* = 65 day 1 and 69 day 3 worms; *lgg-1*(RNAi) *n* = 63 day 1 and

58 day 3 worms; pooled from three replicates. **h**, Imaging of Sec61–mCherry and Vph1–mNG in young (top) and aged (bottom) control and mutant yeast cells with an *atg1* deletion (*Δatg1*). **i**, Mean cytoplasmic Sec61–mCherry intensity in control and *Δatg1* yeast at 0 and 72 h; *n* = 30 yeast per group, pooled from two replicates. **b–d,g,i**, Analysed using a two-way ANOVA, followed by a two-tailed post-hoc Šidák test; *P* values are shown. **j**, Representative fluorescence image (top) showing accumulation of ER reporter TRAP-1::mCherry within lysosomes (SCAV-3::GFP) during early ageing (day 3). The boxed region is displayed as maximum intensity (middle) and 3D (bottom) projections. **h,j**, Scale bars, 5 μm. **k**, Representative TEM images of autolysosomal compartments filled with characteristic multilamellar ER membrane whorls in aged (day 7) *C. elegans*. Scale bar, 500 nm. Error bars indicate the mean ± 95% CI. N, nucleus. Source numerical data are provided.

and GFP::SEC-61.B labels reveal the same ER network organization (Extended Data Fig. 1c,d). However, TRAP-1::mCherry uniquely begins accumulating in distinct puncta by day 3 of adulthood (Fig. 5e). Consistent with the age-onset formation of these puncta, we observed low

co-localization of TRAP-1::mCherry with lysosomal membrane marker, SCAV-3::GFP, in animals on the first day of adulthood and a dramatic increase of TRAP-1::mCherry in SCAV-3+ lysosomes by day 3 (Fig. 5e,g). Atg8/*lgg-1* depletion prevented the formation of TRAP-1 puncta and

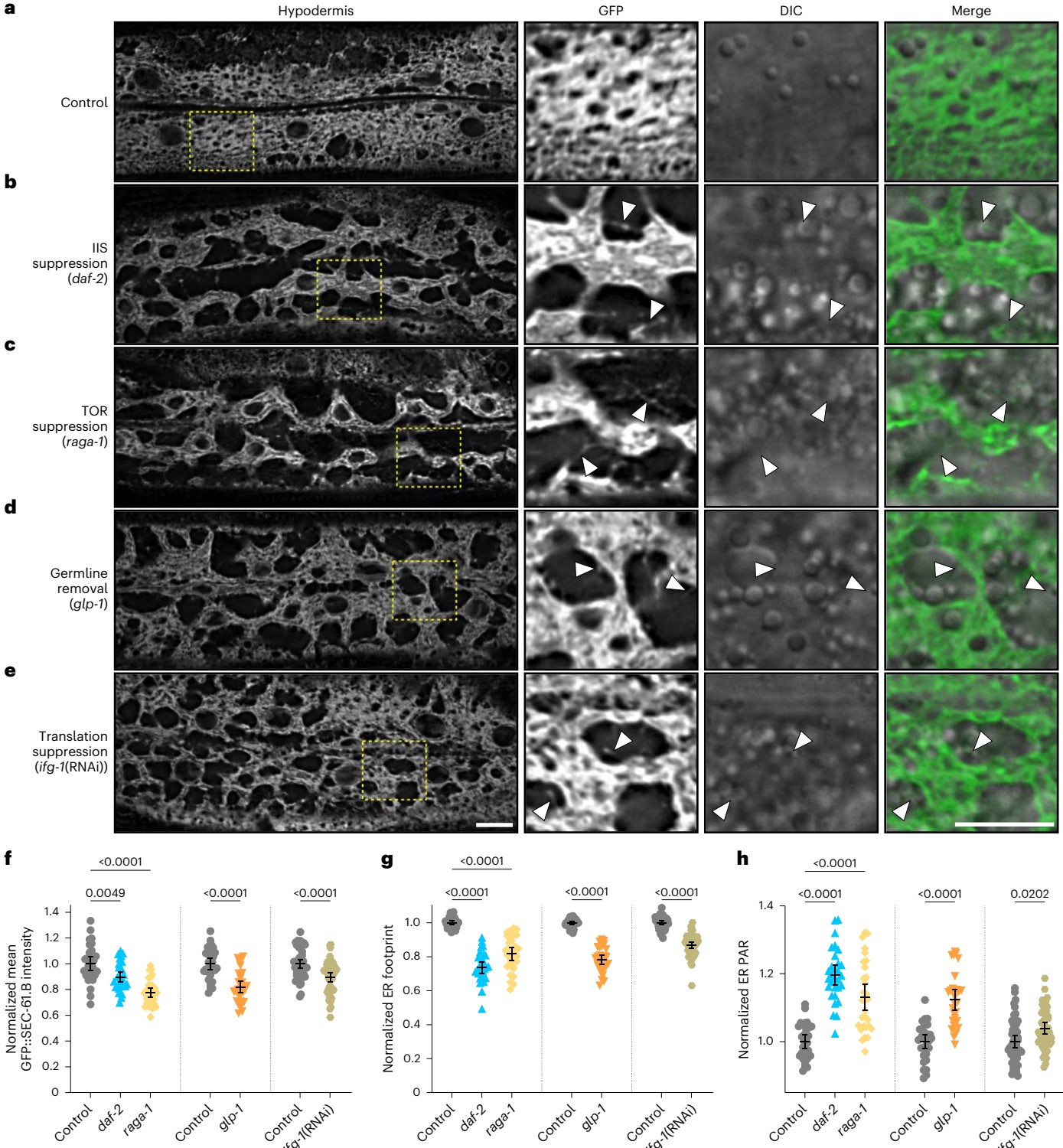

**Fig. 6 | ER remodelling is a common feature of diverse lifespan-extension paradigms. a–e**, Imaging of hypodermal GFP::SEC-61.B in young worms from control (**a**), insulin/insulin-like signalling (IIS) suppression (**b**), TOR suppression (**c**), germline removal (**d**) and translation suppression (**e**) conditions. Magnified views of the boxed regions are provided; arrowheads indicate sparse ER tubules; DIC, differential interference contrast. Scale bars, 10 μm. **f–h**, Normalized mean intensity (**f**), footprint (**g**) and PAR (**h**) of GFP::SEC-61.B relative to wild-type (control) worms. Control, IIS and TOR suppression, $n = 30$ for all groups; germline removal (25 °C), $n = 30$ for all groups; translation suppression, $n = 50$ control (empty vector) and 52 *ifg-1*(RNAi) worms; pooled from three replicates. Worms subjected to different experimental conditions are grouped with their respective controls. Error bars indicate the mean ± 95% CI. Analysed using a one-way ANOVA, followed by a two-tailed post-hoc Šidák test. Source numerical data are provided.

suppressed lysosomal targeting (Fig. 5f, g). Furthermore, the role of autophagy in driving age-onset ER loss is evolutionarily conserved, as the decline of Sec61 in yeast correlated with re-localization to the vacuole via an Atg1-dependent mechanism (Fig. 5h,i). We confirmed the appearance of ER components inside lysosomes through both three-dimensional (3D) fluorescence imaging (Fig. 5j) and TEM (Fig. 5k)

of *C. elegans*. Overall, these results are consistent with a model where activation of ER-phagy in adulthood promotes turnover and remodelling of the ER.

## ER remodelling is a common feature of lifespan extension

We next exploited *C. elegans* to investigate whether established paradigms of lifespan extension involve ER remodelling by ER-phagy. We selected a panel of mechanistically diverse interventions, that is, reduced insulin/insulin-like signalling (*daf-2(e1370)*), reduced mechanistic target of rapamycin (mTOR) signalling (*raga-1(ok386)*), germline removal (*glp-1(e2141)*) and inhibition of translation (dsRNA targeting *ifg-1*, *ifg-1*(RNAi)). Consistent with ER remodelling acting as an adaptive step in ageing, these life-extending interventions universally induced dramatic ER remodelling, even at the onset of adulthood (Fig. 6a–e). All interventions promoted substantial reductions in GFP::SEC-61.B intensity and footprint as well as increased PAR (Fig. 5f–h). Although these gross trends resemble changes during normal ageing, we also observed aspects unique to long-lived animals. In contrast to the evenly distributed rough ER sheets under normal conditions (Fig. 6a), cisternal ER networks were condensed into perinuclear regions with large peripheral regions devoid of translocon-labelled ER instead occupied by sparse tubular networks (Fig. 6b–e). This organization suggests enhanced functional compartmentalization of ER networks into distinct sheet- versus tubule-filled regions of the cell, which may correspond with remodelling of other organelle networks and the subcellular architecture more globally. We found that GFP::SEC-61.B intensity, footprint and PAR were restored to levels comparable to controls in animals fed Atg8/*lgg-1* dsRNA to impair autophagy (Extended Data Fig. 7a–d), indicating that ER remodelling in long-lived animals involves ER-phagy. However, the regionalized clustering of ER sheets persisted during Atg8/*lgg-1* inhibition (Extended Data Fig. 7a), indicating that this aspect of ER network remodelling in long-lived animals is independent of autophagy. Finally, ageing in these conditions resulted in additional declines in ER (Extended Data Fig. 7e–h), consistent with a model where life-extending interventions promote ER-phagy throughout life. Overall, these findings highlight ER network dynamics as an aspect of both life-extending interventions and the normal ageing process.

## Candidate screening reveals a role for TMEM-131 in age-onset ER remodelling

Mediators of selective ER-phagy are only beginning to be explored in *C. elegans*[54], so we next aimed to elucidate ER-centric processes and molecular mediators linked to turnover during ageing. We thus performed a candidate screen of 35 genes that are orthologues of established ER-phagy mediators or ER-resident proteins harbouring an LIR motif (Supplementary Table 2). As context for our screen, we employed mTOR/*raga-1*-pathway mutants due to their robust ER depletion even in young animals (Fig. 6c) and the utility of TOR impairment in previous genetic screens for ER-phagy receptors[34,55,56]. Surprisingly, examination of RET-1::GFP in this background demonstrated that neither canonical ER-phagy receptor orthologues nor the COPII components *sec-23*, *sec-24.1*, *sec-24.2* and *sec-31* (ref. 57) had any discernible effect on ER remodelling in the *raga-1* mutants. However, RNAi of *tmem-131* (*tmem-131*(RNAi)), a conserved ER transmembrane protein with a predicted LIR motif, resulted in amelioration of ER loss and an expansion of ER sheets (Fig. 7a). We thus investigated whether *tmem-131* promotes turnover of ER during ageing, and found that *tmem-131*(RNAi) restored SEC-61.B levels and the ER footprint of aged animals to youthful levels (Fig. 7b–e). These results indicate that *tmem-131* is required for ER clearance in ageing contexts.

The luminal N terminus of TMEM-131 bears a procollagen binding domain, and the cytoplasmic tail interacts with the transport protein particle (TRAPP) III complex via TRAPPC8/*trpp-8* to promote COPII trafficking of collagen cargoes[58]. However, bioinformatic predictions also indicate a cytosolic domain architecture reminiscent of established ER-phagy receptors[33,59], including intrinsically disordered regions[60,61] and a conserved LIR motif[62] (residues 1517–1520) outside the TRAPPIII-binding region[58] (Fig. 7d,e). Furthermore, TMEM-131 was recently proposed as a candidate autophagy receptor in human cells based on proteomic profiling of autophagosomes[63]. Considering procollagens are an established substrate for both basal and stress-induced ER-phagy[64,65], these features suggest TMEM-131 could link luminal ER collagen state to ER abundance (Fig. 7d).

We first tested whether collagen secretion defects could recapitulate the ER phenotypes of *tmem-131*. Although knockdown of TMEM-131's binding partner, TRAPPC8/*trpp-8*, replicates the collagen export defects of *tmem-131* (ref. 58), *trpp-8*(RNAi) failed to rescue ER mass similarly to *tmem-131* or Atg8/*lgg-1* depletion (Fig. 7b,c and Extended Data Fig. 8a,b). In addition, COPII components play key roles in collagen trafficking, but their impairment did not produce a notable rescue of the ER (Extended Data Fig. 8c). Knockdown of the collagen metallopeptidase *dpy-31* (ref. 66) also contrasted sharply with *tmem-131*, producing a mild reduction in SEC-61.B levels in young animals and a small shift in ER PAR in aged animals (Extended Data Fig. 8d–g). These results collectively argue against a model where general collagen export defects are sufficient to explain the expanded ER mass in ageing *tmem-131* animals.

The data instead illustrate a more specialized role for TMEM-131. Aligned with previous reports that procollagen stress is relieved by micro-ER-phagy and ER-to-lysosome-associated degradation-like ER-phagy[64,65], impairment of collagen export via either *tmem-131*(RNAi) or *trpp-8*(RNAi) elevated baseline TRAP-1 co-localization with lysosomes (Fig. 7f and Extended Data Fig. 8h). Consistent with this, immunoblots of GFP::LGG-1 revealed a pattern in *tmem-131* animals consistent with elevated autophagic flux, including reductions in total and lipidated LGG-1 with a twofold increase in free proteolyzed GFP (Extended Data Fig. 8i–l). This suggests that bulk autophagy is activated for procollagen quality control in the absence of *tmem-131*. However, despite the elevated baseline, *tmem-131* blunted the age-induced increase in ER trafficking to lysosomes to a similar extent as Atg8/*lgg-1* depletion (Fig. 7f,g) and reduced overall lysosomal trafficking of ER relative to both aged control and *trpp-8*(RNAi) worms (Extended Data Fig. 8h). This distinction suggests that TMEM-131 plays a specific role in the age-onset macro-ER-phagy that depletes ER volume, which can be at least partly uncoupled from procollagen quality control.

Providing support for the potential for direct recruitment of autophagy machineries, pull-down assays using the TMEM-131 cytosolic domain revealed clear binding with the human Atg8/*lgg-1* orthologue GABARAP (Extended Data Fig. 8l). Mutation of the bioinformatically identified LIR motif reduced human GABARAP binding by 25–30% (Extended Data Fig. 8m,n). Although single-LIR receptors generally exhibit stronger dependence, ER-phagy receptors containing multiple LIR motifs (for example, Rtnl3) can form multivalent interactions with the autophagy machinery and exhibit similar partial effects[67]. Following this result, we indeed discovered at least one additional LIR motif in the cytosolic domain on manual examination (1559: FMNL). To investigate whether TMEM-131 behaves similarly to canonical ER-phagy receptors in vivo, we labelled and imaged endogenous TMEM-131 under ER-phagy-inducing starvation conditions. At baseline, TMEM-131 seemed to localize throughout the ER, forming many foci that did not correspond well with autophagosomes and probably relate to roles in collagen secretion (Extended Data Fig. 8o). Starvation resulted in a clear decline in TMEM-131 levels and footprint, reflecting a global loss of ER, yet TMEM-131 co-localization with autophagosomal puncta increased (Extended Data Fig. 8o–q). Atg8/LGG-1 puncta were consistently present at the boundaries of TMEM-131-labelled ER, including examples of fully co-localized puncta indicative of autophagosomal recruitment (Extended Data Fig. 8o, inset). Finally, consistent with TMEM-131 regulating selective turnover of the ER, *tmem-131* depletion

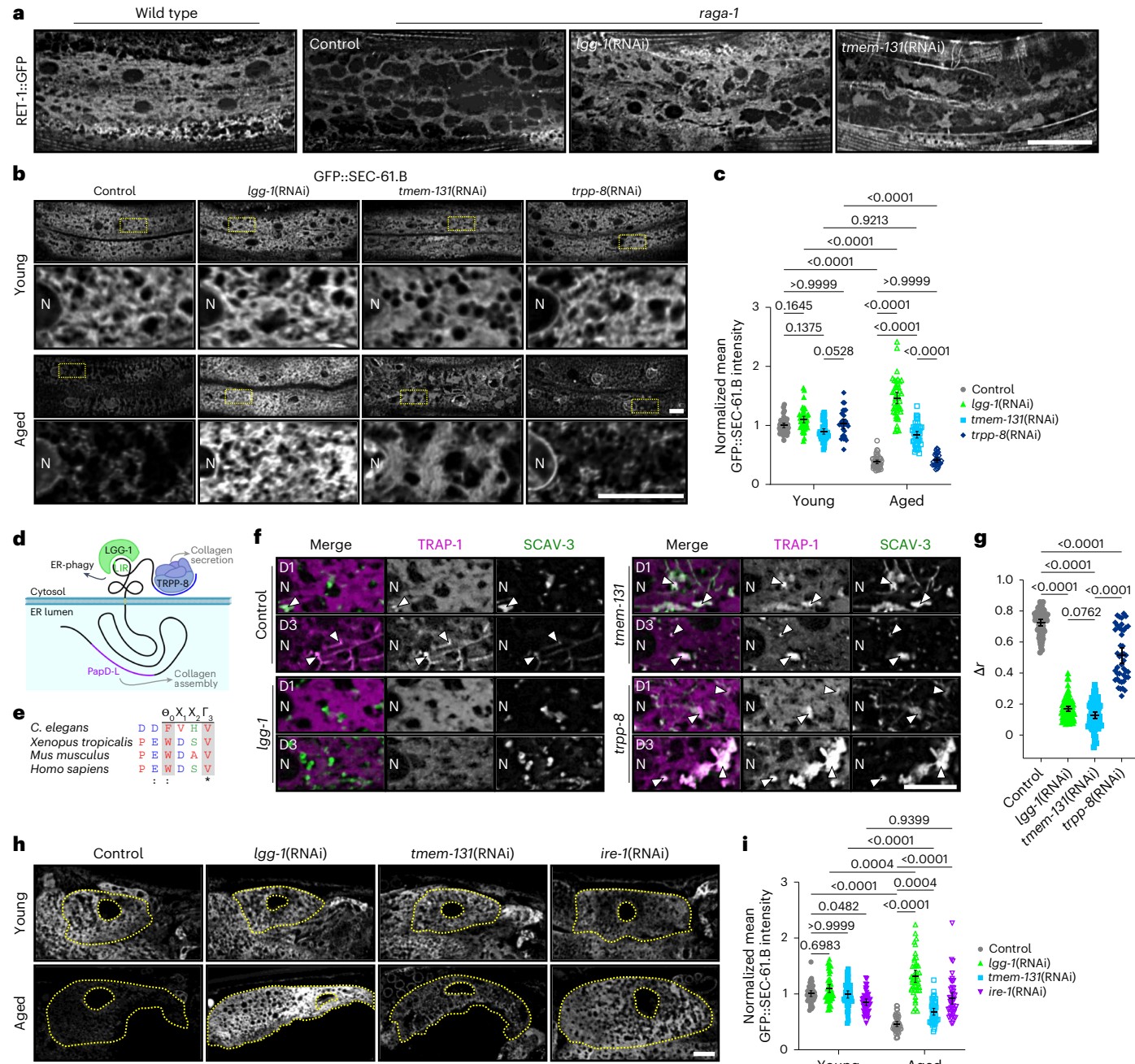

**Fig. 7 | Tissue-specific pathways promote age-onset ER turnover in metazoans.**
**a**, Imaging of hypodermal RET-1::GFP in wild-type and *raga-1* worms fed dsRNA targeting *lgg-1* or *tmem-131*, or empty vector control as part of ER-phagy regulator screen. **b**, Imaging of hypodermal GFP::SEC-61.B in young (top) and aged (bottom) control, *lgg-1*(RNAi), *tmem-131*(RNAi) and *trpp-8*(RNAi) worms. Magnified views of the boxed regions are provided beneath the main images. **c**, Normalized mean intensity of hypodermal GFP::SEC-61.B in young and aged control, *lgg-1*(RNAi), *tmem-131*(RNAi) and *trpp-8*(RNAi) worms. Control (empty vector), *lgg-1*(RNAi) and *tmem-131*(RNAi), *n* = 45 worms per group, pooled from three replicates; *trpp-8*(RNAi), *n* = 30 worms per group, pooled from two replicates. **d**, Cartoon of TMEM-131 and its functional domains. The amino-terminal PapD-L chaperone domain facilitates collagen assembly. The cytosolic LIR motif promotes ER-phagy through binding with Atg8 protein LGG-1/GABARAP. The C-terminal TRAPP III-interacting domain recruits the TRAPPIII complex via TRPP-8 to promote collagen secretion. **e**, Conservation of the putative LIR ($\Theta_0$-$X_1$-$X_2$-$\Gamma_3$) motif in TMEM-131 across species. Residues are colored by side chain class: negatively charged (blue), hydrophobic and proline (red) and hydrophilic

(green). **f**, Hypodermal TRAP-1::mCherry and SCAV-3::GFP in day 1 (D1) and 3 (D3) control, *lgg-1*(RNAi), *tmem-131*(RNAi) and *trpp-8*(RNAi) worms. Arrowheads mark ER accumulation in lysosomes. **g**, Relative change in Pearson's correlation coefficient ($\Delta r$) between mCherry and GFP in the hypodermis of control, *lgg-1*(RNAi), *tmem-131*(RNAi) and *trpp-8*(RNAi) worms during early ageing. Control (empty vector), *n* = 66 worms; *lgg-1*(RNAi), *n* = 76 worms; *tmem-131*(RNAi), *n* = 72 worms; *trpp-8*(RNAi), *n* = 44 worms; pooled from five replicates. **h**, Imaging of intestinal GFP::SEC-61.B in young (top) and aged (bottom) control, *lgg-1*(RNAi), *tmem-131*(RNAi) and *ire-1*(RNAi) worms. Dotted yellow lines demarcate the cell membrane and nuclear envelope. **i**, Normalized mean intensity of intestinal GFP::SEC-61.B in young and aged control, *lgg-1*(RNAi), *tmem-131*(RNAi) and *ire-1*(RNAi) worms. Control (empty vector), *n* = 42 young and 44 aged worms; *lgg-1*(RNAi), *n* = 45 young and 44 aged worms; *tmem-131*(RNAi) and *ire-1*(RNAi), *n* = 45 worms in all groups; pooled from three replicates. **c,g,i**, Analysed using a one-way (**g**) or two-way (**c,i**) ANOVA, followed by a two-tailed post-hoc Šidák test; *P* values are shown. Error bars indicate the mean ± 95% CI. **a,b,f,h**, Scale bars, 10 μm. N, nucleus. Source numerical data are provided.

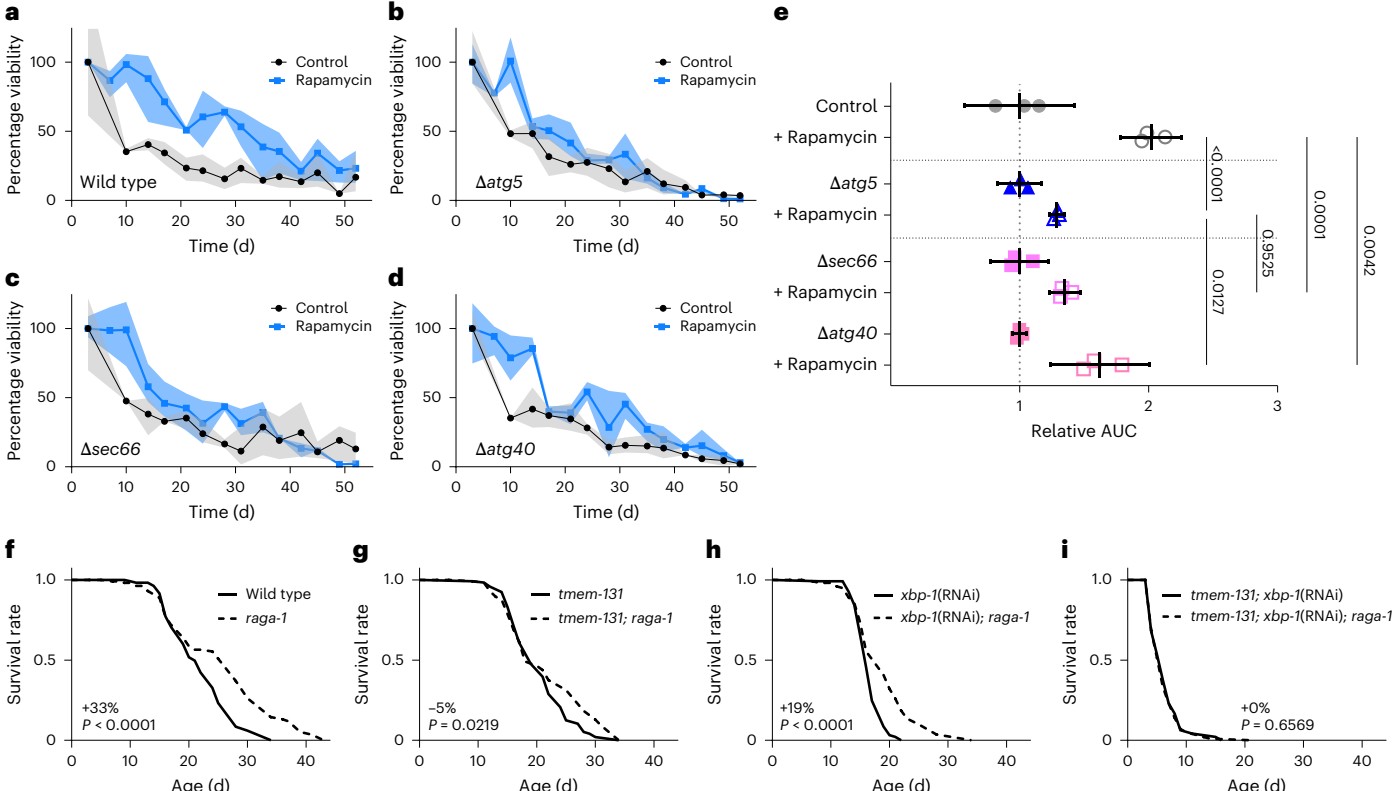

**Fig. 8 | ER-phagy is important for mTOR-dependent lifespan extension in yeast and *C. elegans*. a–d**, Chronological lifespan of yeast treated with control or 15 nM rapamycin in wild-type (**a**), macro-autophagy-mutant (Δ*atg5*; **b**) or ER-phagy-mutant (Δ*sec66*; **c**; and Δ*atg40*; **d**) strains. Three biological replicates per group. Lines indicate the group mean and shading indicates the group range. **e**, Relative area under the curve (AUC) of lifespan analyses in **a–d** normalized to control groups. Three biological replicates per group. Analysis using a one-way ANOVA, followed by a two-tailed post-hoc Šidák test; *P* values are shown. Error

bars indicate the mean ± 95% CI. **f–i**, Lifespan analysis of mTOR/*raga-1* mutant worms under control conditions (**f**), loss of either *xbp-1* (**g**) or *tmem-131* (**h**) alone, and loss of *tmem-131* and *xbp-1* in combination (**i**); *n* = 120 animals per condition, representative of three (**f,i**) or two (**g,h**) independent replicates. Analysed using the Mantel–Cox log-rank test; *P* values as well as the per cent median lifespan extension by *raga-1* are provided. Worm lifespan replicate data are available in Supplementary Table 3. Source numerical data are provided.

failed to promote an increase in mitochondrial volume during ageing (Extended Data Fig. 8r,s). Notably, age-associated mitochondrial fragmentation seemed to be accelerated by *tmem-131*(RNAi), which may suggest that early stage ER remodelling can promote downstream consequences for other organelle networks. Collectively, these data highlight the requirement of TMEM-131 for age-onset ER turnover. Structural similarities to ER-phagy receptors, evidence for Atg8 binding and recruitment to autophagosomes suggest that TMEM-131 may play direct roles in the targeting of ER by the autophagy machinery. Given that declines in global collagen and matrix secretion are a conserved facet of ageing[68], our data may point to a model where TMEM-131 promotes ER turnover when collagen clients are depleted, thus coordinating ER demand with supply.

**TMEM-131 and IRE-1–XBP-1 act in distinct tissues to promote age-onset ER turnover**

Because we observed ER-phagy-based remodelling of the ER across many tissue types in the worm, we investigated whether TMEM-131's role extends to other tissues. In contrast to the hypodermis, however, we observed relatively little effect of *tmem-131*(RNAi) on intestinal ER during ageing (Fig. 7h,i and Extended Data Fig. 9a,b). We therefore returned to the panel of candidate ER-phagy receptors that we initially screened in the hypodermis to test whether an alternate mediator may act in the intestine. We again recovered a single hit from our panel that in this case seemed to fully rescue the age-dependent loss of intestinal ER: the highly conserved, central player in UPR inositol-requiring enzyme (IRE)-1 (Fig. 7h,i and Extended Data Fig. 9a,b). *Ire-1* had no impact on the

age-dependent changes of the hypodermal ER (Extended Data Fig. 9c–f), just as *tmem-131* had little impact on the intestine (Fig. 7h,i). Follow-up experiments revealed that knockdown of the IRE-1 target, the soluble transcription factor XBP-1, also rescues ER loss during ageing in the intestine (Extended Data Fig. 9g–j). Conversely, parallel branches of the UPR, *atf-6* and PERK/*pek-1*, did not demonstrate roles in age-induced ER remodelling (Extended Data Fig. 10). Collectively, these results indicate that the IRE-1–XBP-1 axis promotes age-dependent ER turnover and remodelling, probably through downstream signalling rather than generalized ER stress or a direct role for IRE-1 as an ER-phagy receptor. Together, these findings indicate that the physiological mediators of ER-phagy are not universal but may instead align with the predominant functions of the ER in each tissue. Here TMEM-131 bridges collagen secretion and ER-phagy in the hypodermis, which is responsible for supporting the collagen-based cuticle, whereas *ire-1–xbp-1* performs similarly in the intestine, a central hub of UPR signalling in *C. elegans*[1].

**ER-phagy promotes mTOR-dependent longevity in yeast and *C. elegans***

To investigate causal roles of ER-phagy in ageing and lifespan, we turned to yeast and *C. elegans* models. In yeast, previously validated ER-phagy-specific adaptors allow us to experimentally decouple ER-phagy from bulk macro-autophagy. Given that mTOR inhibition exhibited the strongest ER remodelling phenotype in *C. elegans* (Fig. 6) and robustly induces both ER-phagy and lifespan extension in yeast[34,56,69,70], we investigated whether ER-phagy mediators are important for longevity in rapamycin-treated cells. Alongside

macro-autophagy-deficient *atg5*-deletion mutants (Δ*atg5*) as positive controls, we utilized deletion mutants of the canonical receptor for ER-phagy of peripheral ER (Δ*atg40*)[55] as well as deletion mutants of *sec66* (Δ*sec66*), which we selected based on its selective localization in ER membranes and its experimentally verified impairment of ER-phagy but not macro-autophagy[34]. Strikingly, the Δ*sec66* mutants phenocopied the complete suppression of lifespan of macro-autophagy-deficient Δ*atg5* mutants and the Δ*atg40* mutants similarly exhibited a blunted rapamycin effect (Fig. 8a–e). We also tested the role of the *tmem-131* and *ire-1–xbp-1* ER-phagy-inducing pathways in long-lived *C. elegans* harbouring mutations in the mTOR activator *raga-1*. Given the tissue-specific effects of both *tmem-131* and *ire-1–xbp-1* on ER remodelling, we suspected that multitissue ER-phagy impairment via the combined loss of both pathways would exhibit the strongest impact on lifespan. Consistent with this prediction, we observed a statistically significant suppression of longevity when either *tmem-131* or *xbp-1* were impaired alone (Fig. 8f–h), with a much more complete suppression when impaired in combination (Fig. 8i). The combined loss of *tmem-131* and *xbp-1* resulted in a notably short-lived animal regardless of mTOR/*raga-1* status. Besides emphasising the importance of ER-phagy in animal health, this dramatic result may also reflect secondary interactions between the two pathways[58]. Collectively, our results indicate that ER-phagy plays a key role in mTOR-dependent longevity.

## Discussion

The morphological dynamics of the ER have received little attention in the context of ageing. Here we established tools in *C. elegans* for high-resolution live imaging of ER networks in ageing metazoans, which revealed profound shifts in ER network morphology that are driven by ER-phagy. Across a variety of tissues, we consistently found a decrease in ER protein levels and cellular ER volume, and a structural shift from densely packed sheets to diffuse tubular networks. The ER content also declined in yeast and mammalian systems, and proteomic atlases of the ageing process in worms and mammals showed that age-onset collapse in ER proteostasis function is a broadly conserved aspect of the ageing process. We found that Atg8-dependent ER-phagy is the key mechanism driving turnover and remodelling of the ER network during ageing. A targeted screen for mediators in *C. elegans* revealed that the physiological triggers of ER-phagy in an ageing metazoan model are cell-type specific. Tissue-specific roles of ER-phagy receptors may help to explain why the ubiquitous macro-autophagy machinery seems to be a universal requirement for longevity assurance in metazoan genetic studies, whereas the importance of selective ER-phagy mediators has been slower to emerge. Subsequently, we demonstrate that the two pathways capable of blocking age-associated ER-phagy, TMEM-131 and IRE-1-XBP-1, are required for mTOR-dependent lifespan extension in *C. elegans*.

Importantly, not all changes that occur during ageing reflect pathogenesis. The earliest remodelling events are likely to be adaptive responses to the cessation of developmental programmes and rising metabolic and cellular damage. We propose a model where age-dependent ER remodelling serves as an adaptive step in the ageing process associated with reprogramming of the proteostasis network[42,43]. This model is based on our findings that (1) the ER proteome shifts away from proteostasis machineries with age in conjunction with established, global declines in protein synthesis; (2) ER-phagy-dependent turnover of ER promotes yeast and *C. elegans* lifespan; and (3) long-lived animals adopt alternative ER network configurations throughout their adult life history. However, although data indicate that the net effect of ER-phagy on lifespan is positive, we speculate that early pronounced remodelling of ER structures is likely to trigger pleiotropic trade-offs later, especially in longer-lived cells and animals. For instance, several late-stage hallmarks of ageing are strongly influenced by ER contributions[71]. Examples of these hallmarks include declines in autolysosomal functions, where the

ER can serve as a key source of membrane for both autophagosome biogenesis and lysosome repair[72,73], and mitochondrial dysfunction, where ER tubules are an emergent scaffold for regulating mitochondrial fission–fusion processes and biogenesis[3]. Because the ER is a central hub of inter-organelle interactions across all compartments of the cell, its early remodelling is likely to reverberate across all organelle networks.

Our results align well with several recent findings and provide additional dimensions for future studies. First, having emerged as a candidate selective autophagy receptor among a pool of 144 other proteins[63], TMEM-131 exhibits several similarities with canonical ER-phagy receptors[31,32]. TMEM-131 is anchored in the ER, contains cytoplasmic intrinsically disordered regions for membrane fragmentation[33] and harbours multiple LIR motifs, enabling interaction with ATG8 family proteins. Consistent with expectations for a receptor, TMEM-131 knockdown suppresses ER-phagy-dependent remodelling and TMEM-131 puncta increase their association with Atg8 under ER-phagy-inducing starvation. However, several aspects limit assignment of TMEM-131 as a canonical receptor. First, its luminal domain is specialized for collagen processing[58], whereas other receptors typically play ER-shaping roles. Importantly, however, knockdown of TMEM-131's partners in collagen secretion (*trpp-8* and *dpy-31*) or COPII trafficking (*sec-24*, *sec-24* and *sec-31*)[58,74] fail to phenocopy its effects on ER remodelling, thus still providing support for a second role in ER-phagy. In addition, mutation of the TMEM-131 LIR motif reduces but does not abolish GABARAP binding in vitro, whereas most canonical receptors show more complete dependence. This result may suggest that TMEM-131 functions as a member of a recruitment complex or forms multivalent interactions through additional LIR motifs[67]. Finally, TMEM-131 localizes to punctate structures at baseline, probably corresponding with ER exit sites. This feature muddles the interpretability of its shift to autophagosomal structures following ER-phagy induction. Additional live-imaging experiments are required to place TMEM-131 spatially and temporally in the induction and targeting processes of ER-phagy, including monitoring how *tmem-131* modulates autophagosomal capture of other ER cargoes. For these reasons, we interpret TMEM-131 as an ER-phagy-promoting factor whose precise molecular mechanism remains to be defined. Future work, particularly with higher-resolution structural and imaging approaches, will be needed to clarify whether TMEM-131 is sufficient alone or in a complex for triggering ER membrane fragmentation and traffic to lysosomes. Surprisingly, our candidate screen for ER-phagy receptors did not reveal roles for established ER-phagy receptors bearing conserved RHDs. This result may stem from functional redundancy between RHD proteins or suggest that their roles are exerted in other tissues or contexts.

Another important outcome of our study is the integration of ER-phagy into the relationship between UPR signalling and ageing. Although the IRE-1–XBP-1 axis is traditionally associated with ER biogenesis, this role arises largely from acute stress contexts. During chronic activation, IRE-1–XBP-1 can also upregulate ER-phagy to counterbalance ER expansion[32,75]. A parallel study indeed revealed how the IRE-1–XBP-1 axis can act with BiP chaperones as a sensor system for activation of autophagy and subsequent ER turnover, and this pathway may be important in explaining our observations in the intestine[76]. Although IRE-1–XBP-1 can serve as an upstream activator of ER-phagy, previous studies muddle this relationship by reporting a loss of XBP-1 activity and inducibility in ageing *C. elegans* in the same time frames in which we observe ER-phagy activation[9,77]. Given that ER-phagy can ameliorate IRE-1–XBP-1 activation[35,78], the crosstalk between these pathways seems to involve a feedback loop, where age-induced ER-phagy may promote the altered UPR outputs observed during ageing. Similarly, our finding that the *ire-1–xbp-1* axis is required for mTOR-dependent longevity is consistent with its requirement in other longevity paradigms, including insulin signalling and dietary restriction[15,16]. Paradoxically, UPR activation seems to be lowered in these long-lived animals, despite the genetic requirement for longevity and enhanced stress resistance[15,16],

but the cell biological mechanism remains vague. Given our consistent evidence for broadscale ER reduction and remodelling across diverse life-extending interventions, our study opens avenues for understanding ER remodelling and ER-phagy as key aspects in explaining the requirement of *ire-1–xbp-1* in life extension.

Last, our data indicate an emerging link between ER remodelling and the balance of protein:lipid metabolic functions, which is also apparent in mammalian health. High-fat diets cause shifts from sheet-enriched to tubular ER networks in the liver and preventing these ER alterations by enforcing sheet formation ameliorates obesity-related metabolic dysregulation[24]. Furthermore, ageing is associated with ectopic lipid accumulation in diverse cells[79], which our results suggest may be supported by an underlying shift in ER-structure function. Our results illuminate a need to explore whether ER structure is linked to age-related metabolic dysfunction, which may open therapeutic avenues. The genetic connections between ER-shaping factors and diseases such as hereditary spastic paraplegia already provide further support for the sufficiency of altered ER morphology in driving disease in the nervous system and beyond[25,26,80]. Thus, we provide foundational evidence supporting ER-structure function as an important facet of geroscience, highlighting the potential for ER dynamics to serve as a targetable ageing process with the potential to modulate diverse forms of age-dependent pathophysiology.

## Online content

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

[1]Department of Cell and Developmental Biology, Vanderbilt University School of Medicine, Nashville, TN, USA. [2]Department of Molecular & Integrative Physiology, University of Michigan, Ann Arbor, MI, USA. [3]Department of Biochemistry, Vanderbilt University School of Medicine, Nashville, TN, USA. [4]Department of Biological Sciences, Vanderbilt University, Nashville, TN, USA. [5]National Center for Microscopy and Imaging Research, Department of Neurosciences, University of California San Diego, La Jolla, CA, USA. [6]Department of Molecular Physiology and Biophysics, Vanderbilt University School of Medicine, Nashville, TN, USA. ✉e-mail: kristopher.burkewitz@vanderbilt.edu

## Methods

This research complies with all ethical guidelines and regulations of the Vanderbilt University School of Medicine with approval from the Institutional Biosafety Committee and Institutional Animal Care and Use Committee.

### *C. elegans* husbandry

Worms were grown and maintained at 20 °C on 6-cm nematode growth media (NGM) plates seeded with *Escherichia coli* (OP50-1). Single *E. coli* colonies were cultured overnight in LB medium at 37 °C; each NGM plate was then seeded with 100 μl liquid culture was and cultured for two days at room temperature before use. For all experiments, worms were synchronized via timed egg lay. In experiments involving *glp-1* mutants, *glp-1* mutants and controls were moved to 25 °C 24 h after synchronization and incubated for 24 h to impair germline development. For experiments involving males, synchronized egg lays were conducted with mated worms and early larval stage L4 progeny were segregated by sex for subsequent experiments. For starvation experiments, young L4 animals were transferred to unseeded NGM plates containing carbenicillin (100 μg ml$^{-1}$) to prevent bacterial growth. A complete list of worm strains used in this study is in Supplementary Table 4.

### RNAi experiments

Gene-specific RNAi feeding clones (HT115) were obtained from the Ahringer RNAi Library (Source Bioscience), sequence-verified and cultured as described earlier for the OP50-1 strain, with the exception of the addition of 100 μg ml$^{-1}$ carbenicillin (final concentration) to the LB medium and NGM plates. At least an hour before use, control and experimental lawns were spotted with 100 μl IPTG solution (0.1 M isopropyl β-D-1-thiogalactopyranoside (IPTG), 100 μg ml$^{-1}$ carbenicillin and 12.5 μg ml$^{-1}$ tetracycline) to induce dsRNA expression. For *ifg-1* RNAi, feeding began at the L4 developmental stage. For the *lgg-1* and *unc-51* RNAi ageing experiments, the parental generation was also fed dsRNA from hatch for maximum penetrance. Otherwise, the worms were fed dsRNA from hatch.

### *C. elegans* fluorescence microscopy

Synchronized worms were raised as described earlier. For each condition, an agarose mounting pad—10% agarose in M9 buffer was prepared before imaging by briefly heating the agarose solution and flattening it between two slides (#1.5, VWR). Approximately 3 μl of 0.1 μm polybead microsphere suspension (Polysciences) was added to the pad, the worms were manually picked into the bead solution and a coverslip (#1.5, VWR) was added to immobilize the worms. For neuron-specific imaging, the worms were mounted instead on a 2% agarose pad in 6 μl of 10 mM levamisole (Fisher Scientific). Imaging was performed on an Eclipse Ti2 inverted microscope with a Yokogawa CSU-W1 spinning disc, a Plan Apo λ objective (×10/0.45 or ×100/1.45; Nikon) and a Prime 95B sCMOS camera (Teledyne Photometrics). Type B or LDF immersion oils (Cargille Labs) were used, always with consistent oil type across all replicates. Fluorophores were excited at 488 and 561 nm (dichroic mirror Di01-T405/488/568/647-13×15×0.5 (Semrock)), with respective ET525/36 m and ET605/52 m emission filters (Chroma). In addition, DIC images (89101m emission filter) were captured for each fluorescence image. For native TMEM-131::mKate2 imaging, 4× signal integration was used to compensate for low expression/intensity levels. Super-resolution images were obtained using an LSM980 microscope (Zeiss) with a ×63/1.40 Plan-Apochromat objective and Airyscan 2 detector. Acquisition settings, including exposure times, were identical across all conditions in each experiment. For consistency, focal planes were selected at nuclear midlines in the same cells across all conditions, specifically the anterior region of the hypodermis (*hyp7*), the first intestinal rings (*int1*), body-wall muscles and the ALN neuron. A plane was selected within the myofilament lattice to image the sarcoplasmic reticulum. For neural ER projections, imaging was acquired via *Z*-stack, with analysis performed on maximum intensity projections of the ALM, DA7 and PLM.

### *Saccharomyces cerevisiae* culture and lifespan assays

For imaging experiments, fluorescently labelled SEY6210 yeast cultures were cultured in synthetic complete dextrose medium at 30 °C in a gyratory shaker (220 r.p.m.) to mid-log phase (0 h). At each experimental time point, samples were removed for imaging and the culture was then maintained for subsequent time points. A chronological lifespan assay was performed as previously described[81]. Control and mutant yeast strains from the BY4741 gene-knockout library were cultured to mid-log phase in Longo's Aging synthetic complete dextrose medium buffered at pH 6.0 before being subcultured into fresh medium with a starting optical density at 600 nm of 0.1 and treated with either 15 nM rapamycin (in 95% ethanol) or an equivalent volume of 95% ethanol. The cultures were then grown to post-replicative phase after incubation for three days at 30 °C in a gyratory shaker (220 r.p.m.). For viability measurements, the cell samples were diluted in sterile water (10:3 dilution), plated on YPD plates and incubated at 30 °C. Following a three-day incubation, the colony-forming units were counted, with subsequent colony-forming-unit counts normalized as a percentage of the day 3 colony-forming-unit count. For each condition, the AUC value from three independent biological replicates was calculated as the lifespan statistic for comparison[69]. The AUC values were normalized within each genotype as the ratio between the rapamycin- and ethanol-treated groups. A complete list of yeast strains used in this study is provided in Supplementary Table 4.

### *S. cerevisiae* fluorescence microscopy

Before imaging, yeast were concentrated via centrifugation at 3,500*g*, pipetted onto a slide and immobilized with a cover slip. Images were acquired with a DeltaVision Elite Imaging system (Olympus IX-71 inverted microscope, Olympus ×100 oil objective (1.4 numerical aperture) and DV Elite sCMOS camera; GE Healthcare). Images were obtained in red (Alexa Fluor 594; 475 nm excitation, 523 nm emission), green (fluorescein isothiocyanate; 575 nm excitation, 632 nm emission), blue (4′,6-diamidino-2-phenylindole; 390 excitation, 435 emission) and DIC channels.

### Image processing and data analysis

Super-resolution Airyscan 2 images were deconvolved using the ZEN Blue v.3.5 software (Zeiss). NIS-Elements Advanced Research (NIS-AR, Nikon) was used to process images for all other worm experiments. Before analysis, images were subjected to automated two-dimensional or 3D deconvolution and subsequent denoising via the Denoise.ai module. Rolling-ball background removal was performed on each colour channel with identical settings for all images in the experimental replicate. For categorical analysis of ER morphology, images were blinded and the JavaScript command 'Math.random();' was used within the GA3 module of NIS-AR to select three random regions within the hypodermal region of interest. Subregion morphology was categorized using the Qualitative Annotations plugin in Fiji. For ER intensity and morphological quantitation, regions of interest were manually drawn around the cell of interest, specifically excluding nuclei and autofluorescent gut granules. GA3 was used for threshold-based segmentation of the ER. For experiments in which marked variability in ER fluorescence precluded the use of constant thresholds, thresholds were manually set on blinded images. In experiments involving ageing of lysosomal resistant mCherry markers, an upper threshold was also applied for ER segmentation to exclude ER contained in lysosomes. Images for yeast experiments were deconvolved using softWoRx and the integrated density of fluorescence intensity units were normalized to the cell surface area using Fiji. Yeast ER footprint and PAR were calculated from maximum intensity projections. To enable visualization of ER in aged samples where fluorescence intensity was substantially lower, minimal

modifications to brightness, contrast and γ were applied on representative images after analysis and always uniformly across all conditions.

## Transmission electron microscopy and analysis
Worms were collected from bacterial lawns and washed 3× with M9 buffer. The worm pellets were resuspended in 0.15 M sucrose prepared in M9 buffer and loaded into a 200-μm-deep well of an A-type carrier (Leica). The assembly was covered with the flat side of B-type carrier (Leica) and vitrified using a high-pressure freezing machine (Leica EM ICE). The frozen specimens were stored in liquid nitrogen until further processing with a modified freeze-substitution process. Briefly, freeze-substitution was performed using a cocktail of 0.5% glutaraldehyde and 0.1% tannic acid in acetone. The samples were transferred into an automatic freeze-substitution machine (AFS2, Leica) pre-cooled to −140 °C. They were then warmed to −90 °C over 3 h, followed by incubation at −90 °C for 114 h. Next, the samples were washed (30 min each) 4× with acetone. The final acetone wash was replaced with 2% osmium tetroxide in acetone. Following this, the samples were warmed to −20 °C over 12 h and maintained at −20 °C for an additional 10 h, after which they were warmed to 0 °C over 4 h. The samples were then transferred into an ice-filled bath and washed (30 min each) 4× with acetone and infiltrated with Spurr's resin (Electron Microscopy Sciences). During the final steps of resin infiltration, individual worms were transferred from the carriers into resin moulds with a fine needle and the samples were cured at 60 °C for 72 h. Thin sections (70 nm) were cut using a Leica UC7 ultramicrotome and a Diatome diamond knife, positioned on grids and stained with uranyl acetate and Reynold's lead citrate. These sections were then imaged on a Tecani T12 TEM operating at 100 keV using an AMT CMOS camera. The Amira software (Thermo Fisher Scientific) was used for image processing and analysis. Briefly, the hypodermal ER and cytosolic area were manually segmented with the freehand masking tool and the ER footprint and PAR were calculated.

## M. musculus husbandry
Male FVB/NJ mice obtained from the Jackson Laboratory were housed in cages of five mice and maintained on a 12-h light–dark cycle with controlled temperature (21 °C) and humidity. The mice had ad libitum access to chow food (Rodent Diet 20 5053, PicoLab) and water. All animal procedures were approved by the Institutional Animal Care and Use Committee of Vanderbilt University (M2000086-00/01).

## Scanning electron microscopy and analysis
Brain tissue for scanning electron microscopy imaging was prepared as previously described. Briefly, one male mouse from each age group (6 and 18 months) was killed and perfused with 37 °C Ringer's solution (0.79% NaCl, 0.038% KCl, 0.02% MgCl$_2$·6H$_2$O, 0.018% Na$_2$HPO$_4$, 0.125% NaHCO$_3$, 0.03% CaCl$_2$·2H$_2$O, 0.2% dextrose and 1,000 U heparin) for 30 s, followed by a transcardiac perfusion with fresh ice-cold 2.5% glutaraldehyde and 2% paraformaldehyde in 0.15 M sodium cacodylate at 10 ml min$^{-1}$ for 10 min. The brains were dissected and sliced along the sagittal plane with a vibratome to prepare 150-μm-thick sections and post-fixed overnight at 4 °C. The brain slices were washed in cold 0.15 M sodium cacodylate buffer and then fixed in 2% osmium tetroxide and 1.5% potassium ferrocyanide in 0.15 M sodium cacodylate buffer for 1 h at room temperature (approximately 23 °C). The samples were washed several times in double distilled water (ddH$_2$O), placed in a 0.5% thiocarbohydrazide solution for 30 min and then washed 5× in ddH$_2$O. Next, the slices were placed in a 2% aqueous osmium tetroxide solution for 1 h, thoroughly washed in ddH$_2$O, transferred to a 2% aqueous uranyl acetate solution and incubated at 4 °C overnight. The next day, the slices were washed with ddH$_2$O, placed into Walton's lead aspartate solution for 30 min and baked at 60 °C in a bench-top oven. The baked samples were thoroughly washed with ddH$_2$O, followed by a serial dehydration series on ice using ice-cold 70%, 90%, 100% and 100% ethanol, followed by dry acetone (10 min per step). The brain slices

were subsequently washed in 1:3, 1:1 and 3:1 Durcupan ACM:acetone solutions (12 h in each solution). Finally, the slices were incubated in three washes (24 h each) of 100% Durcupan ACM before being baked for 48 h at 65 °C for solidification.

Slices containing layer 2 motor cortical neurons were imaged on a Zeiss Gemini class scanning electron microscope. Images were blinded and one neural soma from the central region of each image was selected for analysis. The ER and somatic cytosolic area were manually segmented in NIS-AR and the ER footprint was calculated.

## C. elegans cloning and transgenesis
To generate extrachromosomal array wbmEx222[eft-3p::GFP::sec-61.b::unc-54 UTR], a C. elegans codon-optimized GFP(S65C,Q80R) sequence was fused to the native sec-61.b sequence in an expression vector to generate plasmid pKB22. The plasmid was column-purified and micro-injected as previously described.

To generate ret-1::GFP and ret-1::mKate2, codon-optimized GFP(S65C,Q80R) and mKate2 sequences were amplified with an N-terminal GGGGS linker from the plasmids pKB51(GFP) and pKB52(mKate2) using a common forward primer (Fwd, 5′-GTGCTCCAGTCGCCGCTGAAGAGAAGAAGGATCAAGGTG-GCGGAGGTTCTGG-3′) and specific reverse primers (Rev-GFP, 5′-CAATGGCAAAGTGTGTTCTTCTTTCAATCGATTTATTTG-TATAGTTCATCCATG-3′ or Rev-mKate2, 5′-CAATGGCAAAGTGT-GTTCTTCTTTCAATCGATTTAACGGTGTCCGAGCTTGGATG-3′; IDT). The repair templates were column-purified and injected as previously described with a guide (Dharmacon) against the following target sequence: 5′-AATGGCAAAGTGTGTTCGGA-3′. For ret-1::wrmScarlet11, a single-stranded-oligodeoxynucleotide repair template including the wrmScarlet11 sequence (5′-GTGCTCCAGTCGCCGCTGAAGAGAA-GAAGGATCAAGGTGGCGGAGGTTCTTACACCGTCGTCGAGCAATAC-GAGAAGTCCGTCGCCCGTCACTGCACCGGAGGAATGGATGAGTTATA-CAAGTAAATCGATTGAAAGAAGAACACACTTTGCCATTGTTTTTC-3′) was synthesized (IDT). The repair template was injected with the same guide used for the GFP and mKate2 fusions. To generate the tmem-131::mKate2 allele, 2×GGGGS::mKate2 was amplified with from pKB52 via subsequent rounds of PCR using the primers: F1, 5′-AATGCCACAGGACACCGACAATGAAAACGACGAGA-AGAACAATGGTGGCGGAGGTTCTGG-3′; R1, 5′-ATATAATGGAGAAT-GTGTTGTAGAATGGAATTATTAACGGTGTCCGAGCTTGGATG-3′; F2, 5′-CAACACAACAACCATCAACTTCTCAAATGCCACAGGACACCGAC-3′ and R2, 5′-CATATAGATTTGGATAGCAGATAAGTAGATATATAATG-GAGAATGTGTTGTAG-3′ (IDT). The purified repair template was injected with guides targeting the following target sequences: 5′-TTCTCAAATGCCACAAGATA-3′ and 5′-CAGATAAGTAGATATAATAT-3′ (Dharmacon). All fusion strains were sequence-verified and outcrossed with N2 worms at least four times.

For neuron-specific RET-1 fluorescence, we first generated extrachromosomal array bugEx10[rab-3p::wrmSct1-10::unc-54 UTR + rab-3p::GFP::unc-54 UTR] through the cloning and subsequent co-injection of pBK01 (62 ng μl$^{-1}$) and pBK02 (53 ng μl$^{-1}$) into N2 worms. To generate plasmid pBK01(rab-3p::GFP::unc-54 UTR), the atf-6 sequence was excised from plasmid pKB41 via BamHI and AgeI digest, and the backbone was re-ligated after Klenow treatment. To generate plasmid pBK02(rab-3p::wrmSct1-10::unc-54 UTR), the wrmSct1-10::unc-54 UTR sequence was amplified from plasmid pJG100 with the primers Fwd, 5′-CCCGGGATGGTATCGAAGGGAGAAGC-3′ and Rev, 5′-ACTAGTCTTCCACTGAGCCTCAAA-3′ (IDT). The PCR product was ligated into a vector containing the rab-3p sequence following XmaI and SpeI digestion. After micro-injection, GFP$^+$ worms were crossed with ret-1::wrmScarlet11 worms to generate the strain BUZ112(ret-1::wrmScarlet11; bugEx10).

The TMEM-131 transmembrane segment was identified using Deep TMHMM[82], after which the coding sequence for the cytosolic C terminus (residues 1191–1831, 'TMEM-131$_C$') was amplified from

N2 worm complementary DNA using the primers F, 5′-TATCTGG AAGGTGATCGTGCTATTGC-3′ and R, 5′-TTAGTTATTCTTTTCATC ATTCTCGTTGTCCGTATC-3′ (IDT). Homology to plasmid pMal-c5e was added via PCR with the primers F, 5′-TACCGCATATGTCCATGGGC TATCTGGAAGGTGATCGTG-3′ and R, 5′-CTGAGCCTTTCGTTT-TATTTGATTAGTTATTCTTTTCATCATTCTCGTTGTCC-3′ (IDT). The PCR product was column-purified (Invitrogen) and inserted into linearized custom 6×HIS::MBP pMal-c5e vector (NotI-HF/HindIII-HF) via Gibson assembly (NEB) to generate plasmid pED09. To generate a LIR mutant TMEM peptide recoding FVHV as SSSS (TMEM ΔLIR), plasmid pED09 was amplified with the primers F, 5′-GAGCTCCTCCCCACCAACTTCTTTAAC-3′ and R, 5′-GAGTCAT CAGTTGGAGCTGACGTC-3′ (IDT). The PCR product was isolated via gel purification and treated with KLD enzyme mix (NEB) to degrade any remaining template and circularize the product as plasmid pED10. Both plasmids were sequence-verified and individually transformed into Rosetta2 (DE3) competent cells for protein expression.

### S. cerevisiae cloning and transgenesis

PCR was used to amplify mCherry (pFA6a-mCherry-KanMX6), mNeonGreen (pFA6a-mNeonGreen-NATMX) and *mTagBFP2* (*pFA6a-mTagBFP2-TRP1*) to create DNA tagging cassettes as previously described[83]. The cassettes were gel-purified and transformed into SEY6210 yeast cells as previously described[84]. C-terminal integration of the fluorescent tag into the endogenous coding sequence was confirmed by PCR and fluorescence microscopy.

### Protein isolation and SDS−PAGE

Synchronized populations of at least 400 worms were raised as described earlier on 10-cm NGM plates seeded with 500 μl overnight OP50-1 cultures. To assist with ageing a large population, L4 worms expressing 3×FLAG::SEC-61.B were transferred to lawns spotted with 250 μl 5-fluoro-2′-deoxyuridine (1 mg ml$^{-1}$ solution) 24 h before use. For RET-1::GFP blotting, germline tumours were observed with extremely high levels of RET-1, necessitating a germlineless *glp-1* background. *Ret-1::gfp; glp-1* worms were heat shocked as described earlier. The worms were collected from bacterial lawns and washed 3× with M9; the worm pellets were then flash-frozen with liquid nitrogen and stored at −80 °C. They were then lysed at 4 °C by sonication in RIPA buffer (50 mM Tris, 150 mM NaCl, 1 mM EDTA, 1% Triton X-100, 0.1% sodium deoxycholate and 0.1% SDS, pH 7.5) containing protease inhibitors (Roche Applied Science). The lysates were centrifuged at 14,000*g* and 4 °C for 15 min and the supernatants were collected. Protein concentrations were determined using a Pierce BCA assay (Thermo Scientific). An equal mass of each sample was diluted in Laemmli Sample Buffer (BioRad) with 2-mercaptoethanol, heated (95 °C for 15 min) and resolved via SDS−PAGE electrophoresis.

### Western blotting, protein quantification and data analysis

Proteins were transferred to activated polyvinylidene fluoride membranes over 2 h at 4 °C. Ponceau (Sigma Aldrich) staining, imaging and washing was conducted as per the manufacturer's recommendations. Target proteins were probed through the following steps: an overnight 4 °C incubation in blocking solution (5% non-fat milk in TBST solution), incubation in primary antibody solution (5% milk in TBST + antibody − anti-FLAG (1:1,500) or anti-GFP (1:1,000; Santa Cruz)) for 2 h at room temperature, three TBST washes (10 min each) at room temperature and incubation with secondary antibody solution (5% milk in TBST + antibody (horseradish peroxidase (HRP)-conjugated, anti-mouse; 1:10,000)) for 1 h at room temperature. After another three TBST washes (10 min) at room temperature, the membranes were incubated with HRP substrate (Pierce ECL western blotting substrate, Thermo Scientific) as per the manufacturer's recommendations. All wash and incubation steps were performed with gentle nutation. Ponceau and ECL imaging were performed using an Amersham Imager 600 (GE Life Sciences).

GFP::LGG-1 blots included the following modifications: room temperature transfer using a Trans-Blot Turbo system (BioRad), blocking for 1 h at room temperature, overnight incubation with anti-GFP (Roche) primary antibody at 4 °C, HRP-conjugated secondary antibody incubation for 1 h, developing with ProSignal Dura ECL (Genesee) and imaging with Invitrogen iBright500. Bands were manually selected and quantified using Fiji and HRP signal was normalized to α-tubulin for analysis.

### Protein purification

We expressed 6×HIS::GST::GABARAP in a pGEX-6P-1 in BL21(DE3)pLysS cells (Invitrogen) for 16–20 h at 22 °C after induction with 0.4 mM IPTG. Both 6×HIS::MBP::TMEM-131$_c$ and 6×HIS::MBP::TMEM-131(LIR Mutant)$_c$ were expressed from Rossetta2 (DE3) cells in LB broth containing 100 μg ml$^{-1}$ ampicillin cultured to an optical density at 600 nm of approximately 0.5 and induced with 1 mM IPTG at 18 °C for about 16 h. Purification of TMEM-131 and GABARAP protein variants were performed identically. Briefly, the cells were resuspended in buffer (25 mM HEPES, 500 mM NaCl, 20% (wt/vol) glycerol, 1 mM dithiothreitol (DTT) and 0.4 M L-arginine, pH 7.4) and lysed by homogenization, spun at 18,000 RCF for 35 min. The lysates were filtered and passed over a HISTrap column (Cytiva). The resin was washed with buffer 1 (25 mM HEPES, 500 mM NaCl, 20% (wt/vol) glycerol and 1 mM DTT, pH 7.4). Protein was eluted in with buffer 1. The pooled fractions were loaded onto a HiPrep 26/60 Sepharcryl S300 column (Cytiva) equilibrated with buffer 1. After size exclusion, the final samples were collected and stored at −80 °C.

### MBP pull-down

The entire pull-down protocol was performed on ice or at 4 °C. Amylose resin (New England Biolabs) was washed with buffer A (25 μM HEPES, 500 mM NaCl and 20% (wt/vol) glycerol, pH 7.5) and then incubated with 6×HIS::MBP::TMEM-131, 6×HIS::MBP::TMEM-131(LIR mutant) or MBP for 30 min. The slurry was washed thoroughly with buffer B (25 mM HEPES, 125 mM NaCl and 5% (wt/vol) glycerol, pH 7.5). Next, 6×HIS::GST::GABARAP was added to the solution at a final in-reaction concentration of 28 μM and incubated for 30 min at a final salt concentration of 159 mM NaCl. Each slurry was washed three times with 10× reaction volume of buffer B, completely removing the remaining supernatant after the last wash. Between washes, the beads were pelleted by centrifugation at 5,000*g*. The beads were incubated with 10 μl elution buffer (25 mM HEPES, 175 mM NaCl and 100 mM maltose) for 15 min. Small holes were poked in the bottom of each tube with 21 G needles. The supernatant was separated by centrifugation into a second 1.5-ml tube. Next, SDS sample buffer was added to the samples, which were then boiled. The positive control was loaded at a 2% positive loading control. The samples were run on a 10% bis-Tris gel (Thermo Scientific) at 150 V for 120 min. The gel was stained with SimplyBlue SafeStain Coomassie dye (Thermo Scientific) and imaged using a Bio-Rad Universal Hood II. For quantitation, band-intensity-measurement images (tagged image file format, tiff) of scanned Coomassie gels were made with FIJI and GABARAP-TMEM pull-down bands were normalized to the input.

### Quantification, statistics and reproducibility

Data from published *C. elegans* proteomic datasets[43] were mined by identifying proteins annotated for ER localization (Wormbase WS292, GO_0005783) and WormCat2.0 annotated functions in either proteostasis (categories: chaperone, protein modification and stress response) or lipid metabolism (Supplementary Table 1), and relative abundances for each set of proteins were contrasted between day 6 and day 1. For mining of the Tabula Muris Senis RNA expression atlas, expression *Z* scores of genes annotated as 'ER protein processing' (KEGG mmu04141) from Schaum et al.[46] were extracted from public processed data repository (twc-stanford.shinyapps.io/maca), averaged across all tissues as in the

original manuscript and replotted. For mammalian proteome mining, processed expression data from soluble whole-tissue-lysate samples[48] were extracted from public repository (https://aging-proteomics.info), using $\log_2$-transformed fold changes relative to 6-month-old mice. Pantherdb (version 19.0) was used to extract genes annotated for mitochondria, ER and protein localization to ER gene ontology terms, and the Panther enrichment test[85] (Fisher's Exact with FDR) was used to determine statistical enrichment of specified gene sets relative to a reference set of all genes. A cutoff of FDR < 0.05 was used for determining up- or downregulated proteins among all ER and mitochondrial proteomes. To identify enrichment of cellular processes within subsets of differentially regulated ER proteins, the Panther overrepresentation test was used and the top ten biological processes by FDR are listed. Imaging experiments were conducted in three independent repeats and data were pooled before analysis. Sample sizes were based on previous experiments with similar designs. For imaging experiments, randomization occurred via the picking of random worms to a slide and the experimenter was blinded to experimental conditions before either image acquisition or analysis, as feasible. Blinding was performed with the Fiji script blind-files, provided as part of the Lab-utility-plugins update site. Isolated comparisons between two independent groups were conducted via a two-tailed Student's $t$-test or Welch's $t$-test when noted otherwise. Western blot data were analysed via two-tailed paired ratio $t$-tests. Categorical morphologic analyses were conducted using a $\chi^2$ test. For experiments requiring multiple comparisons, one- or two-way ANOVAs were conducted as appropriate with post-hoc two-tailed tests to determine intergroup differences. Data distribution was assumed to be normal but not formally tested. Prism (v.9 and v.10, GraphPad) was used for statistical analysis and plot generation. Graph midlines indicate the mean ± 95% CI unless otherwise specified. Statistical tests, $P$ values and sample sizes are indicated in the figure legends or on the corresponding graphs.

### Reporting summary

Further information on research design is available in the Nature Portfolio Reporting Summary linked to this article.

## Data availability

Data supporting this work are available through the supplementary information. Worm and yeast strains generated in this study will be available on request and/or deposited to the *Caenorhabditis* Genetics Center (CGC) repository. Source data are provided with this paper. All other data supporting the findings of this study are available from the corresponding author on reasonable request.

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

## Acknowledgements

We thank all of the Burkewitz laboratory members for constructive feedback, and A. Orgel, E. Diao, H. L. Singkhek, A. J. Kyle and A. Pefanco for technical assistance. We thank A. Arruda and G. Parlakgul for technical advice, C. Wright for constructive feedback, and V. Gama, M. Patel, W. Mair and D. Miller for sharing reagents and equipment. In addition to BQ5 from P. Hu and XW8056 provided by X. Wang, The *Caenorhabditis* Genetics Center (NIH Office of Research Infrastructure Programs P40 OD010440) provided the worm strains used in this work. TEM sample preparation was performed through the Washington University Center for Cellular Imaging (WUCCI), supported by the Washington University School of Medicine, The Children's Discovery Institute of Washington University and St. Louis Children's Hospital (CDI-CORE-2015-505 and CDI-CORE-2019-813), and the Foundation for Barnes-Jewish Hospital (3770 and 4642). Airyscan and TEM imaging were performed through the Vanderbilt Cell Imaging Shared Resource (supported by NIH grants CA68485, DK20593, DK58404, DK59637 and EY08126). This work was supported by NIA F31AG076290 and NIGMS T32GM007347 (E.K.F.D.), NIGMS T32GM152286 (S.S.), NIGMS R35GM155303 (A.F.), NIGMS R35GM144112 (J.M.), NIA R00AG052666 and R01AG073354 (K.B.), and the Glenn Foundation for Medical Research/American Federation for Aging Research (K.B.).

## Author contributions

Conceptualization: E.K.F.D. and K.B. Methodology: E.K.F.D., N.L.H., A.W.F., J.A.M. and K.B. Investigation: E.K.F.D., N.L.H., E.M.R., A.G.M., B.K., N.D.U., L.P., S.S., D.J.J., R.A.D., A.W.F. and K.B. Writing–original draft: E.K.F.D. and K.B. Writing–review and editing: all authors. Funding acquisition: E.K.F.D. and K.B. Resources: N.S.W., L.P.J., M.H.E. and R.A.D. Supervision: A.W.F., M.C.T., J.A.M. and K.B.

## Competing interests

The authors declare no competing interests.

## Additional information

**Extended data** is available for this paper at https://doi.org/10.1038/s41556-025-01860-1.

**Correspondence and requests for materials** should be addressed to Kristopher Burkewitz.

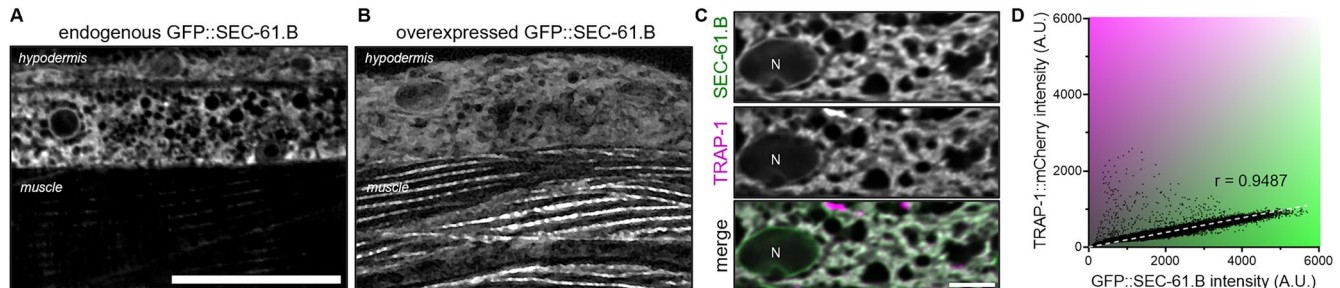

**Extended Data Fig. 1 | Validation of ER subdomain reporters. a**, Representative images of endogenously tagged GFP::SEC-61.B, which is enriched in the hypodermis but excluded from the neighbouring muscle sarcoplasmic reticulum (SR). Expression pattern observed consistently in 100 animals across five independent experiments. Scale: 10 μm. **b**, Overexpressed GFP::SEC-61.B causes mislocalization, with the fusion protein now also highly expressed in the muscle SR. Image representative of 10 animals across three independent experiments. **c**, Representative image of hypodermal ER endogenously co-labelled with translocon subunits GFP::SEC-61.B and TRAP-1::mCherry. Image representative of 10 animals across three independent experiments. Scale: 10 μm. **d**, Pearson's correlation (*r*) between GFP::SEC-61.B and TRAP-1::mCherry. *r* = 0.9487. Source numerical data are provided.

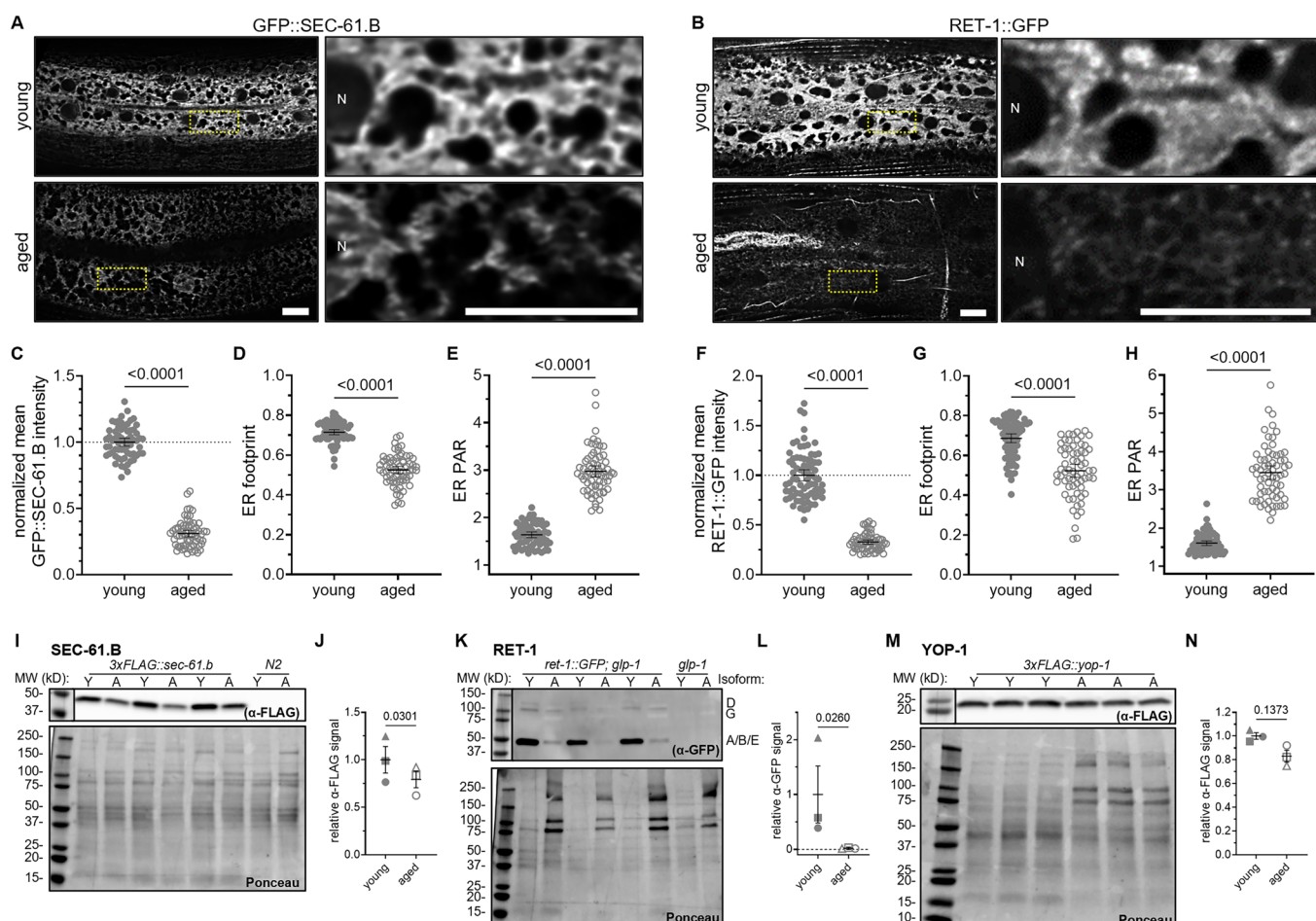

**Extended Data Fig. 2 | Altered expression of ER subdomain-enriched proteins in ageing. a**, Hypodermal GFP::SEC-61.B in young and aged worms. All scales: 10 μm. **b**, Hypodermal RET-1::GFP in young and aged worms. All scales: 10 μm. **c–e**, Quantification of normalized mean intensity (**c**), footprint (**d**), and perimeter:area ratio (**e**) in young vs. aged GFP::SEC-61.B worms. N = 64 young, 67 aged, pooled across three replicates. Analysed via two-tailed Student's *t*-tests. **f–h**, Quantification of normalized mean intensity (**f**), footprint (**g**), and PAR (**h**) in young vs. aged RET-1::GFP worms. N = 76 young, 65 aged, pooled across three replicates. Analysed via two-tailed Student's *t*-tests. **i**, Western blotting of SEC-61.B (α-FLAG) on lysates from *3xFLAG::sec-61.b* worms, with N2 controls. N = 3 young d1 samples (Y) and aged d7 samples (A). Ponceau staining included for total protein normalization. **j**, Quantification of *sec-61.b* α-FLAG signal in young and aged worms, relative to total protein levels (Ponceau) and normalized to d1. Replicates indicated by similar shapes. Analysed via two-tailed ratio paired *t*-test. **k**, Western blotting of RET-1 (α-GFP) on lysates from *ret-1::GFP; glp-1* worms, with

*glp-1* controls. Lower molecular weight bands are consistent with predictions for RET-1.A/B/E; higher molecular weight bands are consistent with predictions for RET-1.D/G. N = 3 young d1 samples (Y), 3 aged d7 samples (A). Ponceau staining included for total protein normalization. **l**, Quantification of α-GFP signal in young and aged worms, relative to total protein levels (Ponceau) and normalized to d1. Replicates indicated by similar shapes. Analysed via two-tailed ratio paired *t*-test. **m**, Western blotting of YOP-1 (α-FLAG) on lysates from *3xFLAG::sec-61.b* worms, with N2 controls. N = 3 young d1 samples (Y) and aged d7 samples (A). Ponceau staining included for total protein normalization. **n**, Quantification of *yop-1* α-FLAG signal in young and aged worms, relative to total protein levels (Ponceau) and normalized to d1. Replicates indicated by similar shapes. Analysed via two-tailed ratio paired *t*-test. For graphs C-H, error bars indicate mean ± 95% CI; for graphs J, L, N: mean ± SEM. For images, "N" = nucleus. Source numerical data and unprocessed blots are provided.

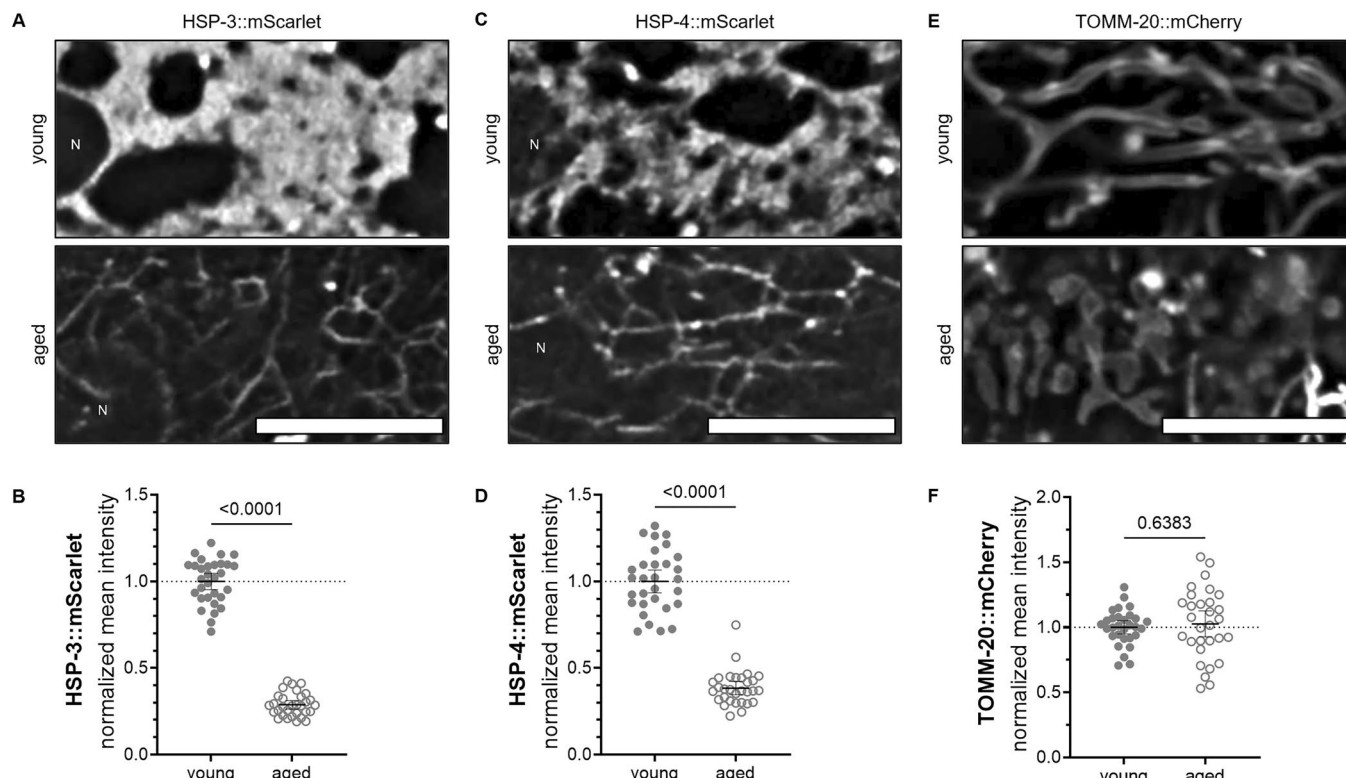

**Extended Data Fig. 3 | Fluorescence imaging of additional organelle markers during ageing. a**, Hypodermal HSP-3::mScarlet in young and aged worms. Scale: 10 μm. **b**, Quantification of normalized mean intensity in young vs. aged *hsp-3::mScarlet* worms. N = 31 worms/age group, pooled across two replicates. Analysed via two-tailed Student's *t*-test. **c**, Hypodermal HSP-4::mScarlet in young and aged worms. Scale: 10 μm. **d**, Quantification of normalized mean intensity in young vs. aged *hsp-4::mScarlet* worms. N = 30 worms/age group, pooled across two replicates. Analysed via two-tailed Student's *t*-test. **e**, Hypodermal TOMM-20::mCherry in young and aged worms. Scale: 10 μm. **f**, Quantification of normalized mean intensity in young vs. aged *tomm-20::mCherry* worms. N = 30 worms/age group, pooled across two replicates. Analysed via two-tailed Student's *t*-test. For all graphs, error bars indicate mean ± 95% CI. For images, "N" = nucleus. Source numerical data are provided.

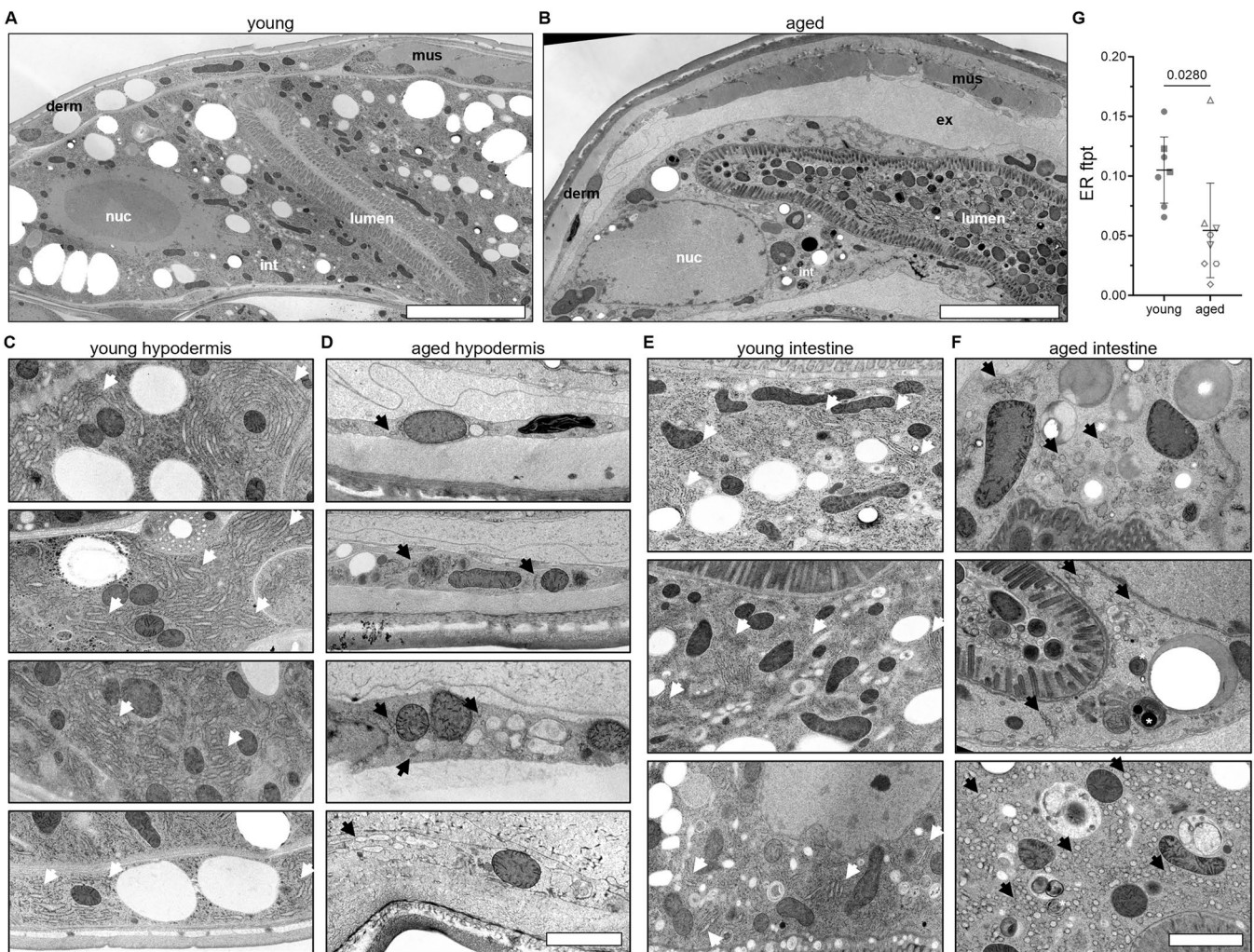

**Extended Data Fig. 4 | TEM of ER in young and aged *C. elegans*. a,b,** Tiled TEM images of cross-sections from young (**a**) and aged (**b**) adult *C. elegans* showing intestine, hypodermal and muscle cells. Undigested bacterial cells are visible in the intestinal lumen of the aged worm. Scale bar, 5 μm. Intestine: int, Intestinal nucleus: nuc, hypodermis: derm, extracellular pseudocoelom: ex, muscle: mus. **c,d,** Representative fields of view of the ER in hypodermal cells in different young (**c**) and aged (**d**) *C. elegans* individuals. White arrows = cisternal ER stacks. Black arrows = ER tubules. Scale bar, 1.5 μm. **e,f,** Representative fields of view of the ER in intestinal cells in different young (**e**) and aged (**f**) *C. elegans* individuals. White arrows = cisternal ER stacks. Black arrows = ER tubules. Asterisk = autolysosomes containing ER whorls. Scale bar, 1.5 μm. **g,** ER footprint measured from TEM of the intestine. $N_{young}$ = 7 sections from two worms; $N_{aged}$ = 8 sections from four4 worms; worms indicated by symbols. Analysed via two-tailed Welch's *t*-test. Error bars indicate mean ± 95% CI. Source numerical data are provided.

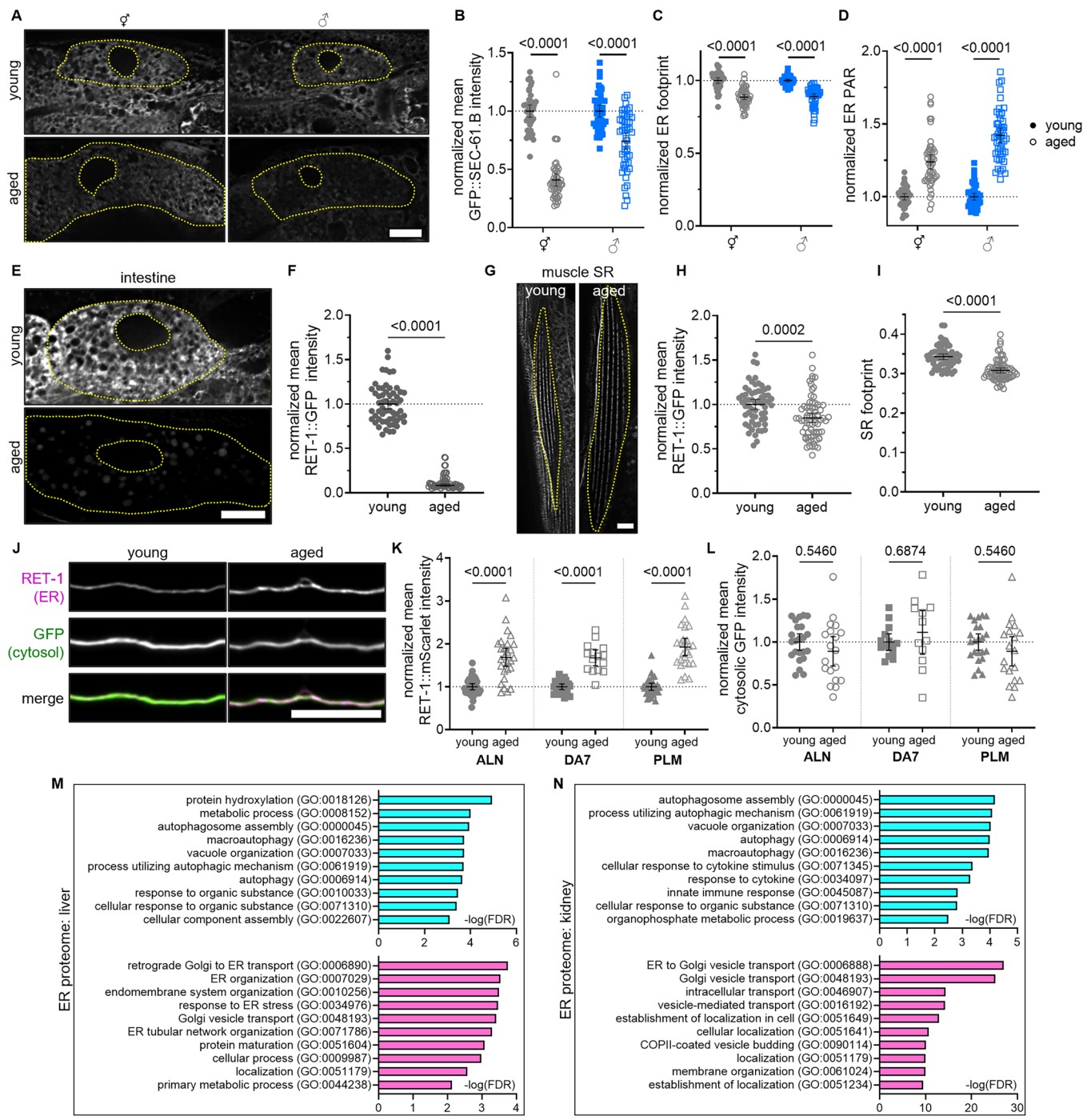

**Extended Data Fig. 5 | Multiple tissues and ER structures undergo age-onset ER remodelling. a,** Imaging of intestinal GFP::SEC-61.B during ageing of hermaphrodite (♀) and male (♂) worms. Scale: 10 μm. **b–d,** Quantification of mean intensity (J), footprint (K), and PAR (L) in intestine of GFP::SEC-61.B worms across age, normalized within each sex to d1. $N_{♂}$ = 42 young, 50 aged; $N_{♂}$ = 43 young, 49 aged; pooled across three replicates. Analysed via two-tailed Student's *t*-tests. **e,** Imaging of intestinal RET-1::GFP in young and aged worms. Scale: 10 μm. **f,** Quantification of intestinal RET-1::GFP intensity, normalized to d1. N = 58 young, 59 aged, pooled across three replicates. Analysed via two-tailed Student's *t*-test. **g,** Imaging of RET-1::GFP of muscle myofilament-associated SR in young and aged animals. Scale: 10 μm. **h,i,** Quantification of normalized mean intensity (**h**), and footprint (**i**) of RET-1::GFP in muscle SR. N = 61 young, 72 aged, pooled across three replicates. Analysed via two-tailed Student's *t*-tests. **j,** Maximum

intensity projections of neuron-specific RET-1::wrmScarlet in the neurites of the ALN neuron marked with cytoplasmic GFP. Scale = 10 μm. **k,l,** Quantification of mean RET-1::wrmScarlet (**k**) and cytoplasmic GFP (**l**) intensity along the ALN, DA7, and PLM neurites in young and aged worms. For (**g**) $N_{ALN}$ = 34 young, 27 aged; $N_{DA7}$ = 23 young, 15 aged; $N_{PLM}$ = 28 young, 26 aged; pooled across three replicates. For (**h**), $N_{ALN}$ = 23 young, 19 aged; $N_{DA7}$ = 15 young, 12 aged; $N_{PLM}$ = 23 young, 19 aged; pooled across two replicates. All data analysed via one-way ANOVAs, p < 0.0001 (**k**) and p = 0.2906 (**l**), post hoc Šidák tests. **m,n,** Top 10 enriched gene categories among up- and downregulated ER proteins in liver (**m**) or kidney (**n**). **a,e,g** Dotted yellow lines demarcate the nuclear envelope (**a,e**) and cell membrane (**a,e,g**). For all graphs, error bars indicate mean ± 95% CI. Source numerical data are provided.

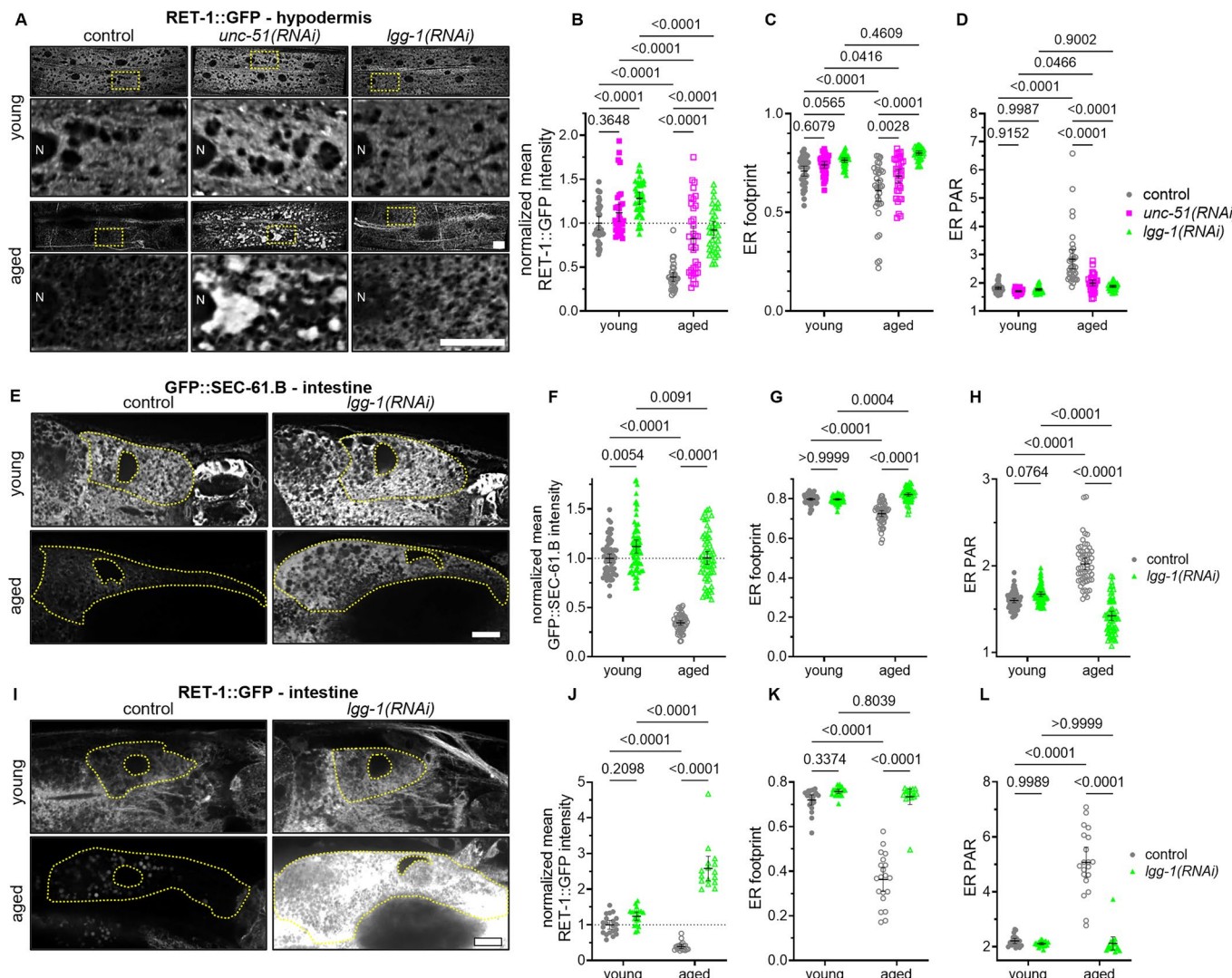

**Extended Data Fig. 6 | Autophagy regulates the turnover of multiple ER subdomain proteins. a**, Hypodermal RET-1::GFP in in young and aged control (e.v.), *unc-51(RNAi)*, and *lgg-1(RNAi)* worms. All scales: 10 μm. "N" = nucleus. **b**–**d**, Hypodermal RET-1::GFP normalized mean intensity (**b**), footprint (**c**), and PAR (**d**) in young and aged control, *unc-51(RNAi)*, and *lgg-1(RNAi)* worms. Young = filled shapes; aged = hollow shapes. $N_{e.v.}$ = 35 young, 35 aged; $N_{unc-51}$ = 35 young, 35 aged; $N_{lgg-1}$ = 35 young, 31 aged; pooled across two replicates. Analysed via two-way ANOVAs, post hoc Šidák tests. **e**, Intestinal GFP::SEC-61.B in young and aged control and *lgg-1(RNAi)* worms. All scales: 10 μm. **f**–**h**, Intestinal GFP::SEC-61.B normalized mean intensity (**f**), footprint (**g**), and PAR (**h**) in young and aged control and *lgg-1* KD worms. Young = filled shapes; aged = hollow shapes. $N_{e.v.}$ = 63 young, 55 aged; $N_{lgg-1}$ = 64 young, 58 aged, pooled across three replicates. Analysed via two-way ANOVAs, post hoc Šidák tests. **i**, Intestinal RET-1::GFP in young and aged control and *lgg-1(RNAi)* worms. All scales: 10 μm. **j**–**l**, Intestinal RET-1::GFP normalized mean intensity (**j**), footprint (**k**), and PAR (**l**) in young and aged control and *lgg-1(RNAi)* worms. Young = filled shapes; aged = hollow shapes. $N_{e.v.}$ = 20 young, 20 aged; $N_{lgg-1}$ = 20 young, 16 aged; from one replicate. Analysed via two-way ANOVAs, post hoc Šidák tests. **e,i** Dotted yellow lines demarcate the cell membrane and nuclear envelope. For all graphs, error bars indicate mean ± 95% CI. Source numerical data are provided.

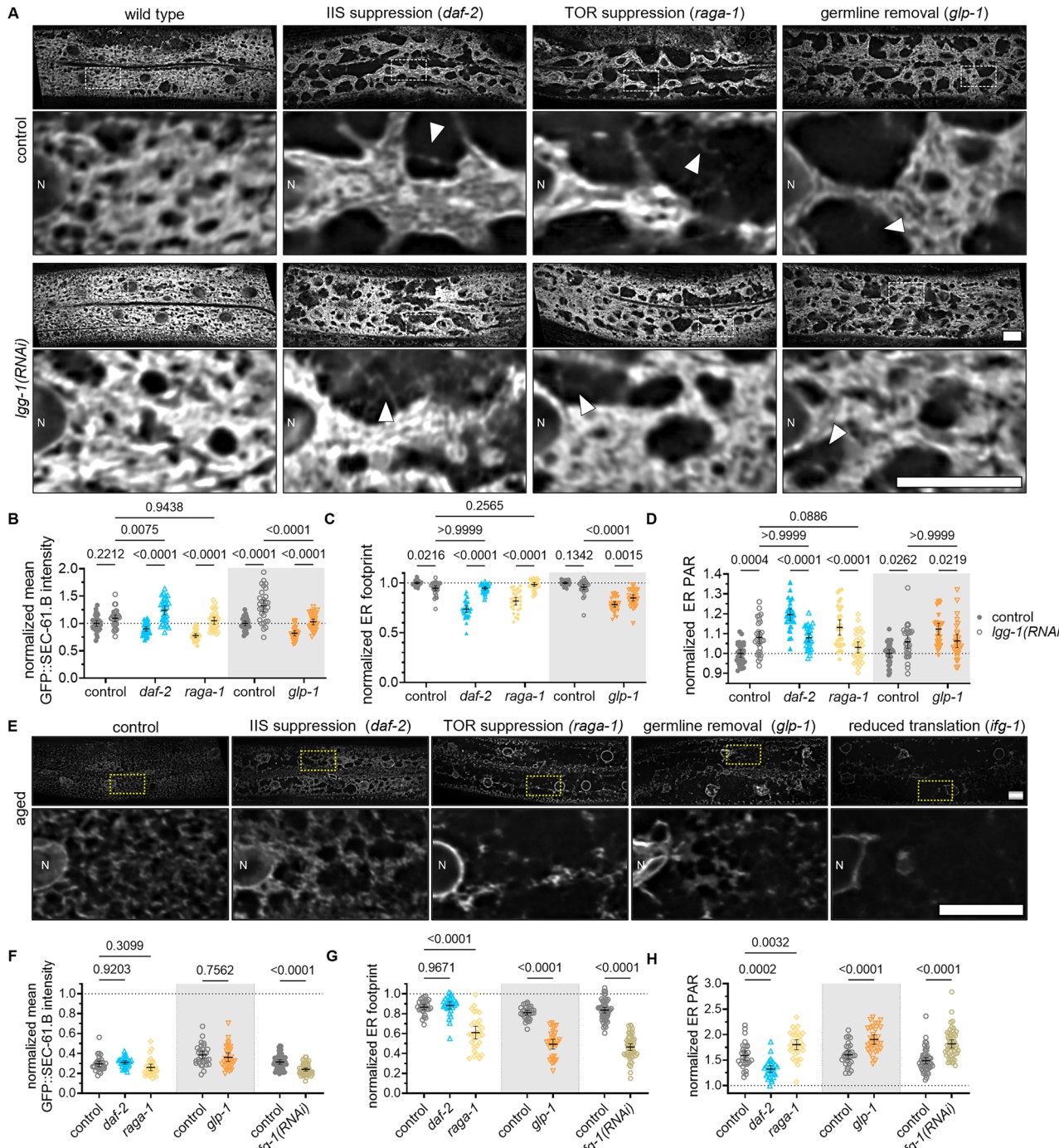

**Extended Data Fig. 7 | Longevity-associated ER remodelling requires autophagy. a**, Imaging of hypodermal GFP::SEC-61.B in young adult worms from wild-type, *daf-2*, *raga-1*, and *glp-1* backgrounds fed control or *lgg-1* dsRNA. All scales: 10 μm. Arrows indicate sparse ER tubules. **b**–**d**, Quantification of mean intensity (**b**), footprint (**c**), and PAR (**d**) of GFP::SEC-61.B in long-lived animals relative to wild-type controls after feeding of empty vector or *lgg-1* dsRNA. Coloured backgrounds group experimental conditions with their respective controls. N = 30 control, 30 *daf-2*, 30 *raga-1*, 30 control(25 °C), 30 *glp-1*, pooled across three replicates. Analysed via two-way ANOVAs, post hoc Šidák tests. **e**, Imaging of hypodermal GFP::SEC-61.B in aged d7 adult worms from wild-type,

*daf-2*, *raga-1*, and *glp-1* backgrounds and worms fed *ifg-1* dsRNA. All scales: 10 μm. **f**–**h**, Quantification of mean intensity (**f**), footprint (**g**), and PAR (**h**) of GFP::SEC-61.B in aged animals relative to d1 controls (normalized to 1.0). Coloured backgrounds group experimental conditions with their respective controls. N = 30 control, 30 *daf-2*, 30 *raga-1*, 30 control(25 °C), 30 *glp-1*, 51 control (empty vector RNAi), 52 *ifg-1*; pooled across three replicates. All data analysed via one-way ANOVAs, p < 0.0001, post hoc Šidák tests. Data in panels **b-d**, **f–h**, and Fig. 6f–h were generated as part of one large experiment. For all graphs, error bars indicate mean ± 95% CI. For images, "N" = nucleus. Source numerical data are provided.

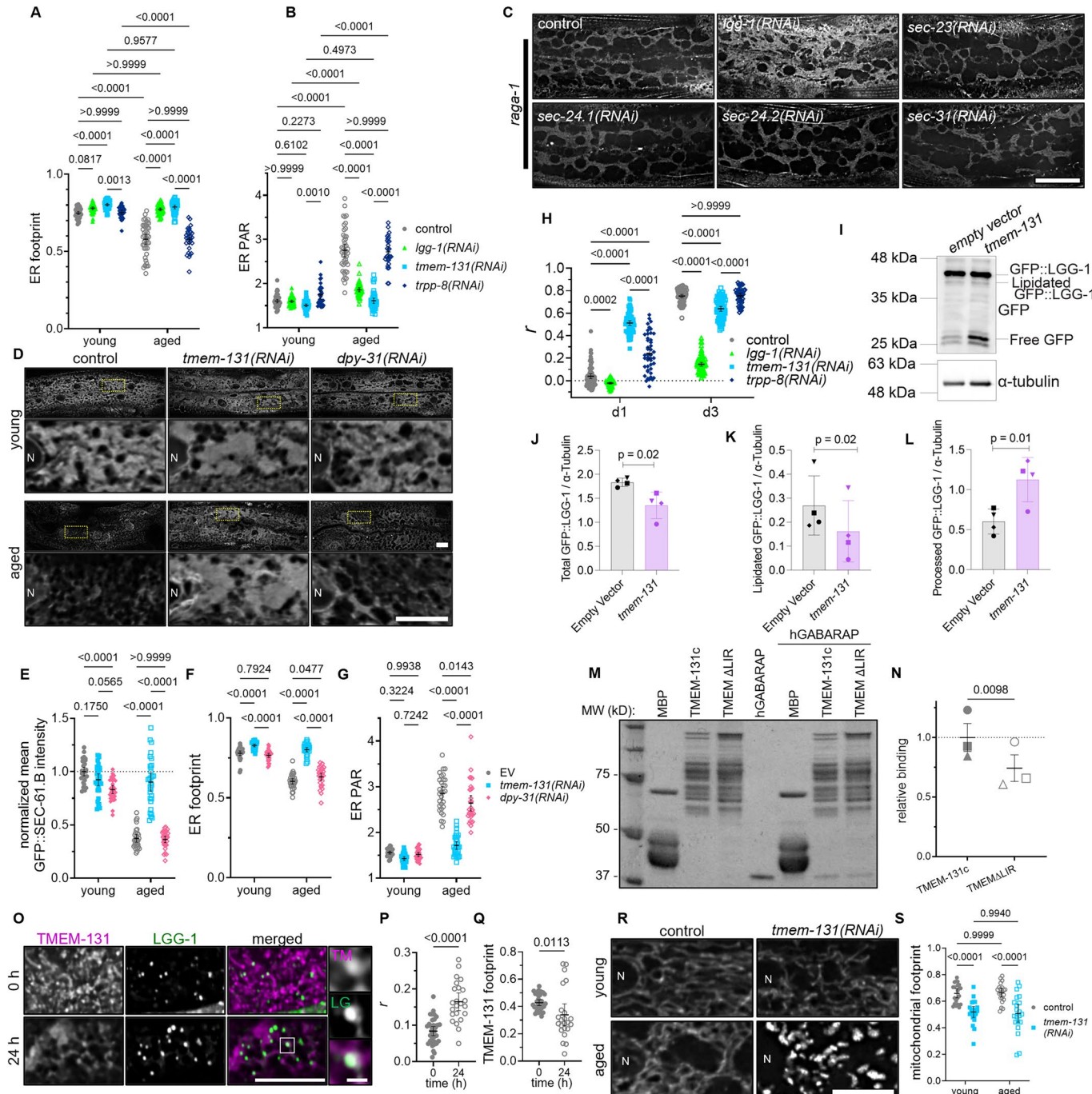

**Extended Data Fig. 8 | See next page for caption.**

**Extended Data Fig. 8 | TMEM-131 interactions with the autophagy machinery.**
**a**,**b**, Footprint (**a**) and PAR (**b**) of hypodermal GFP::SEC-61.B in young and aged control, *lgg-1(RNAi)*, *tmem-131(RNAi)*, and *trpp-8(RNAi)* worms. Young = filled shapes; aged = hollow shapes. $N_{e.v.}$, $N_{lgg-1}$, and $N_{tmem-131}$ = 45 worms per group, pooled across three replicates; $N_{trpp-8}$ = 30 worms per group, pooled across two replicates. Analysed via two-way ANOVAs, post hoc Šidák tests. Data from same experiment as Fig. 7c. **c**, Representative confocal imaging of hypodermal RET-1::GFP in *raga-1* worms fed dsRNA targeting empty vector control, *lgg-1*, or COPII components *sec-23*, *sec-24.1*, *sec-24.2*, or *sec-31* as part of ER-phagy regulator screen. **d**, Representative confocal imaging of hypodermal GFP::SEC-61.B in young and aged control, *tmem-131(RNAi)* and *dpy-31(RNAi)* worms. Scale: 10 µm. **e**–**g**, Quantitation of hypodermal GFP::SEC-61.B, including normalized mean intensity (**e**), footprint (**f**), and PAR (**g**) in young and aged control, *tmem-131(RNAi)*, *dpy-31(RNAi)* worms. Young = filled shapes; aged = hollow shapes. N = 45 worms per group, pooled across three replicates. Analysed via two-way ANOVAs, post hoc Šidák tests. **h**, Pearson's correlation coefficient (*r*) between mCherry and GFP in the hypodermis of control, *lgg-1(RNAi)*, *tmem-131(RNAi)*, and *trpp-8(RNAi)* worms at d1 and d3. $N_{e.v.}$ = 75 d1, 66 d3; $N_{lgg-1}$ = 75 d1, 76 d3; $N_{tmem-131}$ = 75 d1, 72 d3; $N_{trpp-8}$ = 45 d1, 44 d3; pooled across three to five replicates. Analysed via two-way ANOVA, post hoc Šidák test. **i**, Representative western blot of GFP::LGG-1 transgenic animals fed control or *tmem-131* dsRNA. **j**–**l**, Quantification

of indicated GFP::LGG-1 species from **i**. N = 4 independent replicates, indicated by similar shapes. Analysed via two-tailed Student's *t*-test. Error bars indicate ± SD. **m**, Representative Coomassie-stained gel from MBP pull-down experiments using recombinant C-terminal TMEM-131 (TMEM-131$_c$; residues 1191–1831) and LIR mutant TMEM-131$_c$ with His6×-hGABARAP, using free MBP as a negative control. hGABARAP was loaded at a 2% positive control. **n**, Quantification of TMEM-131 binding to hGABARAP. N = 3 independent replicates, indicated by similar shapes. Analysed via two-tailed ratio paired *t*-test. **o**, Representative confocal images of animals co-labelled with TMEM-131::mKate2 and GFP::LGG-1 with zoom inset showing co-labelled TMEM-131( + )/LGG-1(+) puncta. Scale: 10 µm. **p**,**q**, Quantification of TMEM-131::mKate2/GFP::LGG-1 co-localization (**p**) and TMEM-131::mKate2 footprint (**q**) with and without starvation. $N_{control}$ = 35 (P) or 34 (**q**), $N_{starved}$ = 23 (**p**,**q**), pooled across two replicates. Analysed via two-tailed Student's *t*-tests. **r**, Representative confocal images of endogenous mitochondrial COX-4::GFP in the hypodermis of young (d1) and aged (d7) animals. Scale: 10 µm. **s**, Quantification of COX-4::GFP footprint. N = 21 in all conditions, pooled across three replicates. Analysed two-way ANOVA, post hoc Šidák test. For graphs **a**,**b**,**e**–**g**,**h**,**p**,**q**,**s**, error bars indicate mean ± 95% CI; for graphs **j**–**l**: mean ± SD; for graph **n**: mean ± SEM. For images, "N" = nucleus. Source numerical data and unprocessed blots are provided.

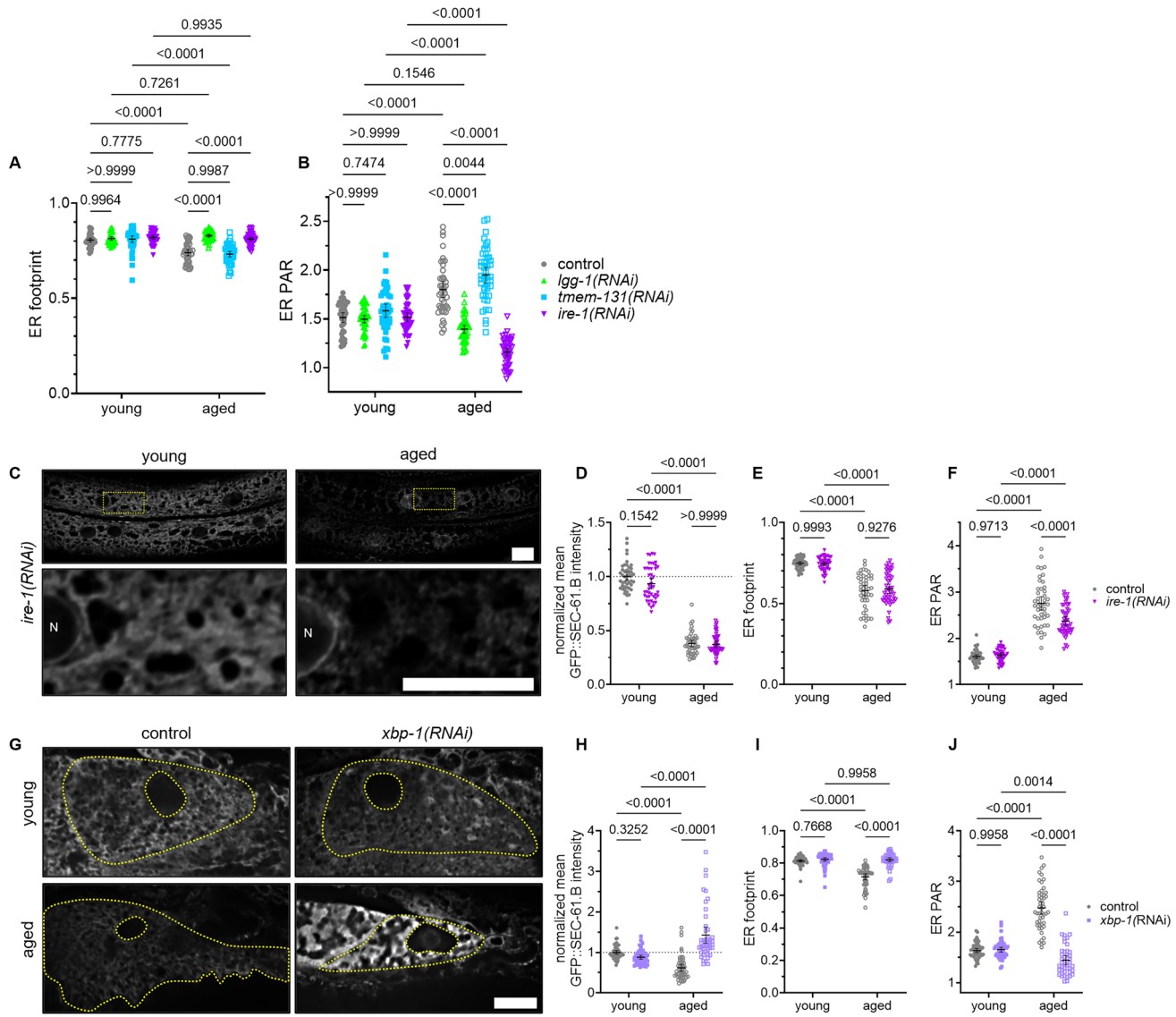

**Extended Data Fig. 9 | Tissue-specific roles for IRE-1–XBP-1 in age-induced ER-phagy. a,b,** Intestinal GFP::SEC-61.B footprint (**a**) and PAR (**b**) in young and aged control, *lgg-1(RNAi)*, *tmem-131(RNAi)*, and *ire-1(RNAi)* worms. Young = filled shapes; aged = hollow shapes. $N_{e.v.}$ = 42 young, 44 aged; $N_{lgg-1}$ = 45 young, 44 aged; $N_{tmem-131}$ = 45 young, 45 aged; $N_{ire-1}$ = 45 young, 45 aged; pooled across three replicates. Analysed via two-way ANOVAs, post hoc Šidák tests. Data from same experiment as Fig. 7i. **c,** Imaging of hypodermal GFP::SEC-61.B in young and aged *ire-1(RNAi)* worms. Scale: 10 μm. "N" = nucleus. **d–f,** Hypodermal GFP::SEC-61.B normalized mean intensity (**d**), footprint (**e**), and PAR (**f**) in young and aged control and *ire-1(RNAi)* worms. Young = filled shapes; aged = hollow shapes.

N = 45 worms per group, pooled across three replicates. Experiment conducted in parallel with conditions in Fig. 7c and Extended Data Fig. 8a,b, using the same control groups. Analysed via two-way ANOVAs, post hoc Šidák tests. **g,** Imaging of intestinal GFP::SEC-61.B in young and aged control and *xbp-1(RNAi)* worms. Scale: 10 μm. Dotted yellow lines demarcate the cell membrane and nuclear envelope. **h–j,** Intestinal GFP::SEC-61.B normalized mean intensity (**h**), footprint (**i**), and PAR (**j**) in young and aged control and *xbp-1(RNAi)* worms. Young = filled shapes; aged = hollow shapes. $N_{e.v.}$ = 52 young, 49 aged; $N_{xbp-1}$ = 53 young, 44 aged; pooled across three replicates. Analysed via two-way ANOVAs, post hoc Šidák tests. For all graphs, error bars indicate mean ± 95% CI. Source numerical data are provided.

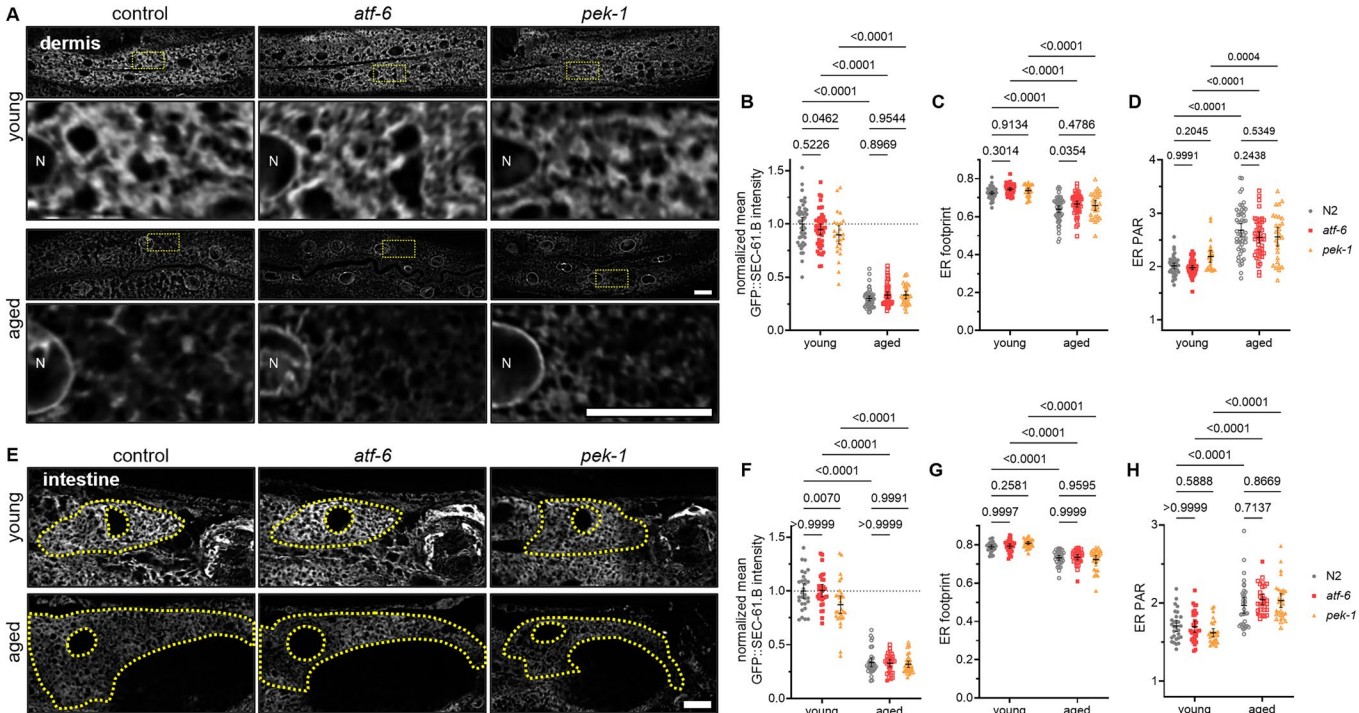

**Extended Data Fig. 10 | ATF-6 and PERK are not involved in age-dependent ER dynamics. a**, Imaging of hypodermal GFP::SEC-61.B in young and aged control, *atf-6*, and *pek-1* worms. All scales: 10 μm. "N" = nucleus. **b**–**d**, Quantification of normalized mean intensity (**b**), footprint (**c**), and PAR (**d**) of hypodermal GFP::SEC-61.B in young and aged control, *atf-6*, and *pek-1* worms. Young = filled shapes; aged = hollow shapes. $N_{control}$ = 43 young, 47 aged and $N_{atf-6}$ = 47 young, 49 aged, pooled across three replicates; $N_{pek-1}$ = 25 young, 29 aged, pooled across two replicates. Analysed via two-way ANOVAs, post hoc Šidák tests. **e**, Imaging of intestinal GFP::SEC-61.B in young and aged control, *atf-6*, and *pek-1* worms. Scale: 10 μm. Dotted yellow lines demarcate the cell membrane and nuclear envelope. **f**–**h**, Quantification of normalized mean intensity (**f**), footprint (**g**), and PAR (**h**) of intestinal GFP::SEC-61.B in young and aged control, *atf-6*, and *pek-1* worms. Young = filled shapes; aged = hollow shapes. $N_{control}$ = 29 young, 34 aged; $N_{atf-6}$ = 30 young, 30 aged; $N_{pek-1}$ = 29 young, 30 aged, pooled across three replicates. Analysed via two-way ANOVAs, post hoc Šidák tests. For all graphs, error bars indicate mean ± 95% CI. Source numerical data are provided.

# Reporting Summary

## Statistics

For all statistical analyses, confirm that the following items are present in the figure legend, table legend, main text, or Methods section.

| n/a | Confirmed | |
|---|---|---|
| ☐ | ☒ | The exact sample size (*n*) for each experimental group/condition, given as a discrete number and unit of measurement |
| ☐ | ☒ | A statement on whether measurements were taken from distinct samples or whether the same sample was measured repeatedly |
| ☐ | ☒ | The statistical test(s) used AND whether they are one- or two-sided *Only common tests should be described solely by name; describe more complex techniques in the Methods section.* |
| ☒ | ☐ | A description of all covariates tested |
| ☐ | ☒ | A description of any assumptions or corrections, such as tests of normality and adjustment for multiple comparisons |
| ☐ | ☒ | A full description of the statistical parameters including central tendency (e.g. means) or other basic estimates (e.g. regression coefficient) AND variation (e.g. standard deviation) or associated estimates of uncertainty (e.g. confidence intervals) |
| ☐ | ☒ | For null hypothesis testing, the test statistic (e.g. *F*, *t*, *r*) with confidence intervals, effect sizes, degrees of freedom and *P* value noted *Give P values as exact values whenever suitable.* |
| ☒ | ☐ | For Bayesian analysis, information on the choice of priors and Markov chain Monte Carlo settings |
| ☒ | ☐ | For hierarchical and complex designs, identification of the appropriate level for tests and full reporting of outcomes |
| ☐ | ☒ | Estimates of effect sizes (e.g. Cohen's *d*, Pearson's *r*), indicating how they were calculated |

*Our web collection on statistics for biologists contains articles on many of the points above.*

## Software and code

Policy information about availability of computer code

| Data collection | ZEN Blue v3.3<br>Nikon Elements AR v5.21.03<br>SerialEM<br>ATLAS v5.0<br>Amersham Imager 600 v1.2.0 |
|---|---|
| Data analysis | Nikon Elements AR v5.21.03<br>ImageJ v1.54f<br>Amira<br>Prism v9/v10 |

For manuscripts utilizing custom algorithms or software that are central to the research but not yet described in published literature, software must be made available to editors and reviewers. We strongly encourage code deposition in a community repository (e.g. GitHub). See the Nature Portfolio guidelines for submitting code & software for further information.

## Data

Policy information about availability of data

All manuscripts must include a data availability statement. This statement should provide the following information, where applicable:
- Accession codes, unique identifiers, or web links for publicly available datasets
- A description of any restrictions on data availability
- For clinical datasets or third party data, please ensure that the statement adheres to our policy

Source data have been provided in Source Data. All other data supporting the findings of this study are available from the corresponding author on reasonable request. Worm strains generated in this study will be available on request and/or deposited to the Caenorhabditis Genetics Center (CGC) repository as appropriate. Yeast strains generated in this study will be available on request. Further information and requests for data, resources, and reagents should be directed to and will be fulfilled by the lead contact, Kristopher Burkewitz (kristopher.burkewitz@vanderbilt.edu). This paper does not report original code.

## Research involving human participants, their data, or biological material

Policy information about studies with human participants or human data. See also policy information about sex, gender (identity/presentation), and sexual orientation and race, ethnicity and racism.

| | |
|---|---|
| Reporting on sex and gender | N/A |
| Reporting on race, ethnicity, or other socially relevant groupings | N/A |
| Population characteristics | N/A |
| Recruitment | N/A |
| Ethics oversight | N/A |

Note that full information on the approval of the study protocol must also be provided in the manuscript.

# Field-specific reporting

Please select the one below that is the best fit for your research. If you are not sure, read the appropriate sections before making your selection.

☒ Life sciences        ☐ Behavioural & social sciences        ☐ Ecological, evolutionary & environmental sciences

For a reference copy of the document with all sections, see nature.com/documents/nr-reporting-summary-flat.pdf

# Life sciences study design

All studies must disclose on these points even when the disclosure is negative.

| | |
|---|---|
| Sample size | Sample sizes were based on previously published experiments (PMID: 32905769, 37127715, 32690699). |
| Data exclusions | According to pre-established criteria, some images were excluded from analysis if they contained obvious technical artifacts such as subject motion or bubbles in the immersion oil. These necessary exclusions allow for appropriate comparisons between biological conditions. |
| Replication | Findings were verified by conducting independent experiments. Replicate numbers are stated for each experiment in the corresponding legend. |
| Randomization | Experimental worms were randomized to their conditions by picking random, healthy parental worms to experimental plates for synchronization. After treatments, worms were then randomly picked for imaging. |
| Blinding | For analyses requiring manual thresholding or segmentation, the experimenter was blinded to experimental conditions prior to imaging or analysis by another lab member or the Fiji script blind-files, provided as part of the Lab-utility-plugins update site. Unblinded experiments such as western blotting utilized identical thresholds and areas across all groups to reduce bias. |

# Reporting for specific materials, systems and methods

We require information from authors about some types of materials, experimental systems and methods used in many studies. Here, indicate whether each material, system or method listed is relevant to your study. If you are not sure if a list item applies to your research, read the appropriate section before selecting a response.

## Materials & experimental systems

| n/a | Involved in the study |
|---|---|
| ☐ | ☒ Antibodies |
| ☒ | ☐ Eukaryotic cell lines |
| ☒ | ☐ Palaeontology and archaeology |
| ☐ | ☒ Animals and other organisms |
| ☒ | ☐ Clinical data |
| ☒ | ☐ Dual use research of concern |
| ☒ | ☐ Plants |

## Methods

| n/a | Involved in the study |
|---|---|
| ☒ | ☐ ChIP-seq |
| ☒ | ☐ Flow cytometry |
| ☒ | ☐ MRI-based neuroimaging |

## Antibodies

| | |
|---|---|
| Antibodies used | 1) Monoclonal ANTI-FLAG® M2 antibody \| Millipore Sigma \| Cat# F1804-50UG, used at 1:1,500<br>2) Monoclonal GFP Antibody (B-2); lot #L1522 \| Santa Cruz Biotechnology \|Cat# sc-9996, used at 1:1,000<br>3) Goat Anti-Mouse IgG H&L (HRP); lot #GR3279214-2 \| Abcam \|Cat# ab205719, used at 1:10,000<br>4) Monoclonal anti-alpha-tubulin, DHSB Cat#12G10<br>5) Monoclonal anti-GFP (7.1+13.1), Roche, Cat#11814460001 |
| Validation | Antibodies were used in western blot experiments and validated for C. elegans through the use of epitope-lacking or RNAi depleted negative controls. |

## Animals and other research organisms

Policy information about studies involving animals; ARRIVE guidelines recommended for reporting animal research, and Sex and Gender in Research

| | |
|---|---|
| Laboratory animals | The C. elegans strains used in this study are N2 Bristol, BQ5, BUZ12, BUZ13, BUZ14, BUZ110, BUZ112, BUZ137, BUZ141, BUZ161, BUZ205, BUZ232, BUZ240, CB4037, WBM509, WBM875, BUZ337, BUZ342, and JJ2586. Genotypes may be found in Supplementary Table 4 (ST4), and ages are noted in each figure.<br><br>For S. cerevisiae, the strains used for imaging were derived from SEY6210, and the strains used for the lifespans were derived from BY4741. Genotypes may be found in ST4, and ages are noted in each figure.<br><br>The M. musculus used were male FVB/NJ, and they were either 6 or 18 months old. |
| Wild animals | N/A |
| Reporting on sex | Standard practice is to use hermaphrodite C. elegans for imaging experiments. Due to concerns regarding energetic burdens associated with hermaphrodite reproduction, we established that the age-related changes were independent of sex (Fig. 3M-P and data not shown). Therefore, all other worm experiments use hermaphrodites only. For mouse SEM imaging, male mice were used for each age. As sex-independence was already demonstrated, and this experiment was included only to test for conservation in a third species, a larger and more expensive sex-based analysis was not conducted. |
| Field-collected samples | N/A |
| Ethics oversight | All mouse procedures were approved by the Institutional Animal Care and Use Committee (IACUC) of Vanderbilt University (M2000086-00/01). Approval was not required for experiments with S. cerevisiae or C. elegans. |

Note that full information on the approval of the study protocol must also be provided in the manuscript.

## Plants

| | |
|---|---|
| Seed stocks | N/A |
| Novel plant genotypes | N/A |
| Authentication | N/A |

