## [Peer Review File · Nature Cell Biology]

ER-phagy reshapes the ER during aging and promotes longevity

Corresponding Author: Dr Kristopher Burkewitz

Version 0:

Decision Letter:

Dear Dr Burkewitz,

Thank you again for submitting your manuscript "ER-phagy drives age-onset remodeling of endoplasmic reticulum structure and lifespan", to Nature Cell Biology. It has now been seen by 3 referees, who are experts in aging, *C. elegans* (Referee #1); ER (Referee #2); and autophagy, selective autophagy (Referee #3), and whose comments are pasted below. In light of their advice, we regret that we cannot offer to publish the study in Nature Cell Biology.

As you will see, although the reviewers found the work interesting, they raised serious concerns that question the strength of the data and of the novel conclusions that can be drawn at this stage. In particular, the reviewers had substantial concerns about the characterizations of ER changes in aging, the links to aging and longevity, the potential conservation of the phenomenon and the implications of autophagy and particular proteins. We have discussed their remarks and unfortunately came to the conclusion that the dataset is too preliminary for publication. We overall decided to return the manuscript to you as the outcome of the new experimentation necessary to address all the comments is uncertain.

Given interest in this area, and the detailed guidance from the reviewers, we would be open to the possibility of considering a revised manuscript that would fully address the referee concerns through an appeal, should you decide to tackle the reviews. However, any decision to re-review such a revised study would depend on the strength of the revisions and the published literature at the time of resubmission.

We are very sorry that we could not be more positive on this occasion, but we thank you for the opportunity to consider this work. We hope you find the reviews constructive as you define the next steps for the manuscript.

With kind regards,
Melina

Melina Casadio, PhD
Senior Editor, Nature Cell Biology
Consulting Editor, Nature Structural & Molecular Biology
ORCID ID: <https://orcid.org/0000-0003-2389-2243>

Reviewers' comments:

Reviewer #1 (Remarks to the Author):

In this manuscript, the authors report age-related ER changes in *C. elegans*. Using fluorescent proteins tagged with tubular and rough ER markers, they revealed the tissue-specific distribution preferences of ER subdomains in adult worms. The authors demonstrated that levels of ER proteins SEC-61.B and RET-1 dramatically decrease beginning at day 3 and continue through day 7. Concurrently, the ER network in the hypodermis shifts from densely packed rough ER sheets to sparsely distributed tubules. Importantly, these age-related changes are observed in both males and hermaphrodites. The authors further analyzed other tissues, observing differential results for SEC-61.B and RET-1. Interestingly, in yeast cells, they also observed an age-related decrease in SEC61 levels, while in mouse cortical neurons, an age-related reduction in ER tubules was detected. Mechanistically, they found that autophagy-mediated ER degradation drives the age-related loss of ER mass and altered morphology. Interestingly, in several long-lived *C. elegans* models, young animals displayed ER changes resembling those observed in aged wild-type worms, which is somewhat counterintuitive. Additionally, in yeast, deletion of ER-phagy-specific factors partially suppressed lifespan extension induced by rapamycin treatment. Lastly, the authors identified two factors, TMEM-131 and IRE-1, as key mediators of ER remodeling in the worm hypodermis and intestine, respectively. Overall, the manuscript is well-written, reporting interesting ER phenotypic changes associated with aging and uncovering new factors that mediate these changes in a tissue-specific manner. However, there are several places where the authors' claims appear overstated based on the current data. Additionally, the relationship between the observed ER changes and aging/healthy aging/longevity remains inconclusive. These concerns need to be addressed before the manuscript can be considered for publication.

Major points:

1. The manuscript focuses on SEC-61.B and RET-1 as markers for age-related ER changes. However, it is unclear whether the observed decreases in protein levels are generalizable to other ER proteins. A broader analysis of additional ER-resident proteins would strengthen the claim of a systemic decline in ER protein levels with age. Furthermore, the observed reduction in rough ER sheets raises questions about its impact on protein translation. With the decrease of rough ER sheets, the translational level is expected to be

decreased.

2. The use of two ER proteins SEC-61.B and RET-1 to quantify age-related ER changes reveals tissue-specific difference, which are distinct from the findings in the hypodermis. Without further evidence like TEM, the authors should be careful about their claim that "Collectively, the shift towards more tubular morphologies observed across cell types..". In addition, to support the conservation of age-onset ER remodeling across species, the authors should quantify ER footprint/PAR in yeast cells as well as SEC61 levels in mouse neurons.
3. Upon unc-51 RNAi knockdown, the ER morphology based on SEC-61.B and RET-1 markers in the hypodermis looks very different, particularly in the RET-1 images (Fig. S4A). The ER network appears highly fused. The authors should note this dramatic morphological change and provide an explanation for this observation. This difference raises questions about the other impact of unc-51 knockdown on ER integrity. The authors' claim that "knockdown of either Atg1/unc-51 or Atg8/lgg-1 suppressed virtually all age-onset changes" should be revised. Similarly, lgg-1 RNAi dramatically affects RET-1 levels in the intestine (Fig. S4I). The authors should also acknowledge this finding and ensure a precise interpretation of the data to provide accurate representation of the results.
4. The observation that ER changes in long-lived worms resemble those observed in aged wild-type worms is unexpected and seemingly contradictory to what would be anticipated. To address this, the authors should examine whether age-related ER changes under these long-lived conditions are attenuated as the animals age, despite the phenotypic similarities observed at the young age. Longitudinal analysis of ER morphology and function in these long-lived models would help clarify whether the observed changes.
5. Dpy-31 RNAi partially rescues the age-related ER changes, as shown in Fig. S6A-D. Based on this result, collagen maturation may still be involved in these processes. The authors should discuss this result with precision. In addition, compared to the control, dpy-31 RNAi decreases the SEC-61.B level in young worms, but not in aged worms. As a result, the age-related reduction of SEC-61B is less pronounced in dpy-31 RNAi worms than in control worms. This comparison should be explicitly noted.
6. Based on the images shown in Fig. 6I, I am not fully convinced that TMEM-131 RNAi suppresses the age-induced colocalization of ER and lysosomes to the same extent as LGG-1 RNAi. There are much more puncta that remain positive for both TRAP-1 and SCAV-3 with TMEM-131 RNAi. To clarify this result, please provide images with TRAP-1::mCherry and SCAV-3::GFP channels separated. This would help better visualize and quantify the degree of colocalization.
7. It is premature to designate TMEM-131 and IRE-1 as ER-phagy receptors without further analysis, including confirming their direct binding to ATG8 in a LIR dependent manner, assessing their recruitment to autophagosomes, and validating their specificity to well-defined ER-phagy induction. The authors should consider adding more experiments to support their claims or dampening their statements.
8. The evidence supporting the notion that ER-phagy-dependent turnover of ER has a positive effect on healthy aging and longevity is currently lacking. The authors should test whether TMEM-131 and IRE-1 are required for ER loss and the lifespan extension observed under different pro-longevity mechanisms.
9. The association between ER loss and ER-UPR requires further investigation. Are other branches of ER-UPR, beyond IRE-1/XBP-1, also involved in mediating the ER loss in the intestine during aging? Are ER-UPR up-regulated in the long-lived models, considering the ER loss observed in these worms?

Minor points:

1. Line 183, Fig. 2A should be Fig. S3F.
2. Line 186, Fig. S2F-G should be Fig. S3F-G
3. Line 187, Fig. S2F and S2H should be S3F and S3H
4. From the representative images shown in Fig. S3I, the fluorescence intensity decrease is stronger in males than in hermaphrodites. However, the quantification result in Fig. S3J shows the opposite. Please make sure to provide correct representative images.
5. Line 201, it is unclear what refers to Fig. 22.

Reviewer #2 (Remarks to the Author):

In this manuscript, Donahue, EK F., et al. reports that aging promotes significant remodeling of ER morphology, characterized by loss of ER mass and a shift in ER sheets to tubules ratio, in various tissues of *C. elegans*. Changes in ER structure is also observed in a model of yeast chronological aging and possibly in neurons of mice. The authors further demonstrate that ER remodeling in aging results from increased levels of autophagy/ERphagy, as downregulation of core components of the autophagy machinery, such as ATG8/lgg-1, rescues ER morphology in old worms. The authors also show that in *C. elegans* ERphagy occurs through tissue-specific mechanisms. While in the hypodermis ERphagy is partially depends on the newly identified TMEM-131 receptor, in the intestine, it is regulated by the IRE-XBP-1 branch of the UPR.

Overall, this is a well written manuscript that presents intriguing findings. However, in several instances, the data provided is overinterpreted. Also, the generalization of the findings across tissues and species are not well supported by the data. Additional data is required to substantiate the exact changes in ER morphology across the different models.

Major comments:

1. The manuscript presents data demonstrating a significant decrease in ER mass in *C. elegans* tissues such as the hypodermis and intestine with aging. This reduction appears to affect both ER sheets and tubules. While the authors suggest a shift in ER domains as a potential adaptive response, this conclusion is not convincingly supported by the data. For instance, in the intestine, the reduction in Ret1-GFP fluorescence is even more pronounced than in SEC61B-GFP. It is essential to clarify whether aging impacts the expression levels of specific ER proteins, such as SEC61, TRAP, Ret-1, or if it in fact results in a massive loss of ER. Although TEM images are provided for the hypodermis (Fig. 2F), it is focused on a very small area and lack clarity. The authors should include lower-magnification and more representative TEM images of both the hypodermis and intestine to better illustrate the observed ER structural changes. Also, the authors should provide fluorescence imaging (either staining or addition endogenous tagged proteins) and western blots for additional ER proteins, such as calreticulin and GRP78, which are more uniformly distributed across ER subdomains.
2. In Fig. 2A, the expression of SEC61B seems to occur inside the nucleus in aged worms. While distortions in nuclear envelope shape are expected during aging, as supported by the article referenced by the authors, this fluorescence pattern is puzzling. Could the authors clarify or provide an explanation for this observation?
3. Since the expression of all the endogenously tagged proteins investigated (SEC61, Ret-1, and TRAP) decreases with aging, it would

be beneficial to include a cytosolic-tagged protein (or another subcellular location) as a control to demonstrate that this is not a general adaptation in worms to degrade tagged proteins over time.

4. The conclusion that the yeast phenotype promoted by aging is similar to the phenotype observed in *C. elegans* tissues is not well supported by the data. The study only presents the Sec61-GFP marker as evidence. It is possible that the observed changes reflect a decrease in Sec61 expression during aging rather than a reduction in overall ER levels. TEM data and additional markers are essential to substantiate the phenotype.

5. In the attempt to generalize their mechanism to mammalian cells/tissues, the authors provide TEM images of motor cortical neurons. It is unclear why the authors chose to perform the experiments in neurons. The most robust data in *C. elegans* is observed in the secretory tissues such as hypodermis and intestine. It would be more logical to explore the effect of aging in tissues with a high abundance of ER sheets and secretory ER such as the liver or pancreas. Neurons exhibit a very particular and specialized ER organization, and the images provided do not clearly demonstrate dramatic changes in ER content. Generalizing this phenotype across other tissues and species based solely on these results is an overreach.

6. The proteomic analysis Fig 2I and J lacks information. Was it performed in specific tissues or whole worms. If the latter, it's hard to relate to the particular ER phenotypes in different tissues.

7. The data demonstrating the rescue of ER mass by Atg8/lgg-1 is strong and clear. However, the rescue phenotype promoted by deletion of Tmem-131 in hypodermis is not convincing. While ER mass appears to increase, the morphology differs noticeably from that of the control. Therefore, referring to this as a "rescue" is an overstatement. Additionally, the impact of Tmem-131 downregulation on Ret-1-GFP is not addressed and should be included to provide a more comprehensive analysis.

8. As acknowledged by the authors, the changes in ER morphology, based on SEC61B-GFP, observed by downregulation of daf-2, raga-1, glp-1 and ifg-1 differ significantly from the loss of ER mass reported in aged worms. Specifically, the mutants show the formation of perinuclear ER patches. These data do not add much to the paper since the authors do not explore its relevance for life span extension, ER function etc...

9. In mammalian cells, XBP1 is well known to drive ER biogenesis specially in secretory cells such as plasma cells, acinar cells (e.g . PMID: 16362047). It's not clear to me how loss of XBP1 would restore ER mass in the intestine?

Minor comments

1. Fig. S4I, why is the fluorescence over saturated in the lgg-1 rescue experiment- it seems the images were not acquired in the same condition.

2. Line 201: Paper references Figure 22 which does not exist. Probably a typo.

3. Line 221: In figure S4M, legend states "during early aging" but no further clarification is included on when this reporter accumulation occurs

4. Line 303: The authors refer to figure S6H-I to demonstrate effects of TMEM131 but it actually shows XBP1 effects. Probably a mislabeling.

Reviewer #3 (Remarks to the Author):

The authors report the interesting phenomenon that the ER levels decrease during aging across species in yeast, *C. elegans*, and mouse. This study applied different approaches to indicate autophagy regulates ER clearance in the aging worm, and putative LIR containing-TMEM-131 regulates ER-phagy. The discovery of ER-phagy in *C. elegans* is novel and the physiological relevance of aging and ER health is important.

Comments to the authors:

Major:

1. Evidence of TMEM-131 as an ER-phagy receptor is not convincing. Since the essential amino acids of putative LIR domain are conserved, evidence of whether these mutation of these sites affect ER clearance is required. Additionally, existing study indicates TMEM-131 interacts with COPII vesicles to facilitate collagen secretion (PMID: 32095531) and COPII components are related to distinct ER clearance pathways (PMID: 31273116, 38593803). It is unclear whether TMEM-131 regulates COPII vesicle formation and whether core COPII coat proteins are required for ER clearance.

2. More evidence about whether TMEM-131 affects bulk autophagy substrates including SQST-1 and LGG-1 is required to strengthen TMEM-131 selectively regulates ER clearance.

3. The TEM results clearly showed ER level decrease in aging worm body but lack key evidence that ER structures exist in autophagosomes. Given the significant decrease of ER during aging, ER structures should be seen enclosed by autophagosomes. If autophagosomes undergo rapid clearance in the worm body, employing genetic or chemical treatments that block lysosomal function and inhibit autophagosome degradation could increase the chances of capturing autophagosomes by TEM.

Minor:

1. Biochemistry experiments indicate more RET-1 is degraded compared to SEC61-B (Fig. S2 I-L) that are not consistent with the observations by fluorescent analysis. The authors need to test whether TMEM-131, UNC-51, or LGG-1 affects RET-1 and SEC61-B protein levels via western blot in aging worms to strengthen that the degradation of these ER subdomain proteins is mainly dependent of autophagy. Descriptions about Fig S2 I-L in the main text are missing.

2. A previous study indicates that IRE-1-XBP-1 mediated ER-UPR is activated in tmem-131 mutant worms that is caused by the failure of collagen secretion (PMID: 32095531). Is IRE-1 required for TMEM-131 dependent ER clearance in hypodermis?

3. Line 303, 307, 314 "Fig. S6M-Q" are not related to the figure.

**For Nature Portfolio general information and news for authors, see <http://npg.nature.com/authors>.

Version 1:

Decision Letter:

Dear Dr Burkewitz,

Thank you for your email asking us to reconsider our decision on your manuscript, "ER-phagy drives age-onset remodeling of endoplasmic reticulum structure-function and lifespan". We are always willing to hear the authors' perspective, but we must first prioritize decisions on new submissions. We appreciate your patience while we considered this appeal.

I have now discussed your manuscript, and the referees' comments, and your rebuttal, in detail with my colleagues, and we are willing to reconsider a revised manuscript provided the following files can be provided for re-review, and that nothing similar is accepted for publication at Nature Cell Biology or published elsewhere in the meantime:

- Please edit the point-by-point response to the reviews to either directly show the data in each response to reviewers' comments or to please explicitly point the reviewers to where the new data are presented in the revised manuscript (e.g., figure XA, etc). It is otherwise challenging and time-consuming to find the new data while assessing the rebuttal and revised manuscript.

- Please provide a Supplementary Figure including unprocessed images of all gels/blots in the form of a multi-page pdf file. Please ensure that blots/gels are labeled and the sections presented in the figures are clearly indicated.

- Please provide a Supplementary Table including all numerical source data in Excel format, with data for different figures provided as different sheets within a single Excel file. The file should include source data giving rise to graphical representations and statistical descriptions in the paper and for all instances where the figures present representative experiments of multiple independent repeats, the source data of all repeats should be provided.

- On resubmission please provide the completed Reporting Summary (found here <https://www.nature.com/documents/nr-reporting-summary.pdf>). This is essential for reconsideration of the manuscript and this document will be available to editors and referees. For more information see below. Please also ensure that the presentation of statistical information in the revised submission complies with Nature Cell Biology's statistical guidelines (see below).

Please use the link below to submit the complete manuscript files, and include a point-by-point response to the complete reviewer comments, verbatim as provided in their reports.

Link Redacted

Please let us know how you wish to proceed and when we can expect your revised manuscript. Thank you for considering NCB for your work.

With kind regards,

Melina Casadio

Melina Casadio, PhD
Senior Editor, Nature Cell Biology
Consulting Editor, Nature Structural & Molecular Biology
ORCID ID: <https://orcid.org/0000-0003-2389-2243>

GUIDELINES FOR EXPERIMENTAL AND STATISTICAL REPORTING

REPORTING REQUIREMENTS – To improve the quality of methods and statistics reporting in our papers we have recently revised the reporting checklist we introduced in 2013. We are now asking all life sciences authors to complete a reporting summary (found here <https://www.nature.com/documents/nr-reporting-summary.pdf>) that collects information on experimental design and reagents. This document is available to referees to aid the evaluation of the manuscript. Please note that this form is a dynamic 'smart pdf' and must therefore be downloaded and completed in Adobe Reader. We will then flatten it for ease of use by the reviewers. If you would like to reference the guidance text as you complete the template, please access these flattened versions at <http://www.nature.com/authors/policies/availability.html>.

STATISTICS – Wherever statistics have been derived the legend needs to provide the n number (i.e. the sample size used to derive statistics) as a precise value (not a range), and define what this value represents. Error bars need to be defined in the legends (e.g. SD, SEM) together with a measure of centre (e.g. mean, median). Box plots need to be defined in terms of minima, maxima, centre, and percentiles. Ranges are more appropriate than standard errors for small data sets. Wherever statistical significance has been derived, precise p values need to be provided and the statistical test used needs to be stated in the legend. Statistics such as error bars must not

be derived from $n < 3$. For sample sizes of $n < 5$ please plot the individual data points rather than providing bar graphs. Deriving statistics from technical replicate samples, rather than biological replicates is strongly discouraged. Wherever statistical significance has been derived, precise p values need to be provided and the statistical test stated in the legend.

Version 2:

Decision Letter:

*Please delete the link to your author homepage if you wish to forward this email to co-authors.

Dear Kris,

Your manuscript, "ER-phagy drives age-onset remodeling of endoplasmic reticulum structure-function and lifespan", has now been seen by the original referees, and their comments are below. As you will see from their comments (attached below), they continue to find the work of interest, and Revs#1-2 are in particular very supportive, but Rev#3 has raised some important points. Although we are also very interested in this study, we believe that their concerns should be addressed before we can consider publication in Nature Cell Biology.

We have discussed the referee reports in detail within the editorial team, including the chief editor. While we limit our manuscripts to a single round of major experimental revision in the interest of limiting the overall time spent in peer review & revision, given the overall support and as the final comments are relatively minor, we are open to a final round of minor revision to address Rev#3's persisting points. They requested additions to strengthen the link between TMEM-131 and ER-phagy. We are committed to providing a fair and constructive peer-review process, so please feel free to contact me if you would like to discuss any of the referee comments further or anticipate issues addressing Rev#3's final points.

Please also pay close attention to our guidelines on statistical and methodological reporting (listed below) as failure to do so may delay the reconsideration of the revised manuscript. In particular, please provide:

- For any revision that includes light microscopy data, we ask our authors to please include a completed light microscopy reporting table [https://www.nature.com/documents/Light_microscopy_reporting_table.xlsx] to ensure the methods are described thoroughly. The table will be available to reviewers and ultimately published should the manuscript be accepted at the journal.

We therefore invite you to take these points into account when revising the manuscript. In addition, when preparing the revision please:

- ensure that it conforms to our format instructions and publication policies (see below and www.nature.com/nature/authors/).

- provide a point-by-point rebuttal to the full referee reports verbatim, as provided at the end of this letter.

Nature Cell Biology is committed to improving transparency in authorship. As part of our efforts in this direction, we are now requesting that all authors identified as 'corresponding author' on published papers create and link their Open Researcher and Contributor Identifier (ORCID) with their account on the Manuscript Tracking System (MTS), prior to acceptance. ORCID helps the scientific community achieve unambiguous attribution of all scholarly contributions. You can create and link your ORCID from the home page of the MTS by clicking on 'Modify my Springer Nature account'. For more information please visit www.springernature.com/orcid.

Link Redacted

We would like to receive the revision within four weeks. If submitted within this time period, reconsideration of the revised manuscript will not be affected by related studies published elsewhere, or accepted for publication in Nature Cell Biology in the meantime. We would be happy to consider a revision even after this timeframe, but in that case we will consider the published literature at the time of resubmission when assessing the file.

We hope that you will find our referees' comments and editorial guidance helpful. Please do not hesitate to contact me if there is anything you would like to discuss. Thank you again for considering NCB for your work.

Best wishes,

Melina

Melina Casadio, PhD
Senior Editor, Nature Cell Biology
Consulting Editor, Nature Structural & Molecular Biology
ORCID ID: <https://orcid.org/0000-0003-2389-2243>

Reviewers' Comments:

Reviewer #1 (Remarks to the Author):

The authors have thoroughly addressed my suggestions and comments, and the revised manuscript is much more improved. I recommend accepting it for publication.

Reviewer #2 (Remarks to the Author):

In the revised version of the manuscript by Donahue EKF et al., the authors have adequately addressed all of my previous comments and concerns. The addition of new data further strengthens their main conclusions regarding the key role of ER-phagy in ER remodeling during aging. I commend the authors on their thorough revisions and recommend that this version be accepted for publication.

Reviewer #3 (Remarks to the Author):

The authors have addressed the key concerns. However, some minor issues need to be addressed:

1. The TEM results (Extended Data Figure 4) show a loss of cellular mass in aging worms. Both the ER and mitochondria appear reduced in the hypodermis and intestines of aged animals, suggesting that both organelles are cleared during aging. Does *tmem-131* affect mitochondrial clearance during aging? Quantification of the TEM results is needed.

2. The Western blot results (Extended Data Figure 8) do not provide conclusive evidence supporting a role for *tmem-131* in selective autophagy. The presence of processed free GFP, cleaved from GFP-LGG-1 by lysosomal hydrolyses, indicates that *tmem-131* RNAi worms show a decrease of main GFP-LGG1 band accompanied by an increase in free GFP compared with controls, suggesting enhanced autophagic flux. This observation appears inconsistent with the author's proposed model. Nevertheless, *tmem-131* may function specifically in regulating ER clearance within certain tissues. Since Western blot analysis of whole animals cannot resolve tissue specific effects, it would be informative to examine whether *tmem-131* RNAi alters LGG-1 co-localization with ER markers such as SEC-61.B or RET-1, given that LGG-1 is recruited to *tmem-131* structures by immunostaining (Extended Data Figure 8).

GUIDELINES FOR SUBMISSION OF NATURE CELL BIOLOGY ARTICLES

ARTICLE FORMAT

ABSTRACT – should not exceed 150 words and should be unreferenced. This paragraph is the most visible part of the paper and should briefly outline the background and rationale for the work, and accurately summarize the main results and conclusions. Key genes, proteins and organisms should be specified to ensure discoverability of the paper in online searches.

TEXT – the main text consists of the Introduction, Results, and Discussion sections and must not exceed 3500 words including the abstract. The Introduction should expand on the background relating to the work. The Results should be divided in subsections with subheadings, and should provide a concise and accurate description of the experimental findings. The Discussion should expand on the findings and their implications. All relevant primary literature should be cited, in particular when discussing the background and specific findings.

REFERENCES – are limited to a total of 70 in the main text and Methods combined. They must be numbered sequentially as they appear in the main text, tables and figure legends and Methods and must follow the precise style of Nature Cell Biology references. References only cited in the Methods should be numbered consecutively following the last reference cited in the main text. References only associated with Supplementary Information (e.g. in supplementary legends) do not count toward the total reference limit and do not need to be cited in numerical continuity with references in the main text. Only published papers can be cited, and each publication cited should be included in the numbered reference list, which should include the manuscript titles. Footnotes are not permitted.

Methods should be written concisely, but should contain all elements necessary to allow interpretation and replication of the results. As a guideline, Methods sections typically do not exceed 3,000 words. The Methods should be divided into subsections listing reagents and techniques. When citing previous methods, accurate references should be provided and any alterations should be noted. Information must be provided about: antibody dilutions, company names, catalogue numbers and clone numbers for monoclonal antibodies; sequences of RNAi and cDNA probes/primers or company names and catalogue numbers if reagents are commercial; cell line names, sources and information on cell line identity and authentication. Animal studies and experiments involving human subjects must be reported in detail, identifying the committees approving the protocols. For studies involving human subjects/samples, a statement must be included confirming that informed consent was obtained. Statistical analyses and information on the reproducibility of experimental results should be provided in a section titled "Statistics and Reproducibility".

All Nature Cell Biology manuscripts submitted on or after March 21 2016, must include a Data availability statement as a separate section after Methods but before references, under the heading "Data Availability". For Springer Nature policies on data availability see <http://www.nature.com/authors/policies/availability.html>; for more information on this particular policy see <http://www.nature.com/authors/policies/data/data-availability-statements-data-citations.pdf>. The Data availability statement should include:

- Accession codes for primary datasets (generated during the study under consideration and designated as "primary accessions") and secondary datasets (published datasets reanalysed during the study under consideration, designated as "referenced accessions"). For primary accessions data should be made public to coincide with publication of the manuscript. A list of data types for which submission to community-endorsed public repositories is mandated (including sequence, structure, microarray, deep sequencing data) can be found here <http://www.nature.com/authors/policies/availability.html#data>.
- Unique identifiers (accession codes, DOIs or other unique persistent identifier) and hyperlinks for datasets deposited in an approved repository, but for which data deposition is not mandated (see here for details <http://www.nature.com/sdata/data-policies/repositories>).
- At a minimum, please include a statement confirming that all relevant data are available from the authors, and/or are included with the manuscript (e.g. as source data or supplementary information), listing which data are included (e.g. by figure panels and data types) and mentioning any restrictions on availability.
- If a dataset has a Digital Object Identifier (DOI) as its unique identifier, we strongly encourage including this in the Reference list and citing the dataset in the Methods.

We recommend that you upload the step-by-step protocols used in this manuscript to protocols.io. More details can found at <https://www.protocols.io/help/publish-articles>.

DISPLAY ITEMS – main display items are limited to 6-8 main figures and/or main tables. For Supplementary Information see below.

FIGURES – Colour figure publication costs \$395 per colour figure. All panels of a multi-panel figure must be logically connected and arranged as they would appear in the final version. Unnecessary figures and figure panels should be avoided (e.g. data presented in small tables could be stated briefly in the text instead).

All imaging data should be accompanied by scale bars, which should be defined in the legend.

Cropped images of gels/blots are acceptable, but need to be accompanied by size markers, and to retain visible background signal within the linear range (i.e. should not be saturated). The boundaries of panels with low background have to be demarked with black lines. Splicing of panels should only be considered if unavoidable, and must be clearly marked on the figure, and noted in the legend with a statement on whether the samples were obtained and processed simultaneously. Quantitative comparisons between samples on

different gels/blots are discouraged; if this is unavoidable, it has to be performed for samples derived from the same experiment with gels/blots were processed in parallel, which needs to be stated in the legend.

Regardless of format, all figures must be vector graphic compatible files, not supplied in a flattened raster/bitmap graphics format, but should be fully editable, allowing us to highlight/copy/paste all text and move individual parts of the figures (i.e. arrows, lines, x and y axes, graphs, tick marks, scale bars etc). The only parts of the figure that should be in pixel raster/bitmap format are photographic images or 3D rendered graphics/complex technical illustrations.

Unprocessed scans of all key data generated through electrophoretic separation techniques need to be presented in a supplementary figure that should be labeled and numbered as the final supplementary figure, and should be mentioned in every relevant figure legend. This figure does not count towards the total number of figures and is the only figure that can be displayed over multiple pages, but should be provided as a single file, in PDF or TIFF format. Data in this figure can be displayed in a relatively informal style, but size markers and the figures panels corresponding to the presented data must be indicated.

The total number of Supplementary Figures (not including the "unprocessed scans" Supplementary Figure) should not exceed the number of main display items (figures and/or tables (see our Guide to Authors and March 2012 editorial <http://www.nature.com/ncb/authors/submit/index.html#suppinfo>; <http://www.nature.com/ncb/journal/v14/n3/index.html#ed>). No restrictions apply to Supplementary Tables or Videos, but we advise authors to be selective in including supplemental data.

GUIDELINES FOR EXPERIMENTAL AND STATISTICAL REPORTING

REPORTING REQUIREMENTS – To improve the quality of methods and statistics reporting in our papers we have recently revised the reporting checklist we introduced in 2013. We are now asking all life sciences authors to complete two items: an Editorial Policy Checklist (found here https://www.nature.com/authors/policies/Policy.pdf) that verifies compliance with all required editorial policies and a Reporting Summary (found here https://www.nature.com/authors/policies/ReportingSummary.pdf) that collects information on experimental design and reagents. These documents are available to referees to aid the evaluation of the manuscript. Please note that these forms are dynamic 'smart pdfs' and must therefore be downloaded and completed in Adobe Reader. We will then flatten them for ease of use by the reviewers. If you would like to reference the guidance text as you complete the template, please access these flattened versions at http://www.nature.com/authors/policies/availability.html.

STATISTICS – Wherever statistics have been derived the legend needs to provide the n number (i.e. the sample size used to derive statistics) as a precise value (not a range), and define what this value represents. Error bars need to be defined in the legends (e.g. SD, SEM) together with a measure of centre (e.g. mean, median). Box plots need to be defined in terms of minima, maxima, centre, and percentiles. Ranges are more appropriate than standard errors for small data sets. Wherever statistical significance has been derived, precise p values need to be provided and the statistical test used needs to be stated in the legend. Statistics such as error bars must not be derived from n<3. For sample sizes of n<5 please plot the individual data points rather than providing bar graphs. Deriving statistics from technical replicate samples, rather than biological replicates is strongly discouraged. Wherever statistical significance has been derived, precise p values need to be provided and the statistical test stated in the legend.

Version 3:

Decision Letter:

Our ref: NCB-A56505C

21st November 2025

Dear Dr. Burkewitz,

Thank you for submitting your revised manuscript "ER-phagy drives age-onset remodeling of endoplasmic reticulum structure-function and lifespan" (NCB-A56505C). Thank you for addressing Rev#3's remaining points as discussed in revision. As Rev#3 had shared, the final revisions have resolved the remaining concerns, and therefore, we'll be happy in principle to publish the manuscript in Nature Cell Biology, pending minor revisions to comply with our editorial and formatting guidelines.

We are now performing detailed checks on your paper and will send you a checklist detailing our editorial and formatting requirements in about 1-2 weeks. Please do not upload the final materials and make any revisions until you receive this additional information from us.

Thank you again for your interest in Nature Cell Biology. Please do not hesitate to contact me if you have any questions.

Sincerely,

Melina Casadio, PhD
Senior Editor, Nature Cell Biology
Consulting Editor, Nature Structural & Molecular Biology
ORCID ID: <https://orcid.org/0000-0003-2389-2243>

Version 4:

Decision Letter:

Dear Dr Burkewitz,

I am pleased to inform you that your manuscript, "ER remodeling is a feature of aging and depends on ER-phagy", has now been accepted for publication in Nature Cell Biology.

Thank you for sending us the final manuscript files to be processed for print and online production, and for returning the manuscript

checklists and other forms. Your manuscript will now be passed to our production team who will be in contact with you if there are any questions with the production quality of supplied figures and text.

Please note that *Nature Cell Biology* is a Transformative Journal (TJ). Authors may publish their research with us through the traditional subscription access route or make their paper immediately open access through payment of an article-processing charge (APC). Authors will not be required to make a final decision about access to their article until it has been accepted. [Find out more about Transformative Journals](https://www.springernature.com/gp/open-research/transformative-journals)

Authors may need to take specific actions to achieve compliance with funder and institutional open access mandates. If your research is supported by a funder that requires immediate open access (e.g. according to [Plan S principles](https://www.springernature.com/gp/open-science/plan-s-compliance) or the [NIH public access policy](https://www.springernature.com/gp/open-science/us-federal-agency-compliance)) then you should select the gold OA route, and we will direct you to the compliant route where possible. Because authors warrant under our subscription licensing terms that they haven't committed to licensing any version of their article under a licence inconsistent with the terms of our agreement – including the applicable embargo period – publication under the subscription model isn't suitable for authors whose funders require no embargo.

If you have not already done so, we strongly recommend that you upload the step-by-step protocols used in this manuscript to protocols.io (<https://protocols.io>), an open online resource that allows researchers to share their detailed experimental know-how. All uploaded protocols are made freely available and are assigned DOIs for ease of citation. Protocols and Nature Portfolio journal papers in which they are used can be linked to one another, and this link is clearly and prominently visible in the online versions of both. Authors who performed the specific experiments can act as primary authors for the Protocol as they will be best placed to share the methodology details, but the Corresponding Author of the present research paper should be included as one of the authors. By uploading your Protocols onto protocols.io, you are enabling researchers to more readily reproduce or adapt the methodology you use, as well as increasing the visibility of your protocols and papers. You can also establish a dedicated workspace to collect your lab Protocols. Further information can be found at <https://www.protocols.io/help/publish-articles>.

Nature Cell Biology encourages authors presenting evidence for cell, biological, molecular, and genetic interactions to consider communicating these findings using Biofactoid (<https://biofactoid.org/>). This tool helps users share a searchable representation of interactions (e.g. binding, gene expression, post-translational modification) between genes, gene products, or chemicals. Information added to Biofactoid, with author attribution, is shared on social media and public databases, such as Pathway Commons, where it can be discovered and analyzed in the context of a large and growing corpus of knowledge.

With kind regards,

Melina Casadio, PhD
Senior Editor, Nature Cell Biology
Consulting Editor, Nature Structural & Molecular Biology
ORCID ID: <https://orcid.org/0000-0003-2389-2243>

** Visit the Springer Nature Editorial and Publishing website at http://editorial-jobs.springernature.com?utm_source=ejp_NCB_email&utm_medium=ejp_NCB_email&utm_campaign=ejp_NCB for more information about our career opportunities. If you have any questions please click [here](mailto:editorial.publishing.jobs@springernature.com).

Reviewers' comments:

Reviewer #1 (Remarks to the Author):

In this manuscript, the authors report age-related ER changes in *C. elegans*. Using fluorescent proteins tagged with tubular and rough ER markers, they revealed the tissue-specific distribution preferences of ER subdomains in adult worms. The authors demonstrated that levels of ER proteins SEC-61.B and RET-1 dramatically decrease beginning at day 3 and continue through day 7. Concurrently, the ER network in the hypodermis shifts from densely packed rough ER sheets to sparsely distributed tubules. Importantly, these age-related changes are observed in both males and hermaphrodites. The authors further analyzed other tissues, observing differential results for SEC-61.B and RET-1. Interestingly, in yeast cells, they also observed an age-related decrease in SEC61 levels, while in mouse cortical neurons, an age-related reduction in ER tubules was detected. Mechanistically, they found that autophagy-mediated ER degradation drives the age-related loss of ER mass and altered morphology. Interestingly, in several long-lived *C. elegans* models, young animals displayed ER changes resembling those observed in aged wild-type worms, which is somewhat counterintuitive. Additionally, in yeast, deletion of ER-phagy-specific factors partially suppressed lifespan extension induced by rapamycin treatment. Lastly, the authors identified two factors, TMEM-131 and IRE-1, as key mediators of ER remodeling in the worm hypodermis and intestine, respectively.

Overall, the manuscript is well-written, reporting interesting ER phenotypic changes associated with aging and uncovering new factors that mediate these changes in a tissue-specific manner. However, these are several places where the authors' claims appear overstated based on the current data. Additionally, and the relationship between the observed ER changes and aging/healthy aging/longevity remains inconclusive. These concerns need to be addressed before the manuscript can be considered for publication.

Major points:

1a. A broader analysis of additional ER-resident proteins would strengthen the claim of a systemic decline in ER protein levels with age.

We agree that examining a broader set of ER-resident proteins is useful to support the generality of age-related ER decline. In the revised manuscript, we now include multiple additional lines of evidence demonstrating that ER loss is not limited to the originally reported markers (SEC-61.B and RET-1). Specifically, we added: 1) Fluorescence imaging of natively tagged HSP-3 and HSP-4, the *C. elegans* orthologs of the mammalian GRP78/BiP chaperone, both of which show robust age-dependent signal loss; 2) Western blot analysis of YOP-1, a second curvature-promoting ER-shaping protein in addition to RET-1, which shows a more modest but detectable decline with age; 3) A complementary ER marker (Elo3::mCherry) in yeast, which also shows age-related decline in abundance; 4) focused analysis of mammalian aging transcriptomic and proteomic atlases, resulting in new data showing consistent and broadscale downregulation of ER proteostasis/secretory components in multiple aged mouse tissues and an upregulation of ER proteins with functions in lipid metabolism, etc. Together, these data provide strong support for a systemic and conserved decline in ER protein abundance during aging, combined with a reconfiguration of protein/functional composition. We have revised the text and figure panels accordingly to reflect these findings.

1b. Furthermore, the observed reduction in rough ER sheets raises questions about its impact on protein translation. With the decrease of rough ER sheets, the translational level is expected to be decreased.

We thank the reviewer for highlighting this important connection, which we did not make clear in the original manuscript. Declines in global protein synthesis and translational machinery are established effects during both aging and in the context of multiple longevity paradigms (PMIDs 30733602,

30562164), and we also include data showing that impairing translation initiation (ifg-1 RNAi) is sufficient to induce the longevity-associated ER remodeling phenotype. Rather than replicate the findings from past studies regarding protein synthesis rates in age, we have cited past work and included more explicit focus on incorporating this point into our model in the Introduction and Discussion.

2. The use of two ER proteins SEC-61.B and RET-1 to quantify age-related ER changes reveals tissue-specific difference, which are distinct from the findings in the hypodermis. Without further evidence like TEM, the authors should be careful about their claim that “Collectively, the shift towards more tubular morphologies observed across cell types..”. In addition, to support the conservation of age-onset ER remodeling across species, the authors should quantify ER footprint/PAR in yeast cells as well as SEC61 levels in mouse neurons.

We thank the reviewer for highlighting the need to more precisely discuss and support the nuanced extrapolation of our observations across both tissues and species. We have provided a new figure of additional EM of aging worms, which supports both an overall loss of ER and a shift from stacked sheet morphologies in aging animals in both the hypodermis and intestine. Additionally, we provide western blots of YOP-1, an alternative tubule-promoting protein which is much more robust to the effects of age than RET-1. This result reveals that the relative composition of specific ER shaping proteins is dynamic across age and provides an explanation for how tubules can be maintained at a higher ratio in some tissues even despite a more pronounced loss of RET-1. However, given the heterogeneity of ER structures and functions across tissue types, we agree with the reviewer that many aspects of ER dynamics are likely to be tissue-specific, and thus we have added this point explicitly in the text while softening claims about cells in which we have not verified closely.

To support conservation across species, we tested an additional ER protein in yeast while adding footprint and PAR measurements; and we mined the Tabula Muris Senis scRNA-seq atlas, revealing that “Protein Processing in the ER” gene set is part of a universal (cross-tissue) aging signature that declines with age across tissues. The newly added mammalian proteomics data correspond well with our *C. elegans* data by demonstrating consistent downregulation of rough ER/protein translocation machinery in skin, liver, and kidney. These multiple levels of evidence reinforce that the cross-tissue, cross-species pattern we observe for the ER is surprisingly robust even in the face of widely varied structural/functional enrichment in distinct cell types at baseline.

Taken together, these new data demonstrate that while aspects of ER remodeling are likely to vary across tissues and contexts — as is true for any complex and dynamic cellular structure — our work establishes the fundamental principle that ER morphology and its functional composition change significantly with age, opening the door to many future investigations of the consequences and mechanisms of these dynamics in diverse systems and contexts.

3. Upon unc-51 RNAi knockdown, the ER morphology based on SEC-61.B and RET-1 markers in the hypodermis looks very different, particularly in the RET-1 images (Fig. S4A). The ER network appears highly fused. The authors should note this dramatic morphological change and provide an explanation for this observation. This difference raises questions about the other impact of unc-51 knockdown on ER integrity. The authors’ claim that “knockdown of either Atg1/unc-51 or Atg8/lgg-1 suppressed virtually all age-onset changes” should be revised. Similarly, lgg-1 RNAi dramatically affects RET-1 levels in the intestine (Fig. S4I). The authors should also acknowledge this finding and ensure a precise interpretation of the data to provide accurate representation of the results.

We thank the reviewer raising the need to better highlight these differences, and we have both revised the text and added new data to explain these key details. First, we contextualize the distinction between ULK1 and lgg-1 effects within established ER-phagy pathways and their differential reliance on these factors. Specifically, only macro-ER-phagy is dependent upon ULK-1, while lgg-1/Atg8 is more broadly required across most ER-phagy pathways, including both macro-ER-phagy and vesicular/ERLAD, as well as certain forms of micro-ER-phagy (e.g., PMID 33765438). Based on this, we propose that sheet-expansion during UNC-51(RNAi) is consistent with macro-ER-phagy playing a disproportionate role in the clearance

of rough ER sheets, while *Atg8/lgg-1*(RNAi) results in complete suppression of ER loss with little overt morphology bias. The stronger overall effects of *lgg-1* and accumulation of RET-1 in the intestine under *lgg-1* RNAi is also consistent with this broader role, and all results are consistent with the relative breadth of each factor's roles in ER-phagy. We have revised the text to be more precise on all points.

4. The observation that ER changes in long-lived worms resemble those observed in aged wild-type worms is unexpected and seemingly contradictory to what would be anticipated. To address this, the authors should examine whether age-related ER changes under these long-lived conditions are attenuated as the animals age, despite the phenotypic similarities observed at the young age. Longitudinal analysis of ER morphology and function in these long-lived models would help clarify whether the observed changes.

We thank the reviewer for this point. We have added new data revealing that age-related changes in the ER are not attenuated in long-lived animals, but rather remain enhanced throughout adulthood in an autophagy-dependent manner. This finding is consistent with our working model that removal of ER both during normal aging and in contexts of lifespan extension is adaptive. As long-lived animals are not entirely halting the aging process, we believe it makes sense that at least some age-induced effects persist in these backgrounds.

5. *Dpy-31* RNAi partially rescues the age-related ER changes, as shown in Fig. S6A-D. Based on this result, collagen maturation may still be involved in these processes. The authors should discuss this result with precision. In addition, compared to the control, *dpy-31* RNAi decreases the SEC-61.B level in young worms, but not in aged worms. As a result, the age-related reduction of SEC-61B is less pronounced in *dpy-31* RNAi worms than in control worms. This comparison should be explicitly noted.

We thank the reviewer for bringing this point to our attention; it was not our intention to completely rule out roles for collagen homeostasis. We have added new data and reframed the discussion within the TMEM-131 results to more accurately relay our working model and how we believe collagen processing is involved. We highlight our approach's design to distinguish between models where TMEM might become a relevant trigger for ER-phagy as either a sensor for collagen quality control or client load, i.e., supply vs. demand switch. As explained in the revised manuscript, we believe our results support the latter. Regarding *dpy-31* results specifically, we have clarified the statistically significant differences in line with reviewer suggestions while more precisely making our original point, which was to contrast *dpy-31* effect sizes (small, where present) with large effects by *tmem-131* across all metrics.

6. Based on the images shown in Fig. 6I, I am not fully convinced that TMEM-131 RNAi suppresses the age-induced colocalization of ER and lysosomes to the same extent as LGG-1 RNAi. There are much more puncta that remain positive for both TRAP-1 and SCAV-3 with TMEM-131 RNAi. To clarify this result, please provide images with TRAP-1::mCherry and SCAV-3::GFP channels separated. This would help better visualize and quantify the degree of colocalization.

We thank the reviewer for their attention to detail, and we agree completely with the reviewer's observation. We include channel separated images and both imaging and western blot evidence supporting that *tmem-131* impairment indeed results in baseline elevations in autophagy/ER-phagy. We believe this observation is consistent both with past studies and our revised and clarified model. First, upregulated basal turnover of the ER in both *tmem-131* and *trpp-8* animals is consistent with their established roles in facilitating collagen export and that misfolded and/or accumulated procollagen is a major substrate for ER-phagy (PMIDs: 30287488, 30559329). Indeed our original hypothesis was that TMEM triggers ER-phagy as a sensor of misfolded procollagen, but the increased baseline ER-phagy in *tmem-131*(RNAi) conditions precluded this explanation. Notably, however, *tmem-131*(RNAi) blocked the age-induced effects on ER volume and lysosomal delivery specifically. Thus, *tmem-131* is unique among COPII and collagen maturation factors in that it prevents only the age-induced depletion of ER, indicating the triggers for ER-phagy during collagen impairment and aging are distinct.

We believe our results instead support a model where *tmem-131*'s role in facilitating ER turnover is not based on misfolded procollagen load per se, but rather on the absence of procollagen clients, which have been shown to be reduced with age at multiple levels (PMID: 38177158). Based on this and additional new data (see below), we revised our discussion of these results for clarity and propose this model, while acknowledging further dedicated structure-function studies are required.

7. It is premature to designate TMEM-131 and IRE-1 as ER-phagy receptors without further analysis, including confirming their direct binding to ATG8 in a LIR dependent manner, assessing their recruitment to autophagosomes, and validating their specificity to well-defined ER-phagy induction. The authors should consider adding more experiments to support their claims or dampening their statements.

We have substantially expanded and clarified our model for TMEM-131 action based on new data. We include new data showing that TMEM-131-GABARAP binding is partly dependent on the LIR motif, but that there are also additional LIR motifs within TMEM-131. As suggested, we show that TMEM-131 association with autophagosomes increases during starvation, including examples of internalization that are consistent with the discovery of TMEM-131 as a significant constituent of autophagosomes in HeLa cells (PMID: 37087736). Based on parallel studies and confirmed with those colleagues (PMID: 40760246), we added that the cytoplasmic domain of TMEM-131 possesses intrinsically disordered regions that share traits with canonical ER-phagy receptors that fragment organelle membranes. We have added explicit discussion of the evidence for and against TMEM-131 as an ER-phagy receptor in the revised manuscript, concluding overall that TMEM-131 is an "ER-phagy regulator" at this stage. Based on the canonical receptors, each requiring a dedicated manuscript to meet the field's mechanistic bar, we believe definitive assignment in this case will require additional biochemical structure-function, imaging and genetic analyses beyond the scope of the current manuscript, but our results combined with the promising mammalian leads highlight exciting and likely conserved new roles for TMEM-131 in ER homeostasis and ER-phagy.

We also apologize for the confusion regarding IRE-1/XBP-1: we propose this acts not as a receptor per se, but upstream of a putative receptor-mediated pathway, and have ensured the revised manuscript is clear on this point. We believe our colleague and newly added collaborator on this work, Dr. Truttmann, may have elucidated a relevant *ire-1* dependent autophagy/ER-phagy pathway in parallel work currently under review, thus we propose our studies mutually strengthen and amplify one another.

8. The evidence supporting the notion that ER-phagy-dependent turnover of ER has a positive effect on healthy aging and longevity is currently lacking. The authors should test whether TMEM-131 and IRE-1 are required for ER loss and the lifespan extension observed under different pro-longevity mechanisms.

We thank the reviewer and agree this is a key and exciting question. To address this, we performed new *C. elegans* lifespan experiments in which we combined *tmem-131* and *xbp-1* loss-of-function with mTOR/*raga-1* mutants. Our new data show that while *tmem-131* or *xbp-1(RNAi)* alone largely suppress *raga-1*-mediated lifespan extension, the *tmem-131; xbp-1(RNAi)* double mutant completely abrogates lifespan extension by *raga-1*. This demonstrates that tissue-specific ER-phagy regulators are required for the beneficial effect of mTOR inhibition on lifespan, complementing the similar results from our chronological lifespan assays in yeast.

Importantly, these findings also align with prior work showing that IRE-1/XBP-1 are genetically required for lifespan extension by multiple longevity paradigms, including dietary restriction and reduced insulin/IGF-1 signaling (PMC2906894, PMC2676694). However, the underlying cellular mechanisms that explain the genetic requirement for *ire-1/xbp-1* in these pathways have remained a longstanding, open question, often assumed to reflect vague and general improvement in proteostasis alone. Our results illuminate a surprising and significant new explanation: that ER-phagy and proactive ER remodeling may be critical processes linking IRE-1/XBP-1 activity to healthy aging and lifespan extension in potentially diverse contexts. We have added discussion of this in the revised text.

9. The association between ER loss and ER-UPR requires further investigation. Are other branches of ER-UPR, beyond IRE-1/XBP-1, also involved in mediating the ER loss in the intestine during aging?

We thank the reviewer for this question regarding the specificity of UPR mediators. We have added new imaging experiments clarifying that *atf-6* and *PERK/pek-1* mutants have no consistent effects on age-dependent ER loss in either dermis or intestine, revealing the specificity of the effect to the IRE-1/XBP-1 branch.

Are ER-UPR up-regulated in the long-lived models, considering the ER loss observed in these worms? We thank the reviewer for raising this important point. As alluded to in the previous response, prior studies have consistently shown that longevity paradigms like reduced insulin signaling or dietary restriction lower baseline ER stress marker expression but paradoxically still require IRE-1/XBP-1 for lifespan extension (PMC2906894, PMC2676694). The consensus interpretation of these results currently is that these contexts simply reduce basal ER stress and protein misfolding, but how that is achieved and why *Ire-1/Xbp-1* is required both remain unclear. Our work opens a new cellular mechanism by which this occurs: basal ER stress is reduced by physically downsizing the ER and protein flux through the organelle, and this is achieved through an *Ire-1/Xbp-1* dependent ER-phagy mechanism. We now include this connection in the discussion.

Minor points:

- 1. Line 183, Fig. 2A should be Fig. S3F.**
- 2. Line 186, Fig. S2F-G should be Fig. S3F-G**
- 3. Line 187, Fig. S2F and S2H should be S3F and S3H**
- 4. From the representative images shown in Fig. S3I, the fluorescence intensity decrease is stronger in males than in hermaphrodites. However, the quantification result in Fig. S3J shows the opposite. Please make sure to provide correct representative images.**
- 5. Line 201, it is unclear what refers to Fig. 22.**

We thank the reviewer for identifying these and have corrected all figure references. We have also checked the male imaging data and the images are correct; the confusion may be related to the fact that the intensities are normalized within groups to day 1, and male baseline intensity is lower than hermaphrodites. We have clarified this normalization in the legend.

Reviewer #2 (Remarks to the Author):

In this manuscript, Donahue, EKF., et al. reports that aging promotes significant remodeling of ER morphology, characterized by loss of ER mass and a shift in ER sheets to tubules ratio, in various tissues of *C. elegans*. Changes in ER structure is also observed in a model of yeast chronological aging and possibly in neurons of mice. The authors further demonstrate that ER remodeling in aging results from increased levels of autophagy/ERphagy, as downregulation of core components of the autophagy machinery, such as *ATG8/lgg-1*, rescues ER morphology in old worms. The authors also show that in *C. elegans* ERphagy occurs through tissue-specific mechanisms. While in the hypodermis ERphagy is partially depends on the newly identified *TMEM-131* receptor, in the intestine, it is regulated by the IRE-XBP-1 branch of the UPR.

Overall, this is a well written manuscript that presents intriguing findings. However, in several instances, the data provided is overinterpreted. Also, the generalization of the findings across tissues and species are not well supported by the data. Additional data is required to substantiate the exact changes in ER morphology across the different models.

Major comments:

1. The manuscript presents data demonstrating a significant decrease in ER mass in *C. elegans* tissues such as the hypodermis and intestine with aging. This reduction appears to affect both ER sheets and tubules. While the authors suggest a shift in ER domains as a potential adaptive response, this conclusion is not convincingly supported by the data. For instance, in the intestine, the reduction in Ret1-GFP fluorescence is even more pronounced than in SEC61B-GFP. It is essential to clarify whether aging impacts the expression levels of specific ER proteins, such as SEC61, TRAP, Ret-1, or if it in fact results in a massive loss of ER.

Although TEM images are provided for the hypodermis (Fig. 2F), it is focused on a very small area and lack clarity. The authors should include lower-magnification and more representative TEM images of both the hypodermis and intestine to better illustrate the observed ER structural changes.

Also, the authors should provide fluorescence imaging (either staining or addition endogenous tagged proteins) and western blots for additional ER proteins, such as calreticulin and GRP78, which are more uniformly distributed across ER subdomains.

We thank the reviewer for drawing out this key distinction between changes in the total mass of the ER vs. shifts in morphology and relative subdomain abundance, and we agree these are separable. In our revised manuscript, we add multiple new lines of evidence that more directly address the loss of total ER content during aging and help to clarify the interpretation of subdomain remodeling.

Specifically, we now include 1) fluorescence imaging of natively tagged GRP78 orthologs (HSP-3, HSP-4), which show clear age-related decline in signal; 2) Western blot analysis of alternative tubulating factor YOP-1, revealing a much more modest decline than observed for RET-1; 3) new EM imaging of both dermal and intestinal tissues, supporting a loss of both total ER and specifically ER sheet structures with age; and 4) fluorescence imaging of alternative ER marker *elo3* in yeast, which shows similar decline; and 5) new transcriptome- and proteome-scale data from aging mammalian atlases that confirm conservation of broadscale shifts in the aging ER proteome.

Taken together, our data strongly and consistently support our conclusion that aging leads to an overall loss of ER mass. Additionally, the differential impacts of age on distinct ER shaping factors, RET-1 vs. YOP-1, now provide a plausible mechanism by which network morphology becomes biased towards more tubular structures in aged animals, as supported by our fluorescence and EM imaging. Consistent with a working model where this more tubular network is achieved by preferential turnover of rough ER, impairment of multiple proposed ER-phagy mediators (*ULK1/unc-51*, *tmem-131*, *ire-1*) appears to result in enhancement of sheet-like structures. While we agree that age-dependent ER structural distinctions are likely much more nuanced and tissue-specific, our data strongly support the model that ER loss is a widespread feature of aging and that changes in network morphology likely reflect a combination of targeted degradation and molecular and functional remodeling processes. We have revised the text throughout to clarify this point and attempt to avoid overgeneralizing our observations.

2. In Fig. 2A, the expression of SEC61B seems to occur inside the nucleus in aged worms. While distortions in nuclear envelope shape are expected during aging, as supported by the article referenced by the authors, this fluorescence pattern is puzzling. Could the authors clarify or provide an explanation for this observation?

We thank the reviewer for highlighting the need to clarify this potentially confusing localization pattern. We agree it can appear that Sec61b enters or mislocalizes somehow in the nucleus from these 2D images, but this reflects the surprisingly deep and sometimes convoluted invaginations of the nuclear envelope (from above or below the focal plane in some cases). We are not currently aware of an fully accepted and empirically-backed mechanism by which these invaginations are forming, but dysregulation of lamins and/or cytoskeletal-nuclear interactions are likely both involved. We have revised the results to help clarify this phenomenon better.

3. Since the expression of all the endogenously tagged proteins investigated (SEC61, Ret-1, and TRAP) decreases with aging, it would be beneficial to include a cytosolic-tagged protein (or

another subcellular location) as a control to demonstrate that this is not a general adaptation in worms to degrade tagged proteins over time.

We thank the reviewer for highlighting the need to validate that age-related changes are not an artifact of fluorescence tags. First, our neurite imaging involves co-expression of an mScarlet ER membrane marker and GFP cytosolic marker, and the ER marker increases intensity in this particular anatomical compartment, while the cytosolic marker stays constant. We have also added imaging of mitochondrial outer membrane marker (TOMM-20) showing that the age-dependent declines in ER are a dynamic that is uncoupled from other compartments and potential non-specific effects of fluorescence tags.

4. The conclusion that the yeast phenotype promoted by aging is similar to the phenotype observed in *C. elegans* tissues is not well supported by the data. The study only presents the Sec61-GFP marker as evidence. It is possible that the observed changes reflect a decrease in Sec61 expression during aging rather than a reduction in overall ER levels. TEM data and additional markers are essential to substantiate the phenotype.

We thank the reviewer for raising this point. To test whether aspects of age-related ER decline are conserved in yeast, we now monitored Elo3-mCherry, a pan-ER marker that labels perinuclear and cortical ER tubules, and included supportive footprint and PAR quantitation. Consistent with our worm data, Elo3 intensity and footprint declines in the cytosolic space, while increasing in the vacuole. These results support the conclusion that ER content declines with aging at the protein level. We have attempted vEM analysis of the ER in yeast cells (preliminary results attached). While the results are promising in that they reveal an obvious reduction in darkly stained endomembrane structures in aged cells, we have not yet been able to achieve what we deem sufficient contrast for confident ER segmentation, and thus we view these data as preliminary. We have revised conclusions in the text to ensure they accurately represent the data.

Given that yeast cells have a relatively unique ultrastructural configuration of their ER, we propose that this deeper analysis may merit its own future study, while highlighting recent work that supports our concept broadly by showing that aging reconfigures ER microdomain organization in yeast cells through apparent changes in membrane thickness (PMID: 38871812).

5. In the attempt to generalize their mechanism to mammalian cells/tissues, the authors provide TEM images of motor cortical neurons. It is unclear why the authors chose to perform the experiments in neurons. The most robust data in *C. elegans* is observed in the secretory tissues such as hypodermis and intestine. It would be more logical to explore the effect of aging in tissues with a high abundance of ER sheets and secretory ER such as the liver or pancreas. Neurons exhibits a very particular and specialized ER organization, and the images provided do not clearly demonstrate dramatic changes in ER content. Generalizing this phenotype across other tissues and species based solely on these results is an overreach.

We thank the reviewer for this thoughtful critique. We agree that ER structure and function vary substantially across mammalian tissues, and that cortical neurons represent a specialized context. We initially selected this tissue due to a combination of technical and historical reasons: first, availability of high-quality aged samples that were perfusion-fixed appropriately for quantitative EM analysis, and

second, a suggestion from neuropathological studies in the 1970s that dysregulation of ER cisternae was a feature of aged rodent brains (PMIDs 4374609, 4696517). We have clarified our motivation in the revised manuscript.

To address this more fully, we have substantially expanded our cross-species analysis in the revised manuscript. Specifically, we now include transcriptomic and proteomic data from large-scale mammalian aging atlases, which show that genes and proteins involved in ER translocation, folding, and secretory flux are consistently downregulated with age in multiple tissues—including skin, liver, and kidney. We present evidence that at least in some tissues, the age-dependent decline in the ER proteome is more pronounced than that of the mitochondria, which notably is well-established as a conserved organellar ‘hallmark of aging.’ These findings support the idea that broad remodeling of the ER occurs with age across diverse tissue types in mammalian models. Coupled to our ultrastructural analysis revealing a clear decline in ER volume with age, we propose that we have extrapolated the concept that age-onset remodeling of ER is occurring in mammals, while acknowledging that a great amount of future, cell-specific work remains. As in the case of mitochondrial dynamics, there is not likely to be a universal morphological shift across every species and cell type, given the heterogeneity of organelle structure and function. Indeed, we believe that opening this avenue of research for the ER will be one of the enduring impacts of our study.

6. The proteomic analysis Fig 2I and J lacks information. Was it performed in specific tissues or whole worms. If the latter, it’s hard to relate to the particular ER phenotypes in different tissues.

We thank the reviewer for highlighting the lack of clarity and have revised the manuscript to make this clearer. Proteomics analyses in worms are whole-organism in this case and generally, due to the small size of the animals, which is one reason we have dissected tissue-level changes more carefully with imaging approaches. We believe the fact that these patterns emerge at the global level supports the relatively consistent results we see across tissues, but acknowledge that these proteomic effects are likely emphasizing the largest *C. elegans* metabolic tissues (e.g., intestine, muscle hypodermis).

7. The data demonstrating the rescue of ER mass by Atg8/Igg-1 is strong and clear. However, the rescue phenotype promoted by deletion of Tmem-131 in hypodermis is not convincing. While ER mass appears to increase, the morphology differs noticeably from that of the control. Therefore, referring to this as a "rescue" is an overstatement. Additionally, the impact of Tmem-131 downregulation on Ret-1-GFP is not addressed and should be included to provide a more comprehensive analysis.

We thank the reviewer for this insightful comment and agree we failed to capture the nuances of *tmem-131* effects on the ER. First, we have expanded and clarified our model for age-induced ER-phagy in the revised manuscript. Specifically, we propose that multiple ER-phagy routes contribute to the broadscale remodeling of the ER, and *Atg8* is a unique genetic node in its broad requirement across diverse ER-phagy pathways. This explains why *Atg8* is likely the only factor capable of full reverse of the age-induced changes, while factors such as *ULK1* (macro-ER-phagy specific) and *TMEM-131* exhibit weaker rescues. As the reviewer notes, however, the increase in ER mass following *tmem-131* knockdown is also accompanied by distinct changes in morphology compared to controls, specifically enrichment of sheet-like structures. While we note that the strength of rescue of ER loss by *tmem-131* knockdown is more robust in the context of physiological aging than in the *mTOR* (*raga-1*) mutant background in which the original screen was performed, the sheet-like structures are still somewhat apparent. This morphological shift is consistent with our working model that macro-ER-phagy plays a role in preferentially promoting degradation of rough ER sheets, consistent with both our *ULK-1* (see above response) and *TMEM-131* knockdown results. We agree that this differs from a simple reversion of the phenotype and have revised our language accordingly to avoid oversimplifying this result. Also, the requirement of *tmem-131* for ER depletion was actually discovered in *RET-1*-labeled animals, as these were the basis of our screen. We have clarified this in the revised manuscript and on the representative images.

8. As acknowledged by the authors, the changes in ER morphology, based on SEC61B-GFP,

observed by downregulation of *daf-2*, *raga-1*, *glp-1* and *ifg-1* differ significantly from the loss of ER mass reported in aged worms. Specifically, the mutants show the formation of perinuclear ER patches. These data do not add much to the paper since the authors do not explore its relevance for life span extension, ER function etc...

We appreciate the reviewer's point that we do not fully expand upon these altered ER configurations in the current manuscript. We include these data as relevant if not important support for the question of whether ER loss is beneficial or maladaptive in the aging process, but we agree the pronounced morphological change opens a new set of questions beyond our intent. We are excited about following up on these changes, but we believe their explanations are beyond the scope of the current study. We do now add data confirming a causal role for the factors that we identify as necessary for age-onset ER turnover, TMEM-131 and IRE-1/XBP-1, in mTOR/*raga-1* paradigms. We also emphasize that macroautophagy components, ULK-1/LGG-1, both strongly suppress ER remodeling and are universally required for longevity assurance across all those pathways. The challenge that we believe our study has tackled and now highlights as a key future research avenue is uncoupling the prevailing concept of 'macroautophagy as a bulk cellular recycling system' from the roles of organelle-selective autophagy and the concept that cells may be using the autophagic network to more precisely remodel organelle networks and resultant metabolic outputs. We aim to explore the mechanisms and outputs of these longevity-associated organelle configurations in subsequent studies.

9. In mammalian cells, XBP1 is well known to drive ER biogenesis specially in secretory cells such as plasma cells, acinar cells (e.g . PMID: 16362047). It's not clear to me how loss of XBP1 would restore ER mass in the intestine?

We agree this is an important and exciting question. First, we agree that IRE-1/XBP-1 axis is traditionally and accurately viewed as driving ER expansion, but it is also not unprecedented that Ire1/Xbp1 or the UPR more broadly play a role in ER turnover, including downstream activation of ER-phagy receptors like CCPG1 (e.g., PMID: 29290589). Additionally, a parallel study (in revision) by our collaborators has highlighted a pathway for autophagy and ER-phagy induction acting downstream of Ire1 in the *C. elegans* intestine (PMID: 39868301). We believe this fits conceptually and mechanistically with the established RecovER-phagy pathway, as a mechanism for cells to sense when an ER stress is resolved and restore the ER to a basal state. Future work will be required to decipher the likely bi-directional regulation between UPR activity state and ER-phagy processes during aging to decipher if age-related changes in this network arise through a vicious cycle-like mechanism or imbalances in biogenesis/degradation. We have revised and expanded discussion of the interplay with the UPR in the revised manuscript.

Minor comments

- 1. Fig. S4I, why is the fluoresce over saturated in the *Igg-1* rescue experiment- it seems the images were not acquired in the same condition.**
- 2. Line 201: Paper references Figure 22 which does not exist. Probably a typo.**
- 3. Line 221: In figure S4M, legend states "during early aging" but no further clarification is included on when this reporter accumulation occurs**
- 4. Line 303: The authors refer to figure S6H-I to demonstrate effects of TMEM131 but it actually shows XBP1 effects. Probably a mislabeling.**

We thank the reviewer for these points. We have corrected erroneous labeling and figure references and clarified the age of the animals from which the 3D projection was taken. We agree the *Igg-1* image in question may appear oversaturated, but this reflects post-acquisition increase in image brightness to ensure that the ER in all samples is visible in representative images and the actual dramatic result that *Igg-1*(RNAi) elevates RET-1 levels well above baseline in the aged animals. With this clarification, we

have left this as-is for now, but are open to modifying the image appearance if deemed important.

Reviewer #3 (Remarks to the Author):

The authors report the interesting phenomenon that the ER levels decrease during aging across species in yeast, *C. elegans*, and mouse. This study applied different approaches to indicate autophagy regulates ER clearance in the aging worm, and putative LIR containing-TMEM-131 regulates ER-phagy. The discovery of ER-phagy in *C. elegans* is novel and the physiological relevance of aging and ER health is important.

Comments to the authors:

Major:

1a. Evidence of TMEM-131 as an ER-phagy receptor is not convincing. Since the essential amino acids of putative LIR domain are conserved, evidence of whether these mutation of these sites affect ER clearance is required.

We thank the reviewer for highlighting the need for more clarity on TMEM-131's role in ER-phagy. In the revised manuscript, we now show that mutation of the conserved LIR motif diminishes binding of the TMEM-131 cytosolic domain to human GABARAP, confirming an Atg8 interaction that is at least partly LIR-dependent. Residual binding likely reflects additional motifs, as reported for RTN3L, and indeed we identified a second candidate LIR motif upon manual inspection, which is now included in the manuscript. While these data strengthen the case for TMEM-131 in ER-phagy, we agree that they fall short of the bar historically required for receptor-focused studies. For this reason, we frame TMEM-131 as a factor linking collagen biology to age-dependent ER remodeling, rather than claiming to have established a new canonical ER-phagy receptor. We also expand the discussion to include a balanced interpretation of its similarities and departures from other classical receptors.

1b. Additionally, existing study indicates TMEM-131 interacts with COPII vesicles to facilitate collagen secretion (PMID: 32095531) and COPII components are related to distinct ER clearance pathways (PMID: 31273116, 38593803). It is unclear whether TMEM-131 regulates COPII vesicle formation and whether core COPII coat proteins are required for ER clearance.

We thank the reviewer for raising this important point. TMEM-131 has indeed been linked to COPII-mediated collagen trafficking through its interaction with TRAPPC8, raising the possibility that its effects on ER loss reflect COPII vesicle formation. To address this, our candidate screen included the COPII coat proteins *sec-23*, *sec-24*, and *sec-31*. None of these knockdowns suppressed ER loss in aging or mTOR-inhibited animals, whereas *tmem-131* RNAi consistently and specifically rescued ER phenotypes (revised manuscript, new images provided). These results argue that TMEM-131's role in ER remodeling is mechanistically separable from general COPII activity. We have revised the text to explicitly discuss COPII factors and to soften language around TMEM-131 as a canonical ER-phagy receptor. As with other established receptors, we believe that unambiguous dissection of TMEM-131's roles in collagen secretion, COPII trafficking, and ER turnover will require a dedicated study. Here, we emphasize its identification as a novel ER-phagy regulator and highlight its potential to act either as a receptor itself or as part of a recruitment complex, meriting further exploration across systems.

2. More evidence about whether TMEM-131 affects bulk autophagy substrates including SQST-1 and LGG-1 is required to strengthen TMEM-131 selectively regulates ER clearance.

We thank the reviewer for identifying this important experiment. We have now performed Western blot analysis of LGG-1 as suggested to investigate *tmem-131* effects on autophagic flux. Consistent with an intact or enhanced basal autophagy, GFP::*LGG-1* processing is increased under *tmem-131* RNAi (total

and lipidated LGG-1 decrease, while free GFP accumulation rises). These results indicate that autophagy is not globally impaired during *tmem-131* depletion and support that TMEM-131's role is specific to the ER.

3. The TEM results clearly showed ER level decrease in aging worm body but lack key evidence that ER structures exist in autophagosomes. Given the significant decrease of ER during aging, ER structures should be seen enclosed by autophagosomes. If autophagosomes undergo rapid clearance in the worm body, employing genetic or chemical treatments that block lysosomal function and inhibit autophagosome degradation could increase the chances of capturing autophagosomes by TEM.

We thank the reviewer for raising this point. To directly assess the presence of ER within autophagosomes during aging, we have performed TEM analysis in worms during aging. Consistent with the high rate of ER-phagy predicted for the substantial decline in ER mass, we observed autolysosomal structures containing multilamellar membrane whorls consistent with ER-derived cargo in sections from 4/5 animals. These observations are consistent with the fluorescence imaging revealing lysosomal containment of ER, which we have also moved adjacent to the new EM panel, and our conclusion that ER-phagy contributes to age-onset ER loss in vivo.

Minor:

1. Biochemistry experiments indicate more RET-1 is degraded compared to SEC61-B (Fig. S2 I-L) that are not consistent with the observations by fluorescent analysis. The authors need to test whether TMEM-131, UNC-51, or LGG-1 affects RET-1 and SEC61-B protein levels via western blot in aging worms to strengthen that the degradation of these ER subdomain proteins is mainly dependent of autophagy. Descriptions about Fig S2 I-L in the main text are missing.

We thank the reviewer for this careful observation. We appreciate that at first glance the RET-1 and SEC-61B Western blot data may appear to contrast with our fluorescence imaging. However, we note that the blots are performed on whole-animal lysates, thus reflecting the net change in protein levels across all tissues and emphasizing larger tissues. In our imaging, we observe that RET-1 declines far more steeply than SEC-61B in the intestine, the largest tissue in the animal. Given these spatial differences, we propose that the biochemical and imaging results are in fact consistent when interpreted in the proper tissue context. Additionally, we provide new Western blot for an alternative tubule-shaping factor, YOP-1, revealing it to be more stable across age and providing an explanation for how tubules can be maintained in the face of a large loss of RET-1. Because the imaging data allow tissue-specific resolution using native gene fusions, we consider these measurements more reliable for assessing age-related dynamics of RET-1 and SEC-61B in the contexts most relevant to our study. We have revised the manuscript to clarify these interpretations and better avoid confusion.

2. A previous study indicates that IRE-1–XBP-1 mediated ER-UPR is activated in *tmem-131* mutant worms that is caused by the failure of collagen secretion (PMID: 32095531). Is IRE-1 required for TMEM-131 dependent ER clearance in hypodermis?

We thank the reviewer for highlighting this intriguing question. We find that *ire-1/xbp-1* suppresses ER loss in the intestine but not the hypodermis, whereas *tmem-131* shows the opposite specificity. Thus, IRE-1 is not required for ER clearance in the same cells that depend on TMEM-131. Nevertheless, as the reviewer notes, *ire-1/xbp-1* responses are activated as a compensatory mechanism when *tmem-131* is mutated and collagen export is reduced, likely producing wide-ranging outcomes including chaperone expansion and enhanced basal ER turnover, as we show in the revised manuscript. To extend this point, we have added new data showing that combined loss of *tmem-131* and *xbp-1* produces synthetic lifespan effects, indicating that the two pathways provide parallel protective functions. Overall, we agree that dissecting the interplay between *tmem-131* and *ire-1* (and more broadly between ER biogenesis and ER turnover) is an important next step. However, because *ire-1/xbp-1* has broad pleiotropic roles, we believe

that interpreting these interactions will require identification of ER-phagy–specific mediators downstream of *ire-1/xbp-1*. As this lies beyond the focus of the current study, we propose it is best addressed in future work.

3. Line 303, 307, 314 “Fig. S6M-Q” are not related to the figure.

We thank the reviewer for catching this mistake and have corrected figure references.

We thank all reviewers for their thoughtful and constructive feedback. Guided by these comments, we believe the revised study more thoroughly establishes ER remodeling as an evolutionarily conserved and mechanistic aspect of the aging process. First, we have deepened our characterization of this phenomenon across worm, mammalian and yeast models through additional fluorescence imaging, immunoblotting, EM/ultrastructural, and transcriptomic/proteomic analyses according to the strengths/limitations of each system. Similarly to the 'hallmark' age-associated dysregulation of mitochondria, we suggest that we should not expect there to be a hard, universal rule for how these dynamics change with age across all cell types. However, our studies reveal surprisingly consistent themes across models and tissues, namely a robust decline in ER mass, particularly rough ER, consistent with a compositional shift from proteostasis to lipid-metabolic roles. Next, we have improved mechanistic clarity and precision. We have reframed and better aligned our interpretations with the current state of the multi-faceted ER-phagy field, specifically that ER remodeling is driven by multiple ER-phagy pathways. Our data suggest macro-ER-phagy makes the largest contributions, but with additional roles for ERLAD and RecovER-phagy-like mechanisms. We add data supporting the conclusion that ER remodeling is an adaptive process, actively promoted by diverse life-extending pathways, and provide more context by discussing how this aligns with conserved autophagy and translational dynamics during aging. We also now show that two distinct suppressors of age- and mTOR-dependent ER remodeling identified through our prior genetic screen, TMEM-131 and the IRE-1/XBP-1, are required for lifespan extension during mTOR impairment in *C. elegans*, providing causal links. Finally, we provide several new experimental insights into the mechanistic roles of these suppressors, showing that TMEM-131 acts distinctly from its partners in collagen maturation and general ER secretion, and thus is likely a novel ER-phagy mediator, while IRE-1/XBP-1 plays a unique role among UPR branches in terms of promoting ER clearance in aging contexts. We hope that, collectively, the new data and conceptual reframing has improved the precision, clarity, and balance of our conclusions that ER-phagy based ER remodeling is a fundamental and important new component of our models for understanding aging cells.

Reviewers' comments:

Reviewer #1 (Remarks to the Author):

In this manuscript, the authors report age-related ER changes in *C. elegans*. Using fluorescent proteins tagged with tubular and rough ER markers, they revealed the tissue-specific distribution preferences of ER subdomains in adult worms. The authors demonstrated that levels of ER proteins SEC-61.B and RET-1 dramatically decrease beginning at day 3 and continue through day 7. Concurrently, the ER network in the hypodermis shifts from densely packed rough ER sheets to sparsely distributed tubules. Importantly, these age-related changes are observed in both males and hermaphrodites. The authors further analyzed other tissues, observing differential results for SEC-61.B and RET-1. Interestingly, in yeast cells, they also observed an age-related decrease in SEC61 levels, while in mouse cortical neurons, an age-related reduction in ER tubules was detected. Mechanistically, they found that autophagy-mediated ER degradation drives the age-related loss of ER mass and altered morphology. Interestingly, in several long-lived *C. elegans* models, young animals displayed ER changes resembling those observed in aged wild-type worms, which is somewhat counterintuitive. Additionally, in yeast, deletion of ER-phagy-specific factors partially suppressed lifespan extension induced by rapamycin treatment. Lastly, the authors identified two factors, TMEM-131 and IRE-1, as key mediators of ER remodeling in the worm hypodermis and intestine, respectively.

Overall, the manuscript is well-written, reporting interesting ER phenotypic changes associated with aging and uncovering new factors that mediate these changes in a tissue-specific manner. However, these are several places where the authors' claims appear overstated based on the current data. Additionally, and the relationship between the observed ER changes and aging/healthy aging/longevity remains inconclusive. These concerns need to be addressed before

the manuscript can be considered for publication.

Major points:

1a. A broader analysis of additional ER-resident proteins would strengthen the claim of a systemic decline in ER protein levels with age.

We agree that examining a broader set of ER-resident proteins is useful to support the generality of age-related ER decline. In the revised manuscript, we now include multiple additional lines of evidence demonstrating that ER loss is not limited to the originally reported markers (SEC-61.B and RET-1). Specifically, we added: 1) Fluorescence imaging of natively tagged HSP-3 and HSP-4, the *C. elegans* orthologs of the mammalian GRP78/BiP chaperone, both of which show robust age-dependent signal loss (ED Fig. 3A-D); 2) Western blot analysis of YOP-1 (ED Fig. 2M-N), a second curvature-promoting ER-shaping protein in addition to RET-1, which shows a more modest but detectable decline with age; 3) A complementary ER marker (Elo3::mCherry) in yeast (Fig. 4P-S), which also shows age-related decline in abundance; 4) focused analysis of mammalian aging transcriptomic and proteomic atlases, resulting in new data showing consistent and broadscale downregulation of ER proteostasis/secretory components in multiple aged mouse tissues and an upregulation of ER proteins with functions in lipid metabolism, etc. (Fig. 4A-I). Together, these data provide strong support for a systemic and conserved decline in ER protein abundance during aging, combined with a reconfiguration of protein/functional composition. We have revised the text and figure panels accordingly to reflect these findings.

1b. Furthermore, the observed reduction in rough ER sheets raises questions about its impact on protein translation. With the decrease of rough ER sheets, the translational level is expected to be decreased.

We thank the reviewer for highlighting this important connection, which we did not make clear in the original manuscript. Declines in global protein synthesis and translational machinery are established effects during both aging and in the context of multiple longevity paradigms (PMIDs 30733602, 30562164), and we also include data showing that impairing translation initiation (*ifg-1* RNAi) is sufficient to induce the longevity-associated ER remodeling phenotype (Fig. 6E-H). Rather than replicate the findings from past studies regarding protein synthesis rates in age, we have cited past work and included more explicit focus on incorporating this point into our model in the Introduction and Discussion.

2. The use of two ER proteins SEC-61.B and RET-1 to quantify age-related ER changes reveals tissue-specific difference, which are distinct from the findings in the hypodermis. Without further evidence like TEM, the authors should be careful about their claim that “Collectively, the shift towards more tubular morphologies observed across cell types..”. In addition, to support the conservation of age-onset ER remodeling across species, the authors should quantify ER footprint/PAR in yeast cells as well as SEC61 levels in mouse neurons.

We thank the reviewer for highlighting the need to more precisely discuss and support the nuanced extrapolation of our observations across both tissues and species. We have provided a new figure of additional EM of aging worms, which supports both an overall loss of ER and a shift from stacked sheet morphologies in aging animals in both the hypodermis and intestine (ED Fig. 4A-F). Additionally, we provide western blots of YOP-1, an alternative tubule-promoting protein which is much more robust to the effects of age than RET-1 (ED Fig. 2M-N). This result reveals that the relative composition of specific ER shaping proteins is dynamic across age and provides an explanation for how tubules can be maintained at a higher ratio in some tissues even despite a more pronounced loss of RET-1. However, given the heterogeneity of ER structures and functions across tissue types, we agree with the reviewer that many aspects of ER dynamics are likely to be tissue-specific, and thus we have added this point explicitly in the text while softening claims about cells in which we have not verified closely.

To support conservation across species, we tested an additional ER protein in yeast while adding footprint and PAR measurements (Fig. 4P-S); and we mined the *Tabula Muris Senis* scRNA-seq atlas, revealing that “Protein Processing in the ER” gene set is part of a universal (cross-tissue) aging signature

that declines with age across tissues (Fig. 4A). The newly added mammalian proteomics data correspond well with our *C. elegans* data by demonstrating consistent downregulation of rough ER/protein translocation machinery in skin, liver, and kidney (Fig. 4B-I). These multiple levels of evidence reinforce that the cross-tissue, cross-species pattern we observe for the ER is surprisingly robust even in the face of widely varied structural/functional enrichment in distinct cell types at baseline.

Taken together, these new data demonstrate that while aspects of ER remodeling are likely to vary across tissues and contexts — as is true for any complex and dynamic cellular structure — our work establishes the fundamental principle that ER morphology and its functional composition change significantly with age, opening the door to many future investigations of the consequences and mechanisms of these dynamics in diverse systems and contexts.

3. Upon *unc-51* RNAi knockdown, the ER morphology based on SEC-61.B and RET-1 markers in the hypodermis looks very different, particularly in the RET-1 images (Fig. S4A). The ER network appears highly fused. The authors should note this dramatic morphological change and provide an explanation for this observation. This difference raises questions about the other impact of *unc-51* knockdown on ER integrity. The authors' claim that "knockdown of either *Atg1/unc-51* or *Atg8/lgg-1* suppressed virtually all age-onset changes" should be revised. Similarly, *lgg-1* RNAi dramatically affects RET-1 levels in the intestine (Fig. S4I). The authors should also acknowledge this finding and ensure a precise interpretation of the data to provide accurate representation of the results.

We thank the reviewer raising the need to better highlight these differences, and we have revised the text to explain these key details. First, we contextualize the distinction between ULK1 and *lgg-1* effects within established ER-phagy pathways and their differential reliance on these factors. Specifically, only macro-ER-phagy is dependent upon ULK-1, while *lgg-1/Atg8* is more broadly required across most ER-phagy pathways, including both macro-ER-phagy and vesicular/ERLAD, as well as certain forms of micro-ER-phagy (e.g., PMID 33765438). Based on this, we propose that sheet-expansion during UNC-51(RNAi) is consistent with macro-ER-phagy playing a disproportionate role in the clearance of rough ER sheets, while *Atg8/lgg-1*(RNAi) results in complete suppression of ER loss with little overt morphology bias. The stronger overall effects of *lgg-1* and accumulation of RET-1 in the intestine under *lgg-1* RNAi is also consistent with this broader role, and all results are consistent with the relative breadth of each factor's roles in ER-phagy. We have revised the text to be more precise on all points.

4. The observation that ER changes in long-lived worms resemble those observed in aged wild-type worms is unexpected and seemingly contradictory to what would be anticipated. To address this, the authors should examine whether age-related ER changes under these long-lived conditions are attenuated as the animals age, despite the phenotypic similarities observed at the young age. Longitudinal analysis of ER morphology and function in these long-lived models would help clarify whether the observed changes.

We thank the reviewer for this point. We have added new data revealing that age-related changes in the ER are not attenuated in long-lived animals (ED Fig. 7A-H), but rather remain enhanced throughout adulthood in an autophagy-dependent manner. This finding is consistent with our working model that removal of ER both during normal aging and in contexts of lifespan extension is adaptive. As long-lived animals are not entirely halting the aging process, we believe it makes sense that at least some age-induced effects persist in these backgrounds.

5. *Dpy-31* RNAi partially rescues the age-related ER changes, as shown in Fig. S6A-D. Based on this result, collagen maturation may still be involved in these processes. The authors should discuss this result with precision. In addition, compared to the control, *dpy-31* RNAi decreases the SEC-61.B level in young worms, but not in aged worms. As a result, the age-related reduction of SEC-61B is less pronounced in *dpy-31* RNAi worms than in control worms. This comparison should be explicitly noted.

We thank the reviewer for bringing this point to our attention; it was not our intention to completely rule out roles for collagen homeostasis. We have added new data and reframed the discussion within the TMEM-131 results to more accurately relay our working model and how we believe collagen processing is involved (ED Fig. 8). We highlight our approach's design to distinguish between models where TMEM might become a relevant trigger for ER-phagy as either a sensor for collagen quality control or client load, i.e., supply vs. demand switch. As explained in the revised manuscript, we believe our results support the latter. Regarding dpy-31 results specifically, we have clarified the statistically significant differences in line with reviewer suggestions while more precisely making our original point, which was to contrast dpy-31 effect sizes (small, where present) with large effects by tmem-131 across all metrics.

6. Based on the images shown in Fig. 6I, I am not fully convinced that TMEM-131 RNAi suppresses the age-induced colocalization of ER and lysosomes to the same extent as LGG-1 RNAi. There are much more puncta that remain positive for both TRAP-1 and SCAV-3 with TMEM-131 RNAi. To clarify this result, please provide images with TRAP-1::mCherry and SCAV-3::GFP channels separated. This would help better visualize and quantify the degree of colocalization.

We thank the reviewer for their attention to detail, and we agree completely with the reviewer's observation. We include channel separated images and both imaging and western blot evidence supporting that tmem-131 impairment indeed results in baseline elevations in autophagy/ER-phagy (Fig. 7F-G; ED Fig. 8I-L). We believe this observation is consistent both with past studies and our revised and clarified model. First, upregulated basal turnover of the ER in both tmem-131 and trpp-8 animals is consistent with their established roles in facilitating collagen export and that misfolded and/or accumulated procollagen is a major substrate for ER-phagy (PMIDs: 30287488, 30559329). Indeed our original hypothesis was that TMEM triggers ER-phagy as a sensor of misfolded procollagen, but the increased baseline ER-phagy in tmem-131(RNAi) conditions precluded this explanation. Notably, however, tmem-131(RNAi) blocked the age-induced effects on ER volume and lysosomal delivery specifically. Thus, tmem-131 is unique among COPII and collagen maturation factors in that it prevents only the age-induced depletion of ER, indicating the triggers for ER-phagy during collagen impairment and aging are distinct.

We believe our results instead support a model where tmem-131's role in facilitating ER turnover is not based on misfolded procollagen load per se, but rather on the absence of procollagen clients, which have been shown to be reduced with age at multiple levels (PMID: 38177158). Based on this and additional new data (see below), we revised our discussion of these results for clarity and propose this model, while acknowledging further dedicated structure-function studies are required.

7. It is premature to designate TMEM-131 and IRE-1 as ER-phagy receptors without further analysis, including confirming their direct binding to ATG8 in a LIR dependent manner, assessing their recruitment to autophagosomes, and validating their specificity to well-defined ER-phagy induction. The authors should consider adding more experiments to support their claims or dampening their statements.

We have substantially expanded and clarified our model for TMEM-131 action based on new data. We include new data showing that TMEM-131-GABARAP binding is significantly dependent on the LIR motif (ED Fig. 8M-N), but that there are also additional LIR motifs within TMEM-131 (Results). As suggested, we show that TMEM-131 association with autophagosomes increases during starvation (ED Fig. 8O-Q), including examples of internalization that are consistent with the discovery of TMEM-131 as a significant constituent of autophagosomes in HeLa cells (PMID: 37087736). Based on parallel studies and confirmed with those colleagues (PMID: 40760246), we added that the cytoplasmic domain of TMEM-131 possesses intrinsically disordered regions that share traits with canonical ER-phagy receptors that fragment organelle membranes. We have added explicit discussion of the evidence for and against TMEM-131 as an ER-phagy receptor in the revised manuscript, concluding overall that TMEM-131 is an "ER-phagy regulator" at this stage. Based on the canonical receptors, each requiring a dedicated manuscript to meet the field's mechanistic bar, we believe definitive assignment in this case will require additional biochemical structure-function, imaging and genetic analyses beyond the scope of the current manuscript, but our results

combined with the promising mammalian leads highlight exciting and likely conserved new roles for TMEM-131 in ER homeostasis and ER-phagy.

We also apologize for the confusion regarding IRE-1/XBP-1: we propose this acts not as a receptor per se, but upstream of a putative receptor-mediated pathway, and have ensured the revised manuscript is clear on this point. We believe our colleague and newly added collaborator on this work, Dr. Truttmann, may have elucidated a relevant ire-1 dependent autophagy/ER-phagy pathway in parallel work currently under review, thus we propose our studies mutually strengthen and amplify one another.

8. The evidence supporting the notion that ER-phagy-dependent turnover of ER has a positive effect on healthy aging and longevity is currently lacking. The authors should test whether TMEM-131 and IRE-1 are required for ER loss and the lifespan extension observed under different pro-longevity mechanisms.

We thank the reviewer and agree this is a key and exciting question. To address this, we performed new *C. elegans* lifespan experiments in which we combined *tmem-131* and *xbp-1* loss-of-function with mTOR/*raga-1* mutants (Fig. 8F-I). Our new data show that while *tmem-131* or *xbp-1(RNAi)* alone largely suppress *raga-1*-mediated lifespan extension, the *tmem-131; xbp-1(RNAi)* double mutant completely abrogates lifespan extension by *raga-1*. This demonstrates that tissue-specific ER-phagy regulators are required for the beneficial effect of mTOR inhibition on lifespan, complementing the similar results from our chronological lifespan assays in yeast.

Importantly, these findings also align with prior work showing that IRE-1/XBP-1 are genetically required for lifespan extension by multiple longevity paradigms, including dietary restriction and reduced insulin/IGF-1 signaling (PMC2906894, PMC2676694). However, the underlying cellular mechanisms that explain the genetic requirement for ire-1/xbp-1 in these pathways have remained a longstanding, open question, often assumed to reflect vague and general improvement in proteostasis alone. Our results illuminate a surprising and significant new explanation: that ER-phagy and proactive ER remodeling may be critical processes linking IRE-1/XBP-1 activity to healthy aging and lifespan extension in potentially diverse contexts. We have added discussion of this in the revised text.

9. The association between ER loss and ER-UPR requires further investigation. Are other branches of ER-UPR, beyond IRE-1/XBP-1, also involved in mediating the ER loss in the intestine during aging?

We thank the reviewer for this question regarding the specificity of UPR mediators. We have added new imaging experiments clarifying that *atf-6* and *PERK/pek-1* mutants have no consistent effects on age-dependent ER loss in either dermis or intestine, revealing the specificity of the effect to the IRE-1/XBP-1 branch (ED Fig. 10).

Are ER-UPR up-regulated in the long-lived models, considering the ER loss observed in these worms? We thank the reviewer for raising this important point. As alluded to in the previous response, prior studies have consistently shown that longevity paradigms like reduced insulin signaling or dietary restriction lower baseline ER stress marker expression but paradoxically still require IRE-1/XBP-1 for lifespan extension (PMC2906894, PMC2676694). The consensus interpretation of these results currently is that these contexts simply reduce basal ER stress and protein misfolding, but how that is achieved and why Ire-1/Xbp-1 is required both remain unclear. Our work opens a new cellular mechanism by which this occurs: basal ER stress is reduced by physically downsizing the ER and protein flux through the organelle, and this is achieved through an Ire-1/Xbp-1 dependent ER-phagy mechanism. We now include this connection in the discussion.

Minor points:

1. Line 183, Fig. 2A should be Fig. S3F.
2. Line 186, Fig. S2F-G should be Fig. S3F-G
3. Line 187, Fig. S2F and S2H should be S3F and S3H
4. From the representative images shown in Fig. S3I, the fluorescence intensity decrease is

stronger in males than in hermaphrodites. However, the quantification result in Fig. S3J shows the opposite. Please make sure to provide correct representative images.

5. Line 201, it is unclear what refers to Fig. 22.

We thank the reviewer for identifying these and have corrected all figure references. We have also checked the male imaging data and the images are correct; the confusion may be related to the fact that the intensities are normalized within groups to day 1, and male baseline intensity is lower than hermaphrodites. We have clarified this normalization in the legend.

Reviewer #2 (Remarks to the Author):

In this manuscript, Donahue, EKF., et al. reports that aging promotes significant remodeling of ER morphology, characterized by loss of ER mass and a shift in ER sheets to tubules ratio, in various tissues of *C. elegans*. Changes in ER structure is also observed in a model of yeast chronological aging and possibly in neurons of mice. The authors further demonstrate that ER remodeling in aging results from increased levels of autophagy/ERphagy, as downregulation of core components of the autophagy machinery, such as ATG8/lgg-1, rescues ER morphology in old worms. The authors also show that in *C. elegans* ERphagy occurs through tissue-specific mechanisms. While in the hypodermis ERphagy is partially depends on the newly identified TMEM-131 receptor, in the intestine, it is regulated by the IRE-XBP-1 branch of the UPR.

Overall, this is a well written manuscript that presents intriguing findings. However, in several instances, the data provided is overinterpreted. Also, the generalization of the findings across tissues and species are not well supported by the data. Additional data is required to substantiate the exact changes in ER morphology across the different models.

Major comments:

1. The manuscript presents data demonstrating a significant decrease in ER mass in *C. elegans* tissues such as the hypodermis and intestine with aging. This reduction appears to affect both ER sheets and tubules. While the authors suggest a shift in ER domains as a potential adaptive response, this conclusion is not convincingly supported by the data. For instance, in the intestine, the reduction in Ret1-GFP fluorescence is even more pronounced than in SEC61B-GFP. It is essential to clarify whether aging impacts the expression levels of specific ER proteins, such as SEC61, TRAP, Ret-1, or if it in fact results in a massive loss of ER.

Although TEM images are provided for the hypodermis (Fig. 2F), it is focused on a very small area and lack clarity. The authors should include lower-magnification and more representative TEM images of both the hypodermis and intestine to better illustrate the observed ER structural changes.

Also, the authors should provide fluorescence imaging (either staining or addition endogenous tagged proteins) and western blots for additional ER proteins, such as calreticulin and GRP78, which are more uniformly distributed across ER subdomains.

We thank the reviewer for drawing out this key distinction between changes in the total mass of the ER vs. shifts in morphology and relative subdomain abundance, and we agree these are separable. In our revised manuscript, we add multiple new lines of evidence that more directly address the loss of total ER content during aging and help to clarify the interpretation of subdomain remodeling.

Specifically, we now include 1) fluorescence imaging of natively tagged GRP78 orthologs (HSP-3, HSP-4), which show clear age-related decline in signal (ED Fig. 3A-D); 2) Western blot analysis of alternative tubulating factor YOP-1, revealing a much more modest decline than observed for RET-1 (ED Fig 2M-N); 3) new EM imaging of both dermal and intestinal tissues, supporting a loss of both total ER and specifically ER sheet structures with age (ED Fig. 4); and 4) fluorescence imaging of alternative ER marker *elo3* in yeast, which shows similar decline (Fig. 4P-S); and 5) new transcriptome- and proteome-

scale data from aging mammalian atlases that confirm conservation of broadscale shifts in the aging ER proteome (Fig. 4A-I).

Taken together, our data strongly and consistently support our conclusion that aging leads to an overall loss of ER mass. Additionally, the differential impacts of age on distinct ER shaping factors, RET-1 vs. YOP-1, now provide a plausible mechanism by which network morphology becomes biased towards more tubular structures in aged animals, as supported by our fluorescence and EM imaging. Consistent with a working model where this more tubular network is achieved by preferential turnover of rough ER, impairment of multiple proposed ER-phagy mediators (ULK1/unc-51, tmem-131, ire-1) appears to result in enhancement of sheet-like structures. While we agree that age-dependent ER structural distinctions are likely much more nuanced and tissue-specific, our data strongly support the model that ER loss is a widespread feature of aging and that changes in network morphology likely reflect a combination of targeted degradation and molecular and functional remodeling processes. We have revised the text throughout to clarify this point and attempt to avoid overgeneralizing our observations.

2. In Fig. 2A, the expression of SEC61B seems to occur inside the nucleus in aged worms. While distortions in nuclear envelope shape are expected during aging, as supported by the article referenced by the authors, this fluorescence pattern is puzzling. Could the authors clarify or provide an explanation for this observation?

We thank the reviewer for highlighting the need to clarify this potentially confusing localization pattern. We agree it can appear that Sec61b enters or mislocalizes somehow in the nucleus from these 2D images, but this reflects the surprisingly deep and sometimes convoluted invaginations of the nuclear envelope (from above or below the focal plane in some cases). We are not currently aware of an fully accepted and empirically-backed mechanism by which these invaginations are forming, but dysregulation of lamins and/or cytoskeletal-nuclear interactions are likely both involved. We have revised the results to help clarify this phenomenon better.

3. Since the expression of all the endogenously tagged proteins investigated (SEC61, Ret-1, and TRAP) decreases with aging, it would be beneficial to include a cytosolic-tagged protein (or another subcellular location) as a control to demonstrate that this is not a general adaptation in worms to degrade tagged proteins over time.

We thank the reviewer for highlighting the need to validate that age-related changes are not an artifact of fluorescence tags. First, our neurite imaging involves co-expression of an mScarlet ER membrane marker and GFP cytosolic marker, and the ER marker increases intensity in this particular anatomical compartment, while the cytosolic marker stays constant. We have also added imaging of mitochondrial outer membrane marker (TOMM-20) showing that the age-dependent declines in ER are a dynamic that is uncoupled from other compartments and potential non-specific effects of fluorescence tags (ED Fig. 3E-F).

4. The conclusion that the yeast phenotype promoted by aging is similar to the phenotype observed in *C. elegans* tissues is not well supported by the data. The study only presents the Sec61-GFP marker as evidence. It is possible that the observed changes reflect a decrease in Sec61 expression during aging rather than a reduction in overall ER levels. TEM data and additional markers are essential to substantiate the phenotype.

We thank the reviewer for raising this point. To test whether aspects of age-related ER decline are conserved in yeast, we now monitored

Elo3-mCherry, a pan-ER marker that labels perinuclear and cortical ER tubules, and included supportive footprint and PAR quantitation (Fig. 4L-S). Consistent with our worm data, Elo3 intensity and footprint declines in the cytosolic space, while increasing in the vacuole. These results support the conclusion that ER content declines with aging at the protein level. We have attempted vEM analysis of the ER in yeast cells (preliminary results attached). While the results are promising in that they reveal an obvious reduction in darkly stained endomembrane structures in aged cells, we have not yet been able to achieve what we deem sufficient contrast for confident ER segmentation, and thus we view these data as preliminary. We have revised conclusions in the text to ensure they accurately represent the data. Given that yeast cells have a relatively unique ultrastructural configuration of their ER, we propose that this deeper analysis may merit its own future study, while highlighting recent work that supports our concept broadly by showing that aging reconfigures ER microdomain organization in yeast cells through apparent changes in membrane thickness (PMID: 38871812).

5. In the attempt to generalize their mechanism to mammalian cells/tissues, the authors provide TEM images of motor cortical neurons. It is unclear why the authors chose to perform the experiments in neurons. The most robust data in *C. elegans* is observed in the secretory tissues such as hypodermis and intestine. It would be more logical to explore the effect of aging in tissues with a high abundance of ER sheets and secretory ER such as the liver or pancreas. Neurons exhibits a very particular and specialized ER organization, and the images provided do not clearly demonstrate dramatic changes in ER content. Generalizing this phenotype across other tissues and species based solely on these results is an overreach.

We thank the reviewer for this thoughtful critique. We agree that ER structure and function vary substantially across mammalian tissues, and that cortical neurons represent a specialized context. We initially selected this tissue due to a combination of technical and historical reasons: first, availability of high-quality aged samples that were perfusion-fixed appropriately for quantitative EM analysis, and second, a suggestion from neuropathological studies in the 1970s that dysregulation of ER cisternae was a feature of aged rodent brains (PMIDs 4374609, 4696517). We have clarified our motivation in the revised manuscript.

To address this more fully, we have substantially expanded our cross-species analysis in the revised manuscript (Fig. 4A-I). Specifically, we now include transcriptomic and proteomic data from large-scale mammalian aging atlases, which show that genes and proteins involved in ER translocation, folding, and secretory flux are consistently downregulated with age in multiple tissues—including skin, liver, and kidney. We present evidence that at least in some tissues, the age-dependent decline in the ER proteome is more pronounced than that of the mitochondria, which notably is well-established as a conserved

organellar 'hallmark of aging.' These findings support the idea that broad remodeling of the ER occurs with age across diverse tissue types in mammalian models. Coupled to our ultrastructural analysis revealing a clear decline in ER volume with age, we propose that we have extrapolated the concept that age-onset remodeling of ER is occurring in mammals, while acknowledging that a great amount of future, cell-specific work remains. As in the case of mitochondrial dynamics, there is not likely to be a universal morphological shift across every species and cell type, given the heterogeneity of organelle structure and function. Indeed, we believe that opening this avenue of research for the ER will be one of the enduring impacts of our study.

6. The proteomic analysis Fig 2I and J lacks information. Was it performed in specific tissues or whole worms. If the latter, it's hard to relate to the particular ER phenotypes in different tissues.

We thank the reviewer for highlighting the lack of clarity and have revised the manuscript to make this clearer. Proteomics analyses in worms are whole-organism in this case and generally, due to the small size of the animals, which is one reason we have dissected tissue-level changes more carefully with imaging approaches. We believe the fact that these patterns emerge at the global level supports the relatively consistent results we see across tissues, but acknowledge that these proteomic effects are likely emphasizing the largest *C. elegans* metabolic tissues (e.g., intestine, muscle hypodermis).

7. The data demonstrating the rescue of ER mass by Atg8/Igg-1 is strong and clear. However, the rescue phenotype promoted by deletion of Tmem-131 in hypodermis is not convincing. While ER mass appears to increase, the morphology differs noticeably from that of the control. Therefore, referring to this as a "rescue" is an overstatement. Additionally, the impact of Tmem-131 downregulation on Ret-1-GFP is not addressed and should be included to provide a more comprehensive analysis.

We thank the reviewer for this insightful comment and agree we failed to capture the nuances of *tmem-131* effects on the ER. First, we have expanded and clarified our model for age-induced ER-phagy in the revised manuscript. Specifically, we propose that multiple ER-phagy routes (e.g., macro-ER-phagy, ERLAD, RecovER-phagy) contribute to the broadscale remodeling of the ER, and *Atg8* is a unique genetic node in its broad requirement across diverse ER-phagy pathways (PMID 33765438). This explains why *Atg8* is likely the only factor capable of full reverse of the age-induced changes, while factors such as *ULK1* (macro-ER-phagy specific) and *TMEM-131* exhibit weaker rescues. As the reviewer notes, however, the increase in ER mass following *tmem-131* knockdown is also accompanied by distinct changes in morphology compared to controls, specifically enrichment of sheet-like structures. While we note that the strength of rescue of ER loss by *tmem-131* knockdown is more robust in the context of physiological aging than in the *mTOR* (*raga-1*) mutant background in which the original screen was performed (Fig. 7A vs. B), the sheet-like structures are still somewhat apparent. This morphological shift is consistent with our working model that macro-ER-phagy plays a role in preferentially promoting degradation of rough ER sheets, consistent with both our *ULK-1* (see above response) and *TMEM-131* knockdown results. We agree that this differs from a simple reversion of the phenotype and have revised our language accordingly to avoid oversimplifying this result. Also, the requirement of *tmem-131* for ER depletion was actually discovered in *RET-1*-labeled animals (Fig. 7A), as these were the basis of our screen. We have clarified this in the revised manuscript and on the representative images.

8. As acknowledged by the authors, the changes in ER morphology, based on SEC61B-GFP, observed by downregulation of *daf-2*, *raga-1*, *glp-1* and *ifg-1* differ significantly from the loss of ER mass reported in aged worms. Specifically, the mutants show the formation of perinuclear ER patches. These data do not add much to the paper since the authors do not explore its relevance for life span extension, ER function etc...

We appreciate the reviewer's point that we do not fully expand upon these altered ER configurations in the current manuscript. We include these data as relevant if not important support for the question of whether ER loss is beneficial or maladaptive in the aging process, but we agree the pronounced

morphological change opens a new set of questions beyond our intent. We are excited about following up on these changes, but we believe their explanations are beyond the scope of the current study. We do now add data confirming a causal role for the factors that we identify as necessary for age-onset ER turnover, TMEM-131 and IRE-1/XBP-1, in mTOR/*raga-1* paradigms. We also emphasize that macroautophagy components, ULK-1/LGG-1, both strongly suppress ER remodeling and are universally required for longevity assurance across all those pathways. The challenge that we believe our study has tackled and now highlights as a key future research avenue is uncoupling the prevailing concept of 'macroautophagy as a bulk cellular recycling system' from the roles of organelle-selective autophagy and the concept that cells may be using the autophagic network to more precisely remodel organelle networks and resultant metabolic outputs. We aim to explore the mechanisms and outputs of these longevity-associated organelle configurations in subsequent studies.

9. In mammalian cells, XBP1 is well known to drive ER biogenesis specially in secretory cells such as plasma cells, acinar cells (e.g . PMID: 16362047). It's not clear to me how loss of XBP1 would restore ER mass in the intestine?

We agree this is an important and exciting question. First, we agree that IRE-1/XBP-1 axis is traditionally and accurately viewed as driving ER expansion, but it is also not unprecedented that Ire1/Xbp1 or the UPR more broadly play a role in ER turnover, including downstream activation of ER-phagy receptors like CCPG1 (e.g., PMID: 29290589). Additionally, a parallel study (in revision) by our collaborators has highlighted a pathway for autophagy and ER-phagy induction acting downstream of Ire1 in the *C. elegans* intestine (PMID: 39868301). We believe this fits conceptually and mechanistically with the established RecovER-phagy pathway, as a mechanism for cells to sense when an ER stress is resolved and restore the ER to a basal state. Future work will be required to decipher the likely bi-directional regulation between UPR activity state and ER-phagy processes during aging to decipher if age-related changes in this network arise through a vicious cycle-like mechanism or imbalances in biogenesis/degradation. We have revised and expanded discussion of the interplay with the UPR in the revised manuscript.

Minor comments

- 1. Fig. S4I, why is the fluoresce over saturated in the lgg-1 rescue experiment- it seems the images were not acquired in the same condition.**
- 2. Line 201: Paper references Figure 22 which does not exist. Probably a typo.**
- 3. Line 221: In figure S4M, legend states "during early aging" but no further clarification is included on when this reporter accumulation occurs**
- 4. Line 303: The authors refer to figure S6H-I to demonstrate effects of TMEM131 but it actually shows XBP1 effects. Probably a mislabeling.**

We thank the reviewer for these points. We have corrected erroneous labeling and figure references and clarified the age of the animals from which the 3D projection was taken. We agree the lgg-1 image in question may appear oversaturated, but this reflects post-acquisition increase in image brightness to ensure that the ER in all samples is visible in representative images and the actual dramatic result that lgg-1(RNAi) elevates RET-1 levels well above baseline in the aged animals. With this clarification, we have left this as-is for now, but are open to modifying the image appearance if deemed important.

Reviewer #3 (Remarks to the Author):

The authors report the interesting phenomenon that the ER levels decrease during aging across species in yeast, *C. elegans*, and mouse. This study applied different approaches to indicate autophagy regulates ER clearance in the aging worm, and putative LIR containing-TMEM-131

regulates ER-phagy. The discovery of ER-phagy in *C. elegans* is novel and the physiological relevance of aging and ER health is important.

Comments to the authors:

Major:

1a. Evidence of TMEM-131 as an ER-phagy receptor is not convincing. Since the essential amino acids of putative LIR domain are conserved, evidence of whether these mutation of these sites affect ER clearance is required.

We thank the reviewer for highlighting the need for more clarity on TMEM-131's role in ER-phagy. In the revised manuscript, we now show that mutation of the conserved LIR motif diminishes binding of the TMEM-131 cytosolic domain to human GABARAP, confirming an Atg8 interaction that is at least partly LIR-dependent (ED Fig. 8M-N). Residual binding likely reflects additional motifs, as reported for RTN3L, and indeed we identified a second candidate LIR motif upon manual inspection, which is now included in the manuscript. While these data strengthen the case for TMEM-131 in ER-phagy, we agree that they fall short of the bar historically required for receptor-focused studies. For this reason, we frame TMEM-131 as a factor linking collagen biology to age-dependent ER remodeling, rather than claiming to have established a new canonical ER-phagy receptor. We also expand the discussion to include a balanced interpretation of its similarities and departures from other classical receptors.

1b. Additionally, existing study indicates TMEM-131 interacts with COPII vesicles to facilitate collagen secretion (PMID: 32095531) and COPII components are related to distinct ER clearance pathways (PMID: 31273116, 38593803). It is unclear whether TMEM-131 regulates COPII vesicle formation and whether core COPII coat proteins are required for ER clearance.

We thank the reviewer for raising this important point. TMEM-131 has indeed been linked to COPII-mediated collagen trafficking through its interaction with TRAPPC8, raising the possibility that its effects on ER loss reflect COPII vesicle formation. To address this, our candidate screen included the COPII coat proteins *sec-23*, *sec-24*, and *sec-31*. None of these knockdowns suppressed ER loss in aging or mTOR-inhibited animals, whereas *tmem-131* RNAi consistently and specifically rescued ER phenotypes (ED Fig. 8C vs. Fig. 7). These results argue that TMEM-131's role in ER remodeling is mechanistically separable from general COPII activity. We have revised the text to explicitly discuss COPII factors and to soften language around TMEM-131 as a canonical ER-phagy receptor. As with other established receptors, we believe that unambiguous dissection of TMEM-131's roles in collagen secretion, COPII trafficking, and ER turnover will require a dedicated study. Here, we emphasize its identification as a novel ER-phagy regulator and highlight its potential to act either as a receptor itself or as part of a recruitment complex, meriting further exploration across systems.

2. More evidence about whether TMEM-131 affects bulk autophagy substrates including SQST-1 and LGG-1 is required to strengthen TMEM-131 selectively regulates ER clearance.

We thank the reviewer for identifying this important experiment. We have now performed Western blot analysis of LGG-1 as suggested to investigate *tmem-131* effects on autophagic flux (ED Fig. 8I-L). Consistent with an intact or enhanced basal autophagy, GFP::LGG-1 processing is increased under *tmem-131* RNAi (total and lipidated LGG-1 decrease, while free GFP accumulation rises). These results indicate that autophagy is not globally impaired during *tmem-131* depletion and support that TMEM-131's role is specific to the ER.

3. The TEM results clearly showed ER level decrease in aging worm body but lack key evidence that ER structures exist in autophagosomes. Given the significant decrease of ER during aging, ER structures should be seen enclosed by autophagosomes. If autophagosomes undergo rapid clearance in the worm body, employing genetic or chemical treatments that block lysosomal

function and inhibit autophagosome degradation could increase the chances of capturing autophagosomes by TEM.

We thank the reviewer for raising this point. To directly assess the presence of ER within autophagosomes during aging, we have performed TEM analysis in worms during aging. Consistent with the high rate of ER-phagy predicted for the substantial decline in ER mass, we observed autolysosomal structures containing multilamellar membrane whorls consistent with ER-derived cargo in sections from 4/5 animals (Fig. 5K). These observations are consistent with the fluorescence imaging revealing lysosomal containment of ER, which we have also moved adjacent to the new EM panel (Fig. 5J), and our conclusion that ER-phagy contributes to age-onset ER loss in vivo.

Minor:

1. Biochemistry experiments indicate more RET-1 is degraded compared to SEC61-B (Fig. S2 I-L) that are not consistent with the observations by fluorescent analysis. The authors need to test whether TMEM-131, UNC-51, or LGG-1 affects RET-1 and SEC61-B protein levels via western blot in aging worms to strengthen that the degradation of these ER subdomain proteins is mainly dependent of autophagy. Descriptions about Fig S2 I-L in the main text are missing.

We thank the reviewer for this careful observation. We appreciate that at first glance the RET-1 and SEC-61B Western blot data may appear to contrast with our fluorescence imaging. However, we note that the blots are performed on whole-animal lysates, thus reflecting the net change in protein levels across all tissues and emphasizing larger tissues. In our imaging, we observe that RET-1 declines far more steeply than SEC-61B in the intestine, the largest tissue in the animal. Given these spatial differences, we propose that the biochemical and imaging results are in fact consistent when interpreted in the proper tissue context. Additionally, we provide new Western blot for an alternative tubule-shaping factor, YOP-1, revealing it to be more stable across age and providing an explanation for how tubules can be maintained in the face of a large loss of RET-1 (ED Fig. 2M-N). Because the imaging data allow tissue-specific resolution using native gene fusions, we consider these measurements more reliable for assessing age-related dynamics of RET-1 and SEC-61B in the contexts most relevant to our study. We have revised the manuscript to clarify these interpretations and better avoid confusion.

2. A previous study indicates that IRE-1–XBP-1 mediated ER-UPR is activated in *tmem-131* mutant worms that is caused by the failure of collagen secretion (PMID: 32095531). Is IRE-1 required for TMEM-131 dependent ER clearance in hypodermis?

We thank the reviewer for highlighting this intriguing question. We find that *ire-1/xbp-1* suppresses ER loss in the intestine but not the hypodermis, whereas *tmem-131* shows the opposite specificity (Fig. 7, ED Fig. 9). Thus, IRE-1 is not required for ER clearance in the same cells that depend on TMEM-131. Nevertheless, as the reviewer notes, *ire-1/xbp-1* responses are activated as a compensatory mechanism when *tmem-131* is mutated and collagen export is reduced, likely producing wide-ranging outcomes including enhanced basal ER turnover (ED Fig. 8H-L), as we show in the revised manuscript. To extend this point, we have added new data showing that combined loss of *tmem-131* and *xbp-1* produces synthetic lifespan effects (Fig. 8), indicating that the two pathways provide parallel protective functions. Overall, we agree that dissecting the interplay between *tmem-131* and *ire-1* (and more broadly between ER biogenesis and ER turnover) is an important next step. However, because *ire-1/xbp-1* has broad pleiotropic roles, we believe that interpreting these interactions will require identification of ER-phagy-specific mediators downstream of *ire-1/xbp-1*. As this lies beyond the focus of the current study, we propose it is best addressed in future work.

3. Line 303, 307, 314 “Fig. S6M-Q” are not related to the figure.

We thank the reviewer for catching this mistake and have corrected figure references.

We thank all the reviewers and editors for their thoughtful evaluation and continued enthusiasm for our study. Here we have made attempts to address all remaining minor concerns with new data. We have incorporated additional quantification of TEM data, new fluorescence imaging of mitochondria to test the specificity of TMEM-131 effects on organelle turnover, and exploratory imaging of animals co-labeled for RET-1 and LGG-1 to help clarify how ER proteins behave during autophagy induction. We made targeted revisions to the results and discussion regarding the explanation of LGG-1 Western blots and the need to spatially and temporally map TMEM-131's place in ER-phagy induction and autophagy recruitment as a key next step in understanding how it promotes age-onset ER clearance.

Reviewers' comments:

Reviewer #1 (Remarks to the Author):

The authors have thoroughly addressed my suggestions and comments, and the revised manuscript is much more improved. I recommend accepting it for publication.

We thank the reviewer for their feedback and support.

Reviewer #2 (Remarks to the Author):

In the revised version of the manuscript by Donahue EKF et al., the authors have adequately addressed all of my previous comments and concerns. The addition of new data further strengthens their main conclusions regarding the key role of ER-phagy in ER remodeling during aging. I commend the authors on their thorough revisions and recommend that this version be accepted for publication.

We thank the reviewer for their feedback and support.

Reviewer #3 (Remarks to the Author):

The authors have addressed the key concerns. However, some minor issues need to be addressed:

1. The TEM results (Extended Data Figure 4) show a loss of cellular mass in aging worms. Both the ER and mitochondria appear reduced in the hypodermis and intestines of aged animals, suggesting that both organelles are cleared during aging. Does *tmem-131* affect mitochondrial clearance during aging? Quantification of the TEM results is needed.

We thank the reviewer for raising the issue of whether *tmem-131*'s effects on organelle clearance are specific to ER. First, as suggested we have quantified the ER footprint shown in new intestinal TEM images (Ext Data Fig 4G), which agree with the light microscopy and hypodermal TEM results,

revealing a pronounced decline. To test for the specificity of *tmem-131*'s effects on ER clearance, we performed live microscopy in the hypodermis of an endogenously labeled mitochondrial protein, COX-4::GFP. We find that *tmem-131(RNAi)* does have a mild impact on mitochondrial morphology (A), resulting in a reduced organelle footprint at baseline (B). In contrast to ER, this is consistent with enhanced or stable turnover of mitochondria. With age (day 7), *tmem-131(RNAi)* results in apparent acceleration of a classical fragmented mitochondrial morphology, which we commonly observe in control

Figure 1. A) Representative confocal images of COX-4::GFP in the hypodermis of young (d1) and aged (d7) worms fed either empty vector control or *tmem-131* dsRNA. B) Footprint measurement of COX-4::GFP. N = 21 per condition pooled across 3 independent repeats; analyzed via 2-way ANOVA.

animals at later ages. This precocious fragmentation is also consistent with our model that *tmem-131*-supported ER-phagy is a protective process, and with our speculation that early shifts in ER homeostasis are likely to cause 'trickle-down' effects on other organelle networks, which we plan to pursue further in future dedicated studies. In contrast to what we observe for the ER, however, the mitochondrial footprint remains steady across age, supporting a model where *tmem-131* selectively promotes turnover of the ER. We have included these data in Extended Data Fig. 8.

2. The Western blot results (Extended Data Figure 8) do not provide conclusive evidence supporting a role for *tmem-131* in selective autophagy. The presence of processed free GFP, cleaved from GFP-LGG-1 by lysosomal hydrolyses, indicates that *tmem-131* RNAi worms show a decrease of main GFP-LGG1 band accompanied by an increase in free GFP compared with controls, suggesting enhanced autophagic flux. This observation appears inconsistent with the author's proposed model. Nevertheless, *tmem-131* may function specifically in regulating ER clearance within certain tissues. Since Western blot analysis of whole animals cannot resolve tissue specific effects, it would be informative to examine whether *tmem-131* RNAi alters LGG-1 co-localization with ER markers such as SEC-61.B or RET-1, given that LGG-1 is recruited to *tmem-131* structures by immunostaining (Extended Data Figure 8).

We thank the reviewer and agree with this interpretation of the Western blot data that *tmem-131(RNAi)* maintains or even enhances bulk autophagic flux, while suppressing the ER-phagy dependent loss of ER. We reason that the bulk autophagic flux may be upregulated in response to collagen defects when TMEM-131 is impaired. Yet, in contrast to other collagen maturation factors, ER mass is uniquely retained when *tmem-131* is depleted, suggesting it is an important factor in the mechanism targeting the ER for turnover. Importantly, we note that for canonical ER-phagy mediators, selective ER turnover is commonly uncoupled from bulk autophagy readouts (e.g., PMC6747008, PMC6168278), so we do not necessarily see these observations as contradictory. We have revised our results to help clarify our model.

To examine autophagic targeting of other ER proteins selectively in the hypodermis, we crossed RET-1::mKate2 and GFP::LGG-1 animals as suggested. We also acutely starved these animals (24 h) to induce higher ER-phagy levels. Notably, while TMEM-131::mKate2 previously showed a distinct redistribution upon starvation, including formation of higher-intensity patches and puncta that partially colocalize with LGG-1, this was not apparent in RET-1, which maintained a consistent patterning and did not exhibit obvious co-labeled puncta (A,B). We believe these differences help to support a model where TMEM-131 plays a more direct role in ER-phagy. However, we do observe a decline in overall RET-1 at 24 hours (C), indicating that it is being turned over. This loss of RET-1 in the absence of obvious LGG-1 colocalization could be due to a combination of factors, including assay timing, ER protein/marker specificity and/or contributions from autophagosome-independent ER-phagy (ERLAD, micro-ER-phagy). This assay thus requires potentially significant optimization and/or improved tools, which we believe may extend beyond the scope of a minor point. Overall, we agree that it is a key next step to spatially and temporally resolve ER-phagy events during aging and in response to perturbations in *tmem-131*, and we hope to pursue this work in future studies specifically designed to genetically dissect the functions of *tmem-131*. We have raised this in the discussion as a current limitation and key next step.

Figure 2 A) Representative confocal images of ER and autophagosome reporters in the dermis before and after acute starvation. B) Co-localization analysis of RET and LGG-1. C) RET-1 footprint analysis. N = 24 (0 hr) and 28 (24 h) animals pooled from 3 independent repeats; analyzed by t-test.